# Achieving Tighter Finite-Time Rates for Heterogeneous Federated Stochastic Approximation under Markovian Sampling

**Feng Zhu**                                                                    *fzhu5@ncsu.edu*
*Department of Electrical and Computer Engineering*
*North Carolina State University*

**Aritra Mitra**                                                                *amitra2@ncsu.edu*
*Department of Electrical and Computer Engineering*
*North Carolina State University*

**Robert W. Heath Jr.**                                                         *rwheathjr@ucsd.edu*
*Department of Electrical and Computer Engineering*
*University of California, San Diego*

**Reviewed on OpenReview:** *https://openreview.net/forum?id=1xRG4ECacS*

## Abstract

Motivated by collaborative reinforcement learning (RL) and optimization with time-correlated data, we study a generic federated stochastic approximation problem involving $M$ agents, where each agent is characterized by an agent-specific (potentially nonlinear) local operator. The goal is for the agents to communicate intermittently via a server to find the root of the average of the agents' local operators. The generality of our setting stems from allowing for (i) Markovian data at each agent and (ii) heterogeneity in the roots of the agents' local operators. The limited recent work that has accounted for both these features in a federated setting fails to guarantee convergence to the desired point or to show any benefit of collaboration; furthermore, they rely on projection steps in their algorithms to guarantee bounded iterates. Our work overcomes each of these limitations. We develop a novel algorithm called `FedHSA`, and prove that it guarantees convergence to the correct point, while enjoying an $M$-fold linear speedup in sample-complexity due to collaboration. To our knowledge, *this is the first finite-time result of its kind*, and establishing it (without relying on a projection step) entails a fairly intricate argument that accounts for the interplay between complex temporal correlations due to Markovian sampling, multiple local steps to save communication, and the drift-effects induced by heterogeneous local operators. Our results have implications for a broad class of heterogeneous federated RL problems (e.g., policy evaluation and control) with function approximation, where the agents' Markov decision processes can differ in their probability transition kernels and reward functions.

## 1 Introduction

In the classical stochastic approximation (SA) formulation (Robbins & Monro, 1951; Borkar, 2009; Borkar & Meyn, 2000; Meyn, 2023), the goal is to solve for a parameter $\theta^\star \in \mathbb{R}^d$ such that $\bar{G}(\theta^\star) = 0$, where $\bar{G} : \mathbb{R}^d \to \mathbb{R}^d$ is a potentially nonlinear operator satisfying $\bar{G}(\cdot) = \mathbb{E}_{o\sim\mu}[G(\cdot,o)]$. Here, $G : \mathbb{R}^d \times \mathcal{X} \to \mathbb{R}^d$ is the noisy version of the true operator $\bar{G}$, and $o$ is an observation random variable drawn with some *unknown* distribution $\mu$ from a sample space $\mathcal{X}$. The agent (learner) tasked with finding $\theta^\star$ only has access to the operator $\bar{G}$ via noisy samples $\{G(\cdot, o_t)\}$, where $\{o_t\}$ is a stochastic observation process that is typically assumed to converge in distribution to $\mu$ (Borkar, 2009). Interestingly, the SA formulation described above captures a large class of problems arising in stochastic optimization, control, and reinforcement learning

(RL). For instance, popular iterative algorithms like stochastic gradient descent (SGD) in optimization, and temporal difference (TD) learning, Q-learning in RL turn out to be specific instances of SA.

Federated learning (FL) has also made extensive use of SA. The general idea of FL is to leverage data collected from multiple agents to train accurate statistical models for downstream prediction (Konečnỳ et al., 2016; Bonawitz et al., 2019; McMahan et al., 2017). SA has primarily been used in FL to solve stochastic optimization problems that arise in the context of empirical risk minimization (ERM), under assumptions like offline data collection and i.i.d. (independent and identically distributed) data at each agent. In contrast, when the data arrives sequentially and exhibits strong temporal correlations - as is the case in multi-agent/federated RL - very little is understood about non-asymptotic/finite-time performance. This is particularly the case when the agents' data-generating processes are potentially non-identical. *In our paper, we develop convergence and sample-complexity results for a general class of federated SA methods that significantly improve upon existing bounds, and provide a deeper understanding of the interplay between data-heterogeneity and temporal correlations in FL.*

**Our Federated SA Setup.** We aim to develop a unified framework that can accommodate a broad class of distributed SA problems. To that end, we consider a setup involving $M$ agents that can exchange information via a central server, as is typical in an FL setting. Each agent $i \in [M]$ has its own true and noisy operators $\bar{G}_i : \mathbb{R}^d \to \mathbb{R}^d$ and $G_i : \mathbb{R}^d \times \mathcal{X}_i \to \mathbb{R}^d$, respectively, where $\bar{G}_i(\cdot) = \mathbb{E}_{o \sim \mu_i}[G_i(\cdot, o)]$. Here, $\mathcal{X}_i$ is the sample space of agent $i$, and the observation random variable $o$ is drawn from an *agent-specific* unknown distribution $\mu_i$. The information available locally at an agent $i$ comprises a sequence of noisy samples $\{G_i(\cdot, o_{i,t})\}$. To capture temporal correlations, we assume that the observation sequence $\{o_{i,t}\}$ is generated from an ergodic Markov chain $\mathcal{M}_i$ with stationary distribution $\mu_i$. Given this setup, the collective objective is to find the root $\theta^\star \in \mathbb{R}^d$ of the "average" operator $\bar{G}$; this objective can be succinctly stated as

$$\text{Find } \theta^\star \text{ s.t. } \bar{G}(\theta^\star) = 0, \text{where } \bar{G} := \frac{1}{M} \sum_{i=1}^{M} \bar{G}_i. \tag{1}$$

Note that to achieve the above objective, communication via the server is *necessary*. In a typical FL scenario, communication takes place over low-bandwidth channels, and can hence be costly and slow. To mitigate the communication bottleneck, we will adhere to the standard *intermittent* communication protocol in FL, where an exchange of information with the server takes places only once in every $H > 1$ time-steps, where $H$ is some predefined synchronization time-period. To summarize, there are three main features in our formulation: (1) **Heterogeneity:** The local operators $\{\bar{G}_i\}$ at the agents can be non-identical and, as such, have different roots; (2) **Markovian Sampling:** The agents' observation sequences are generated in a Markovian manner; and (3) **Sparse Communication:** Agents need to perform local computations in isolation between communication rounds. Additionally, complying with the FL paradigm, the agents are required to solve the problem in (1) without ever exchanging their raw observations.

## 1.1 Motivation and Related Work

When one considers the challenging setting that involves all of the above features, there are significant gaps in the understanding of the problem in (1). We now explain these gaps by considering several motivating applications of (1). Along the way, we also review relevant literature.

1. **Motivation 1: Federated Optimization in Dynamic Environments.**

   - **I.I.D. Data.** When $\bar{G}_i$ is the gradient of a local loss function at agent $i$, and $G_i$ a noisy stochastic version thereof, the problem in (1) reduces to the standard FL setting that has been extensively studied under the assumption that the observation process $\{o_{i,t}\}$ at each agent $i$ is generated in an i.i.d. manner from $\mu_i$ (McMahan et al., 2017). Under this i.i.d. data regime, a large body of work has studied the heterogeneous federated optimization problem (Li et al., 2019; Khaled et al., 2020; Karimireddy et al., 2020; Mitra et al., 2021; Gorbunov et al., 2021; Mishchenko et al., 2022).

- **Markovian Data.** Our formulation substantially generalizes the standard FL setting by allowing the observation process at each agent $i$ to be generated from an ergodic Markov chain that converges in distribution to $\mu_i$. As explained in detail in Sun et al. (2018) and Duchi et al. (2012), there is ample reason to consider such Markovian data streams. Moreover, temporally correlated Markovian data arises naturally in several real-world applications: dynamic environments such as non-stationary wireless systems where the channel statistics drift over time following Gauss-Markovian processes, linear dynamical systems in control with input noise, and distributed sensor networks with correlated measurements. With these reasons in mind, considerable recent attention has been given to the study of Markovian gradient descent in the single-agent setting (Sun et al., 2018; Doan, 2022; Even, 2023; Beznosikov et al., 2024).

  Research Gap: Counterparts of the results in the above papers remain open in FL, i.e., *we are unaware of any finite-time results for federated optimization under Markov data.*

2. **Motivation 2: Heterogeneous Fixed Point Problems.** Our work is particularly inspired by that of Malinovskiy et al. (2020), where the general problem of finding the fixed point of an average of local operators is considered, exactly like we do in (1). As detailed in Malinovskiy et al. (2020), in addition to optimization, heterogeneous fixed-point problems find applications in many other settings such as alternating minimization methods, nonlinear inverse problems, variational inequalities, and Nash equilibria computation.

   Research Gap: The main gap, however, is that Malinovskiy et al. (2020) consider a "noiseless" setting, where each agent can access their true local operator directly, i.e., there is no aspect of i.i.d. or Markovian sampling in their work.

3. **Motivation 3: Collaborative Multi-Agent RL.** In RL, an agent interacts sequentially with an environment modeled as a Markov decision process (MDP). At each time-step, the agent plays an action according to some policy, observes a reward, and transitions to a new state. The goal is for the agent to play a sequence of actions that maximizes some long-term cumulative utility referred to as the *value-function*, without a priori knowledge of the reward functions and the probability transition kernels of the MDP. It turns out that the problem of estimating the value-function corresponding to a given policy (policy evaluation), and that of finding the optimal policy (control), can be both cast as instances of SA (Meyn, 2023). In this context, the sample-complexities of SA-based RL algorithms like TD learning and Q-learning have been studied recently by Bhandari et al. (2018); Srikant & Ying (2019); Khamaru et al. (2020); Wainwright (2019); Guannan Qu (2020); Li et al. (2024); Mitra (2024). This line of work has collectively revealed that for contemporary RL problems with large state and action-spaces, several samples are typically needed to achieve a desired performance accuracy. A natural way to overcome the sample-complexity barrier is via parallel data collection from multiple environments. This has led to a new paradigm called *federated reinforcement learning* (FRL) (Qi et al., 2021), where the idea is to use information from multiple environments (MDPs) to learn a policy that performs "well" on average in all MDPs. Compared to FL, the theoretical aspects of FRL remain poorly understood. To explain the gaps in this context, we broadly categorize SA problems in FRL into two main groups as follows; for a quick comparative summary, see Table 1.

   - **Homogeneous Environments.** In this case, agents share identical MDPs and local operators. The works of Doan et al. (2019) and Liu & Olshevsky (2023) both examine federated TD algorithms under a restrictive i.i.d. sampling assumption. While Shen et al. (2023) analyze federated actor-critic algorithms under both i.i.d. and Markov data, they establish a linear speedup in sample-complexity (w.r.t. the number of agents) only under i.i.d. data. To our knowledge, the first paper to establish a linear speedup in sample-complexity under Markovian sampling for contractive SA problems was (Khodadadian et al., 2022), followed by (Woo et al., 2023) for tabular Q-learning. Follow-up works have focused on the minimum amount of communication needed to achieve linear speedups (Tian et al., 2024; Salgia & Chi, 2024) in sample-complexity, and also the effect of imperfect communication channels (Dal Fabbro et al., 2023; Mitra et al., 2024; Beikmohammadi et al., 2024; Dal Fabbro et al., 2025).

Table 1: Comparison of finite-time analysis for FRL papers that study stochastic approximation problems.

| Work | Heterogeneity | No heterogeneity bias | Linear speedup | Markovian sampling |
|---|---|---|---|---|
| (Doan et al., 2019) | ✗ | - | ✗ | ✗ |
| (Liu & Olshevsky, 2023; Shen et al., 2023) | ✗ | - | ✓ | ✗ |
| (Khodadadian et al., 2022; Woo et al., 2023; Salgia & Chi, 2024; Tian et al., 2024) | ✗ | - | ✓ | ✓ |
| (Jin et al., 2022) | ✓ | ✗ | ✗ | ✗ |
| (Wang et al., 2024b; Zhang et al., 2024) | ✓ | ✗ | ✓ | ✓ |
| This work | ✓ | ✓ | ✓ | ✓ |

- **Heterogeneous Environments.** In practice, it is unreasonable to expect that the agents' environments are *exactly the same.* Thus, it makes more sense to consider a setting where agents interact with potentially distinct MDPs, *leading to distinct local operators.* This setting has received much less attention. The convergence of federated Q-learning algorithms is analyzed in Jin et al. (2022), but no linear speedup is established. The results in Wang et al. (2024b) for federated TD learning, and Zhang et al. (2024) for federated SARSA, do exhibit a linear speedup in the noise variance term, but their bounds also feature an additive heterogeneity-induced bias term that grows with the discrepancy in the agents' environments; such a bias term also appears in Jin et al. (2022), and its presence precludes possible gains from collaboration.

  Research Gap: To sum up, in the context of general heterogeneous stochastic approximation problems in FRL under Markov noise, it remains an open problem to establish a linear collaborative speedup that is not affected by any heterogeneity-induced bias term.

Now that we have elaborated on the motivation for our work and the research gaps in the literature, let us turn our attention to the main questions investigated in this paper.

**Key Questions.** Having motivated the rationale behind studying the problem in (1), we can now state more precisely the key questions of interest to us. To do so, we start by presenting the convergence rates of SA in the single-agent case as a benchmark. Accordingly, consider the SA scheme of Robbins & Monro (1951):

$$\theta^{(t+1)} = \theta^{(t)} + \alpha_t G(\theta^{(t)}, o_t), \tag{2}$$

for $t = 0, ..., T-1$. Here, $T$ denotes the total number of iterations, $\theta^{(t)} \in \mathbb{R}^d$ is the parameter estimate at time-step $t$, and $\{\alpha_t\}$ is the step-size sequence. Asymptotic convergence results for the SA rule in (2) were derived in the seminal works of Tsitsiklis & Van Roy (1997) and Borkar & Meyn (2000). Finite-time rates under Markovian observations were more recently established in Bhandari et al. (2018); Srikant & Ying (2019); Chen et al. (2022); Mitra (2024), where it was shown that under certain mild technical assumptions (which will be detailed later in Section 2), running $T$ iterations of the rule in (2) with the same step-size $\alpha_t = \alpha$ leads to the following error bound:

$$d_T \leq \underbrace{C_1 \exp\left(-\alpha C_2 T\right)}_{\text{bias}} + \underbrace{\alpha C_3 \sigma^2}_{\text{variance}}, \tag{3}$$

where $d_t := \mathbb{E}\left[\left\|\theta^{(t)} - \theta^\star\right\|_2^2\right]$ denotes the MSE at time-step $t$, $C_1, C_2, C_3$ are problem-specific constants, and $\sigma^2$ characterizes the variance of the noise model. The MSE bound consists of two components: (i) a bias term that decays exponentially fast and (ii) a variance term that captures the effect of noise. Thus, the rule in (2) ensures linear convergence to a ball of radius $\mathcal{O}\left(\alpha\sigma^2\right)$ centered around $\theta^\star$. By choosing an $\alpha$ on the order of $\mathcal{O}\left(\log(T)/T\right)$, one can then achieve exact convergence to $\theta^\star$. Now consider the federated SA setup with heterogeneous local operators, Markovian data, and intermittent communication. We ask:

*Is it possible to converge exactly (i.e., without any bias) in the mean-square sense to the root $\theta^\star$ of the average operator $\bar{G}$? Moreover, can one achieve a linear $M$-fold reduction in the variance term, capturing the benefit of collaboration?*

**Contributions.** As far as we are aware, no prior work has been able to establish both *exact convergence* and *linear sample-complexity speedups* in FL under Markovian sampling and heterogeneity of local operators. We close this significant gap via the following contributions.

● **Motivation for New Algorithm.** In Section 2.2, we study the performance of existing FRL algorithms (Jin et al., 2022; Khodadadian et al., 2022; Wang et al., 2024b; Zhang et al., 2024), where each agent performs multiple local parameter updates using just its own operator. Proposition 1 reveals that in a heterogeneous setting, such algorithms fail to match the single-agent SA convergence rate in (3). In particular, when run with a constant non-diminishing step size $\alpha$, such algorithms do not converge exactly to the root $\theta^\star$, even in the absence of noise.

● **Novel Algorithm.** Motivated by the findings from Section 2.2, we develop a new local SA procedure called `FedHSA` in Section 3. The core idea in `FedHSA` is to modify the local update rule to ensure convergence to the correct point $\theta^\star$. We emphasize that while drift-mitigation techniques to combat heterogeneity have been studied before in federated optimization (Karimireddy et al., 2020; Gorbunov et al., 2021; Mitra et al., 2021), `FedHSA` *is not limited to optimization*. Instead, `FedHSA` applies much more broadly to general nonlinear SA problems, including those in RL.

● **Matching Centralized Rates and Linear Speedup.** In Theorem 2 of Section 4, we prove that `FedHSA` matches the centralized rate in (3), and, unlike the results in Jin et al. (2022); Wang et al. (2024b); Zhang et al. (2024), there is *no heterogeneity-induced bias in our final bound*. Furthermore, with a linearly decaying step-size, we prove that `FedHSA` achieves an optimal sample-complexity bound of $\tilde{\mathcal{O}}(1/(MHT))$ after $T$ communication rounds, with $H$ local steps in each round (Corollary 1). *This result is significant because it is the first to establish a collaborative $M$-fold linear speedup for heterogeneous federated SA problems under Markovian sampling, with no additional bias term to negate the benefit of collaboration.*

● **Analysis Technique.** Even in the single-agent SA setting, a finite-time analysis under Markovian data is quite non-trivial. Our FL setting is further complicated by complex statistical correlations that arise from combining data generated by distinct Markov chains, drift effects due to heterogeneous local operators, and multiple local steps. The only other recent papers (Wang et al., 2024b; Zhang et al., 2024) that consider similar settings rely on a projection step in the algorithm to ensure uniform boundedness of iterates. Furthermore, they build on a "virtual MDP framework" for their analysis. Our proof neither requires a projection step nor a virtual MDP. In particular, the lack of a projection step entails a much more involved analysis. We elaborate on the main technical challenges in Section 5.

Although our work is primarily theoretical, we corroborate our main theoretical findings via numerical simulations, covering both optimization and RL, in Section D.

**More Related Work.** Since the focus of this paper is on federated stochastic approximation, we have only reviewed SA-based approaches in FRL. We note that policy gradient-based approaches for FRL have also recently been explored in Xie & Song (2023); Fan et al. (2021); Lan et al. (2023); Wang et al. (2024a); Zhu et al. (2024). In particular, the approaches developed in Wang et al. (2024a) and Zhu et al. (2024) manage to achieve collaborative speedups without incurring additive bias terms due to heterogeneity. That said, even in the single-agent setting, the dynamics of policy gradient methods and SA schemes in RL are considerably different, and entail separate treatments. For instance, the aspect of Markovian sampling does not show up at all in policy-gradient methods.

At the time of preparing this paper, we became aware of a very recent piece of work by Mangold et al. (2024) that looks at federated *linear* stochastic approximation (LSA). In this work, the authors provide a detailed and refined analysis of local update rules - of the form of `FedAvg` - under both i.i.d. and Markov sampling. Furthermore, they develop a bias-corrected algorithm for federated LSA, and show that it simultaneously achieves a linear speedup and no heterogeneity-induced bias. Although the flavor of the results in Mangold et al. (2024) is similar to that of our work, there are significant differences in scope, algorithms, and proof techniques that we outline below. First, the results in Mangold et al. (2024) are limited to linear stochastic approximation (LSA). In contrast, we consider general nonlinear operators throughout this work. An inspection of the refined analysis in Mangold et al. (2024) seems to suggest that the recurrence relations exploited in Mangold et al. (2024) rely heavily on the linearity of the underlying operator. More precisely, the approach in Mangold et al. (2024) is based on the framework in Durmus et al. (2025) which relies on expressing the error dynamics as a linear time-varying system that involves a product of random matrices. Such an error recursion does not arise in our analysis which involves non-linear operators, thereby requiring a different proof approach right from the first step. Second, the results under Markov sampling in Mangold et al.

(2024) are provided only for the vanilla federated LSA schemes, not for the bias-corrected one. Thus, even in light of the contributions made in Mangold et al. (2024), our work is the first to establish a linear speedup result without heterogeneity bias for general (nonlinear) contractive SA under Markov sampling. Third, the authors in Mangold et al. (2024) use the "blocking technique" to handle temporal correlations under Markov sampling. This requires modifying the original algorithm so that it only operates on a sub-sampled data sequence, where the sub-sampling gap is informed by the mixing time of the underlying Markov chain. In contrast, our proof technique is more direct, does not go through the blocking apparatus, and, as such, requires no modification to the algorithm for Markov data.

## 2 Setting and Motivation

### 2.1 Setting

We consider a multi-agent heterogeneous SA problem involving $M$ agents, where each agent $i \in [M]$ has its own local true operator $\bar{G}_i$. Since the true operators are generally hard to evaluate exactly, each agent $i$ can access $\bar{G}_i$ only through a sequence of noisy samples $\{G_i(\cdot, o_{i,t})\}$. The observation $o_{i,t}$ made at time-step $t$ by agent $i$ is sequentially sampled from an underlying *agent-specific* time-homogeneous Markov chain $\mathcal{M}_i$ with stationary distribution $\mu_i$. We further have $\bar{G}_i(\cdot) = \mathbb{E}_{o \sim \mu_i}[G_i(\cdot, o)]$. We consider the case where the agents' Markov chains $\{\mathcal{M}_i\}$ share a common *finite* state space $\mathcal{S}$, but have potentially different probability transition matrices. The collaborative goal is to solve the root-finding problem described in (1) within a federated framework, where the agents communicate intermittently via a central server only once in every $H$ time-steps, while keeping their raw observation sequences $\{o_{i,t}\}$ private.

**Working Assumptions.** We now make certain standard assumptions on the agents' operators and stochastic observation processes for our subsequent analysis.

**Assumption 1** (Lipschitzness)**.** *The local noisy operator $G_i$ for each agent $i \in [M]$ is L-Lipschitz, i.e., there exists a constant $L \geq 1$ such that given any observation $o$, for all $\theta_1, \theta_2 \in \mathbb{R}^d$, we have*

$$\|G_i(\theta_1, o) - G_i(\theta_2, o)\|_2 \leq L \|\theta_1 - \theta_2\|_2. \tag{4}$$

*Furthermore, there exists $\sigma_i \geq 1$ for each $i \in [M]$ such that for any given $\theta, o$, the following holds*

$$\|G_i(\theta, o)\|_2 \leq L(\|\theta\|_2 + \sigma_i). \tag{5}$$

**Assumption 2** (1-point strong monotonicity)**.** *The average true operator $\bar{G}$ is 1-point strongly monotone w.r.t. $\theta^\star$, i.e., there exists some constant $\mu \in (0, 1]$ such that for any $\theta \in \mathbb{R}^d$, we have*

$$\langle \theta - \theta^\star, \bar{G}(\theta) - \bar{G}(\theta^\star) \rangle \leq -\mu \|\theta - \theta^\star\|_2^2. \tag{6}$$

In the context of optimization, Assumption 1 corresponds to a smoothness assumption typical in the analysis of Markovian gradient descent (Doan, 2022), and Assumption 2 corresponds to strong-convexity. As for RL, Assumptions 1 and 2 both hold for TD learning with linear function approximation (LFA) (Bhandari et al., 2018; Srikant & Ying, 2019; Khamaru et al., 2020; Tsitsiklis & Van Roy, 1997; Doan et al., 2019; Liu & Olshevsky, 2023; Khodadadian et al., 2022; Tian et al., 2024; Wang et al., 2024b), and certain variants of Q-learning with LFA (Chen et al., 2022; Zeng et al., 2022), where $\bar{G}_i$ and $G_i$ correspond to the non-noisy and noisy versions, respectively, of the TD/Q-learning update rules. Assumptions 1 and 2 suffice to guarantee the MSE bound in (3) in the centralized case, i.e., when $M = 1$. Since our objective is to obtain such a bound for the multi-agent heterogeneous setting, it is natural for us to work under the same assumptions.

The next assumption on the agents' Markov chains helps control the effect of temporal correlations in the data, and appears in almost all finite-time analysis papers for both single-agent (Bhandari et al., 2018; Srikant & Ying, 2019; Chen et al., 2022; Adibi et al., 2024; Guannan Qu, 2020; Li et al., 2024) and multi-agent RL (Khodadadian et al., 2022; Zeng et al., 2022; Wang et al., 2024b; Woo et al., 2023; Tian et al., 2024; Zhang et al., 2024; Dal Fabbro et al., 2023).

**Assumption 3.** *For each agent $i \in [M]$, the state space of the underlying Markov chain $\mathcal{M}_i$ is finite, and the Markov chain $\mathcal{M}_i$ is aperiodic and irreducible.*

A key implication of Assumption 3 is that it implies geometric mixing of each of the agents' Markov chains (Levin & Peres, 2017). More precisely, for each $i \in [M]$, there exists some $c_i \geq 1$ and some $\rho_i \in (0,1)$, such that the following is true for any state $s \in \mathcal{S}$:

$$d_{TV}(\mathbb{P}(o_{i,t} = \cdot \mid o_{i,0} = s), \mu_i) \leq c_i \rho_i^t, \ \forall t > 0, \tag{7}$$

where $d_{TV}(P,Q)$ is the total variation distance between two probability measures $P$ and $Q$, $o_{i,t}$ denotes the state of agent $i$ at the $t$-th time-step, and $\mu_i$ is the stationary distribution of $\mathcal{M}_i$. In words, Assumption 3 implies that each agent $i$'s Markov chain converges to its stationary distribution $\mu_i$ exponentially fast.

We now define the concept of a *mixing time* that plays a key role in our analysis. Intuitively, the mixing time of a Markov chain measures how long it takes for the chain to "forget" its starting state and become close to its stationary distribution. For any $\varepsilon > 0$, we define the mixing time of the Markov chain $\mathcal{M}_i$ (at precision level $\varepsilon$) as $\tau_i(\varepsilon) := \min\{t \in \mathbb{N}_0 : c_i \rho_i^t \leq \varepsilon\}$. We define $\tau(\varepsilon) := \max_{i \in [M]} \tau_i(\varepsilon)$ as the mixing time corresponding to the slowest-mixing Markov chain.

Our final assumption concerns statistical independence between the data across different agents. Such an assumption is needed to establish "linear speedups" in performance, and has appeared in prior FRL work (Khodadadian et al., 2022; Woo et al., 2023; Wang et al., 2024b; Zhang et al., 2024).

**Assumption 4.** *For every pair of agents $i \neq j \in [M]$, the observation processes $\{o_{i,t}\}$ and $\{o_{j,t}\}$ are statistically independent.*

With the above assumptions in place, we are in a position to describe and analyze our proposed algorithm. Before doing so, however, it is natural to ask: *What is the need for a new federated SA algorithm?* We provide a concrete answer in the next section.

## 2.2 Motivation for a new Federated SA Algorithm

To explain the motivation for developing our algorithm, it suffices to focus on the class of linear stochastic approximation (LSA) problems, where for each agent $i \in [M]$, $\bar{G}_i(\theta) = \bar{A}_i \theta - \bar{b}_i$, and $G_i(\theta, o_{i,t}) = A_i(o_{i,t})\theta - b_i(o_{i,t})$, i.e., the operators are affine in the parameter $\theta$. Furthermore, to isolate the effect of heterogeneity, we will consider a simplified "noiseless" setting where each agent $i$ can directly access its true operator $\bar{G}_i(\theta)$. Our goal is to formally establish that even for this simplified scenario, if one employs the existing algorithms in Jin et al. (2022); Khodadadian et al. (2022); Wang et al. (2024b); Zhang et al. (2024), then it might be impossible to match the convergence rates in the single-agent setting. To see this, we first note that the algorithms in these papers operate in rounds $t = 0, 1, \ldots, T-1$, where within each round $t$, each agent $i$ performs $H \geq 1$ local model-update steps of the following form:

$$\theta_{i,\ell+1}^{(t)} = \theta_{i,\ell}^{(t)} + \eta \bar{G}_i(\theta_{i,\ell}^{(t)}), \ell = 0, 1, \ldots, H-1, \tag{8}$$

where $\theta_{i,\ell}^{(t)}$ is agent $i$'s parameter estimate in local step $\ell$ of communication round $t$, and $\eta \in (0,1)$ is a step-size. For each $i \in [M]$, $\theta_{i,0}^{(t)}$ is initialized from a common global parameter $\bar{\theta}^{(t)}$. At the end of the $t$-th round, each agent $i$ transmits $\theta_{i,H}^{(t)}$ to the server; the server then broadcasts the next global parameter $\bar{\theta}^{(t+1)} = (1/M) \sum_{i \in [M]} \theta_{i,H}^{(t)}$ to all the agents. We will analyze local SA rules of the form in (8) under the standard assumptions made to derive finite-time rates for linear SA in the single-agent case (Srikant & Ying, 2019): for all $i \in [M]$, (i) (Lipschitzness) the 2-norms of $A_i$ and $b_i$ are bounded, and (ii) (strong-monotonicity) all the eigenvalues of $A_i$ have strictly negative real parts, i.e., $A_i$ is Hurwitz. When $M = 1$, i.e., there is only one agent, and $H = 1$, i.e., communication occurs every time-step, the rule in (8) ensures exponentially fast convergence to the root of the underlying operator (Srikant & Ying, 2019). Our next result reveals that this is no longer the case when $M > 1$ and $H > 1$. To state this result, we define $\bar{A} := (1/M) \sum_{i \in [M]} \bar{A}_i$, $\bar{b} := (1/M) \sum_{i \in [M]} \bar{b}_i$, and $\bar{A}' := (1/M) \sum_{i \in [M]} \bar{A}_i^2$.

**Proposition 1.** *Suppose $M > 1$ and $H = 2$. Consider the local SA update rule in (8), and suppose the step-size $\eta$ is chosen such that the matrix $(I + 2\eta\bar{A} + \eta^2 \bar{A}')$ is Schur-stable[1]. Then, we have:* $\lim_{t \to \infty} \bar{\theta}^{(t)} - \theta^\star = \eta v,$

---

[1] A square matrix is Schur-stable if its eigenvalues all lie within the open unit disk in the complex plane, i.e., if the magnitude of each of its eigenvalues is strictly smaller than one.

---

**Algorithm 1** FedHSA

---

1: **Input:** Local step-size $\eta$, global step-size $\alpha_g$, initial parameter $\bar{\theta}^{(0)}$, initial noisy operator $G(\bar{\theta}^{(0)})$.
2: **for** $t = 0, \ldots, T - 1$ **do**
3:      **for** $i = 1, \ldots, M$ **do**
4:      Agent $i$ initializes its local parameter $\theta_{i,0}^{(t)} = \bar{\theta}^{(t)}$.
5:          **for** $\ell = 0, \ldots, H - 1$ **do**
6:          Agent $i$ observes $o_{i,\ell}^{(t)}$ generated from its Markov chain $\mathcal{M}_i$, and updates $\theta_{i,\ell}^{(t)}$ as per (10).
7:          **end for**
8:      Agent $i$ transmits $\Delta_{i,H}^{(t)} = \theta_{i,H}^{(t)} - \bar{\theta}^{(t)}$ to server.
9:      **end for**
10:     Server broadcasts $\bar{\theta}^{(t+1)}$ computed as in (11).
11:     **for** $i = 1, \ldots, M$ **do**
12:     Agent $i$ transmits $G_i(\bar{\theta}^{(t+1)}, o_{i,0}^{(t+1)})$ to server.
13:     **end for**
14:     Server broadcasts average operator $G(\bar{\theta}^{(t+1)})$.
15: **end for**

---

*where $v = (1/M)(2\bar{A} + \eta\bar{A}')^{-1} \sum_{i \in [M]} \bar{A}_i^2 (\theta_i^\star - \theta^\star)$, $\theta_i^\star = \bar{A}_i^{-1}\bar{b}_i$ is the root of the local operator $\bar{G}_i$, and $\theta^\star = \bar{A}^{-1}\bar{b}$ is the root of the global operator $\bar{G}$.*

The main takeaway from Proposition 1 is that even with just 2 local steps (i.e., $H = 2$) and no noise, in the limit, there is a *non-vanishing error* $\eta v$ that depends on how much each local root $\theta_i^\star$ differs from the global root $\theta^\star$. To eliminate this error, the step-size $\eta$ must be diminished with time, i.e., one cannot afford to use a constant step-size like in the single-agent case. Note, however, if a diminishing step-size sequence is used, although convergence to the desired point $\theta^\star$ is guaranteed, **this will come at the cost of a slower, sublinear convergence rate.** A few crucial comments are in order about this result.

1. The phenomenon observed in Proposition 1 has *nothing to do with noise at all*, i.e., whether the noise is i.i.d. or Markovian is immaterial as far as the main message of Proposition 1 is concerned.

2. The key issue at play in Proposition 1 is that the agents' operators have distinct roots, creating the "client-drift" effect under intermittent communication.

3. Heterogeneous federated optimization is a special case of Proposition 1 In particular, Proposition 1 is stated for linear stochastic approximation (LSA) algorithms, and quadratic optimization is a special case of LSA. An immediate corollary of this point is that even in federated optimization (without noise), if one uses a vanilla local update algorithm (with no bias-correction), then a constant step-size will lead to convergence to an incorrect fixed point, i.e., *the client-drift effect cannot be eliminated.* For optimization, this message has already been conveyed by Charles & Konečný (2021). Proposition 1 simply generalizes this observation to more general SA schemes.

Thus, *there is a gap between the bounds in the single-agent case and those achievable with algorithms of the form in (8) in the heterogeneous federated SA setting.* We now proceed to develop FedHSA that will not only close this gap, but also achieve an optimal $M$-fold sample-complexity reduction due to collaboration.

## 3   Proposed Algorithm: FedHSA

In this section, we will develop our proposed algorithm titled Federated Heterogeneous Stochastic Approximation (FedHSA), designed carefully to account for heterogeneous local operators and intermittent communication. We now elaborate on the steps of FedHSA, outlined in Algorithm 1. FedHSA adheres to the standard intermittent communication model in FL, where communication takes place in rounds $t = 0, 1, \ldots, T - 1$. At the beginning of each round $t$, a central server broadcasts the global parameter $\bar{\theta}^{(t)}$ to all the agents, who then perform $H$ steps of local updates; we will describe the local update process shortly.

We denote the local parameter of agent $i$ at the $\ell$-th local step of the $t$-th communication round as $\theta_{i,\ell}^{(t)}$, with $\theta_{i,0}^{(t)}$ initialized from $\bar{\theta}^{(t)}$. In each local step $\ell$ of round $t$, agent $i$ interacts with its own environment and observes $o_{i,\ell}^{(t)}$ from its Markov chain $\mathcal{M}_i$. Using this observation, agent $i$ computes the noisy operator $G_i(\theta_{i,\ell}^{(t)}, o_{i,\ell}^{(t)})$. Note here that we define each local step as a time-step, and thus $o_{i,\ell}^{(t)}$ can be equivalently denoted as $o_{i,tH+\ell}$.

**The Core Idea.** The core technique involves the local update rule at each agent. As revealed in Section 2.2, if each agent makes local updates by simply taking steps along its own operator, then it can be impossible to converge to $\theta^\star$, while maintaining the same convergence rates as in the centralized setting. However, this is precisely what is done in the existing FRL literature (Jin et al., 2022; Khodadadian et al., 2022; Woo et al., 2023; Wang et al., 2024b; Zhang et al., 2024), where the local update rule takes the form

$$\theta_{i,\ell+1}^{(t)} = \theta_{i,\ell}^{(t)} + \eta G_i(\theta_{i,\ell}^{(t)}, o_{i,\ell}^{(t)}), \tag{9}$$

with $\eta > 0$ being the local step-size. When each agent $i$ follows the update rule in (9) for several local steps, it tends to naturally drift towards the root $\theta_i^\star$ of its own local operator $\bar{G}_i$. As such, the reason why update rules of the form in (9) fail to achieve the desired MSE bound in (3) can be attributed to the following simple observation: *in the heterogeneous setting, the root $\theta^\star$ of the global operator $\bar{G}$ may not coincide with the average $(1/M)\sum_{i\in[M]}\theta_i^\star$ of the roots of the agents' local operators.*

We now develop a drift-mitigation technique that overcomes this issue. To start with, we observe that if each agent had the luxury of talking to the server at every time-step, the ideal update rule of the global parameter would be $\bar{\theta}^{(t+1)} = \bar{\theta}^{(t)} + \alpha_g \eta G(\bar{\theta}^{(t)})$, where $G(\bar{\theta}^{(t)}) := (1/M)\sum_{i\in[M]} G_i(\bar{\theta}^{(t)}, o_{i,t})$. Under the intermittent communication model, however, this is not feasible since an agent $i$ does not have access to the information from the other agents in $[M] \setminus \{i\}$ during each local step. Accordingly, our algorithm exploits the *memory of the global operator $G(\bar{\theta}^{(t)})$ from the beginning of communication round $t$ to guide the local updates of each agent during round $t$.* Here, $G(\bar{\theta}^{(t)}) := (1/M)\sum_{i\in[M]} G_i(\bar{\theta}^{(t)}, o_{i,0}^{(t)})$, and the initial global parameter $\bar{\theta}^{(0)}$ can be *arbitrary*. To be concrete, at each local step $\ell$ of round $t$, agent $i$ adds the correction term $G(\bar{\theta}^{(t)}) - G_i(\bar{\theta}^{(t)}, o_{i,0}^{(t)})$ to its local update direction $G_i(\theta_{i,\ell}^{(t)}, o_{i,\ell}^{(t)})$ to account for drift-effects, leading to the update rule for `FedHSA`:

$$\boxed{\theta_{i,\ell+1}^{(t)} = \theta_{i,\ell}^{(t)} + \eta\left(G_i(\theta_{i,\ell}^{(t)}, o_{i,\ell}^{(t)}) + G(\bar{\theta}^{(t)}) - G_i(\bar{\theta}^{(t)}, o_{i,0}^{(t)})\right).} \tag{10}$$

To gain further intuition about the above rule, suppose for a moment that every agent can access the noiseless versions of their local operators. In this case, the noiseless version of `FedHSA` would take the form: $\theta_{i,\ell+1}^{(t)} = \theta_{i,\ell}^{(t)} + \eta\left(\bar{G}_i(\theta_{i,\ell}^{(t)}) + \bar{G}(\bar{\theta}^{(t)}) - \bar{G}_i(\bar{\theta}^{(t)})\right).$ Now suppose the global parameter $\bar{\theta}^{(t)}$ is $\theta^\star$. Since $\theta_{i,0}^{(t)} = \bar{\theta}^{(t)}$ and $\bar{G}(\theta^\star) = 0$ by definition, observe that all subsequent iterates of the agents remain at $\theta^\star$. Said differently, if one initializes `FedHSA` at $\theta^\star$, the iterates never evolve any further, exactly as desired, i.e., the root $\theta^\star$ of the operator $\bar{G}$ is a *stable equilibrium point* of `FedHSA`. After $H$ local steps, each agent $i$ transmits their local parameter change $\Delta_{i,H}^{(t)} := \theta_{i,H}^{(t)} - \bar{\theta}^{(t)}$ to the central server, and the global parameter $\bar{\theta}(t)$ is updated as follows with global step-size $\alpha_g$:

$$\bar{\theta}^{(t+1)} = \bar{\theta}^{(t)} + \frac{\alpha_g}{M}\sum_{i\in[M]}\Delta_{i,H}^{(t)}. \tag{11}$$

This completes the description of `FedHSA`.

**Remark 1.** *We note that in the federated optimization literature, a variety of techniques have been proposed to account for heterogeneity, e.g., proximal methods in Sahu et al. (2018); Pathak & Wainwright (2020); Mishchenko et al. (2022), and gradient-tracking/variance-reduction in Karimireddy et al. (2020); Mitra et al. (2021); Gorbunov et al. (2021). While the update rule of FedHSA might bear a cosmetic similarity with those in these papers, we note that if one considers SA problems other than optimization, such a similarity ceases to exist. For instance, if one considers TD or Q-learning with linear function approximation, then the corresponding FedHSA rule would be one that has not been analyzed before in FL or FRL.*

## 4  Main Results and Discussion

As a warm-up to our main convergence result for `FedHSA`, we first consider a simpler setting where the observation $o_{i,\ell}^{(t)}$ made by each agent $i \in [M]$ at local iteration $\ell$ and round $t$ is drawn i.i.d. from the stationary distribution $\mu_i$ of its underlying Markov chain $\mathcal{M}_i$. With $d_t := \mathbb{E}\left[\left\|\bar{\theta}^{(t)} - \theta^\star\right\|_2^2\right]$, we have the following result for this setting.

**Theorem 1.** *Suppose Assumptions 1 to 4 hold, and consider the i.i.d. sampling model described above. Define $\alpha = H\eta\alpha_g$ as the effective stepsize, and $\sigma := \max\{\{\sigma_i\}_{i\in[M]}, \|\theta^\star\|_2, 1\}$. Then, there exists a universal constant $C$, such that with $\alpha_g = 1$ and $\eta \leq \mu/(2CL^2H)$, FedHSA guarantees the following $\forall T \geq 0$:*

$$d_T \leq \exp\left(-\frac{\mu}{2}\alpha T\right)\left\|\bar{\theta}^{(0)} - \theta^\star\right\|_2^2 + \mathcal{O}\left(\frac{\alpha L^2}{\mu MH} + \frac{\alpha^2 L^4}{\mu^2}\right)\sigma^2.$$

Next, we present our main convergence result under Markovian sampling.

**Theorem 2** (**Main Result**). *Suppose Assumptions 1 to 4 hold. Define $\bar{\tau} = \tau(\alpha^2)$ and $\rho = \max_{i\in[M]} \rho_i$. Then, there exists a universal constant $C' \geq 1$, such that by selecting $\alpha_g = 1, \eta \leq \mu/(C'\bar{\tau}L^2H)$, the following holds for FedHSA for any $T \geq 2\bar{\tau}$:*

$$d_T \leq \exp\left(-\frac{\mu}{4}\alpha T\right)\mathcal{O}\left(d_0 + \sigma^2\right) + \mathcal{O}\left(\frac{\bar{\tau}\alpha L^2}{\mu MH(1-\rho)} + \frac{\alpha^2 L^4}{\mu^2}\right)\sigma^2. \tag{12}$$

The next result is an immediate corollary of Theorem 2.

**Corollary 1** (**Linear Speedup**). *Suppose all the conditions in Theorem 2 hold. Then, by choosing $\eta = 4\log(MHT)/(\mu HT)$, and $T \geq (4L^2\log(MHT)/\mu^2)\max\{C'\bar{\tau}, MH(1-\rho)/\bar{\tau}\}$, FedHSA guarantees the following for any $T \geq 2\bar{\tau}$:*

$$d_T \leq \tilde{\mathcal{O}}\left(\left(d_0 + \frac{\bar{\tau}L^2\sigma^2}{\mu^2(1-\rho)}\right)\frac{1}{MHT}\right). \tag{13}$$

We provide detailed convergence proofs of Theorems 1 and 2 in Appendices B and C, respectively. We will provide a proof sketch for Theorem 2 shortly in Section 5. Before doing so, several comments are in order.

**Discussion.**  Comparing our bounds for the i.i.d. (Theorem 1) and Markov settings (Theorem 2), we note that the only difference comes from the fact that the noise variance term $\sigma^2$ in the Markov case gets inflated by an additional factor $\bar{\tau}/(1-\rho)$ capturing the rate at which the slowest mixing Markov chain approaches its stationary distribution. Such an inflation by the mixing time is typical for problems with Markov data (Nagaraj et al., 2020).

● *Matching Centralized Rates of Convergence.* Theorem 2 reveals that `FedHSA` guarantees exponentially fast convergence to a ball around $\theta^\star$. In particular, comparing (12) with (3), we conclude that `FedHSA` recovers the known finite-time bounds for single-agent SA in Bhandari et al. (2018); Srikant & Ying (2019); Chen et al. (2019; 2022); Mitra (2024).

● *Linear Speedup Effect.* From (12), notice that the radius of the ball of convergence around $\theta^\star$ is the sum of two terms: an $\mathcal{O}(\alpha\bar{\tau}\sigma^2/(MH))$ term that gets scaled down by the number of agents $M$, and a higher-order $\mathcal{O}\left(\alpha^2\sigma^2\right)$ term that can be made much smaller relative to the first term by making $\alpha$ sufficiently small, i.e., the dominant noise term exhibits a "variance-reduction" effect. To further highlight this effect, Corollary 1 reveals that with a decaying step-size, the sample-complexity of `FedHSA` is $\tilde{\mathcal{O}}(\sigma^2/(MHT))$; *this is essentially the best rate one can hope for* since after $H$ local steps in $T$ communication rounds, the total number of data samples collected across $M$ agents is precisely $MHT$. The $M$-fold reduction in sample-complexity makes it explicit that *even in a heterogeneous federated SA setting with time-correlated data, one can achieve linear speedups by collaborating using our proposed algorithm FedHSA.* This is the first result of its kind and significantly generalizes similar bounds for homogeneous FRL (Khodadadian et al., 2022; Woo et al., 2023).

● *Communication Complexity.* First, it should be clear that the notion of communication complexity makes no sense unless we associate it with a desired performance. For instance, our algorithm can involve zero

communication, but then the performance guarantees would be vacuous. Keeping this in mind, recall that for the single-agent centralized setting, given $R$ samples, the standard SA scheme achieves a mean-square error (MSE) on the order of $\tilde{\mathcal{O}}(1/R)$. Now, say each of the $M$ agents in our setup has access to precisely $R$ samples. A natural question then is: *How much communication is needed to achieve an MSE rate on the order of $\tilde{\mathcal{O}}(1/(MR))$, exhibiting the desired linear speedup effect?* We answer this question below.

In our notation, recall that $T$ is the number of communication rounds, and $H$ is the gap between communication rounds. As such, we have $R = TH$. Now note that Theorem 2 of our paper presents the following result:

$$d_T \leq \underbrace{\exp\left(-\frac{\mu}{4}\alpha T\right)\mathcal{O}(d_0 + \sigma^2)}_{(*)} + \mathcal{O}\left(\underbrace{\frac{\bar{\tau}\alpha L^2}{\mu M H(1-\rho)}}_{(**)} + \underbrace{\frac{\alpha^2 L^4}{\mu^2}}_{(***)}\right)\sigma^2.$$

This is essentially saying that the MSE after $T$ communication rounds is upper-bounded by an exponentially decaying term $(*)$ plus a term $(**)$ scaled down by the number of agents $M$, and another term $(***)$ that is higher-order in $\alpha$, but not scaled down by $M$. In order to achieve linear speedup, it is then straightforward that the term $(***)$ should be dominated by $(**)$, imposing the following requirement on the maximum allowable gap $H$ between communication rounds (to preserve the linear speedup effect):

$$H \leq \mu\bar{\tau}/(\alpha L^2 M(1-\rho)).$$

Substituting the choice of $\alpha = H\alpha_g\eta = 4\log(MHT)/(\mu T)$ in Corollary 1, we then have

$$H \leq \frac{\mu^2\bar{\tau}T}{4L^2 M(1-\rho)\log(MHT)}.$$

Finally, using $T = R/H$ in the above display yields the requirement:

$$H \leq \frac{\mu}{2L}\sqrt{\frac{\bar{\tau}R}{M(1-\rho)\log(MR)}}, \tag{14}$$

which is essentially $\tilde{\mathcal{O}}(\sqrt{R/M})$. Concretely, given $M$ and $R$, one can then set $H$ to be

$$\boxed{H = \left\lfloor \frac{\mu}{2L}\sqrt{\frac{\bar{\tau}R}{M(1-\rho)\log(MR)}} \right\rfloor.}$$

With $H$ as above, the communication overhead is $T = R/H$, which is on the order of $\sqrt{MR}$. A key message conveyed by the above discussion is as follows: to achieve the desired linear speedup, the gap $H$ between communication rounds cannot be arbitrarily large; instead, it needs to satisfy the constraint imposed in (14).

• *No Heterogeneity Bias.* In Proposition 1, we saw that if one employs the algorithms in Jin et al. (2022); Wang et al. (2024b); Zhang et al. (2024) in the heterogeneous setting, then there is a heterogeneity-induced bias term in the final bound that can potentially negate the benefits of collaboration. FedHSA effectively eliminates such a bias term *without making any assumptions whatsoever on the level of heterogeneity.* To see this, it suffices to note from (12) that in the noiseless case when $\sigma^2 = 0$, FedHSA guarantees exponentially fast convergence to $\theta^\star$, as opposed to a ball of radius $\mathcal{O}(\eta)$ around $\theta^\star$ like in Proposition 1.

**Remark on Constant/Diminishing Step-sizes.** As one might expect, a constant step-size (adopted in this paper) admits a much cleaner analysis than a diminishing step-size, which is exactly why we adopt it, to avoid distraction from the main message that we are trying to convey, i.e., we are able to fill the gap of achieving linear speedup with no heterogeneity bias under Markovian data.

Next, we note that non-diminishing constant step-sizes are used readily in the literature on SA/RL even in simpler single-agent settings, precisely with the same motivation (as us) of providing a simple, clean analysis;

see, for instance, Bhandari et al. (2018); Srikant & Ying (2019). In fact, note that in Theorem 2 (part (a)) of Bhandari et al. (2018), the constant step-size requires knowledge of the horizon $R = HT$, much like us. Other than picking up some additional log factors, this causes no degradation in the overall learning rate.

Having explained why we chose constant step-sizes, and the fact that such a choice is common in SA/RL, we now mention two pathways for getting rid of the knowledge of $R$. One direct option is to use a diminishing step-size schedule. For this setting, we can borrow well understood proof techniques: for instance, the one used to prove item (c) in Theorem 2 of Bhandari et al. (2018). The other—perhaps easier to adapt technique—is the "doubling trick" in the online learning literature. The basic idea is as follows. We first set the horizon length to say $R'$, decide the step-size as per $R'$ (like in our paper), and run the algorithm for $R' - 1$ iterations. Then, we set $R' \leftarrow 2R'$, reset the step-size, and repeat. This degrades the performance of the algorithm by at most a constant factor. To see why, note that due to doubling, the $k$-th epoch is of duration $2^k$, if $R' = 1$. Suppose the true unknown horizon length is $R$. Let $J$ be the last epoch before the horizon length is exceeded, i.e., $2^{J+1} > R$, implying $2^J > R/2$. But since the duration of the last epoch is precisely $2^J$, we conclude that the last epoch *contains at least half the total number of samples $R$.* Consequently, long story short, the price of not knowing $R$ a priori is a degradation by a constant factor that is no worse than 2.

To corroborate our developed theory, we provide various simulations in Appendix D.

## 5 Challenges and Proof Sketch

Although the analysis of Markovian sampling has appeared in prior SA/RL work, arriving at our main result in Theorem 2 is highly non-trivial, and requires overcoming several technical challenges arising from the interplay between complex statistical correlations, drift effects due to heterogeneity, and multiple local update steps. In what follows, we elaborate on these **challenges** by explaining why existing proof techniques are inadequate for our specific setting.

• *Comparison with homogeneous federated RL/SA papers.* The first paper to analyze the effects of Markov sampling in FRL was Khodadadian et al. (2022), followed by Woo et al. (2023). In Khodadadian et al. (2022), the authors consider a homogeneous setting, where all agents' operators have the same root, and as a result, a standard `FedAvg`-style local update rule suffices. Even so, the analysis in this paper is quite involved and proceeds by using the concept of Generalized Moreau Envelopes. It is not at all apparent whether such a proof technique could be extended to handle the bias-corrected local update scheme in our paper, subject to additional drift effects introduced by heterogeneity in agents' operators. Our proof, in contrast, does not require going through the framework of Generalized Moreau Envelopes, and analyzes a different local update scheme altogether (relative to that in Khodadadian et al. (2022) and Woo et al. (2023)).

While the analysis in Woo et al. (2023) sharpens some of the bounds in Khodadadian et al. (2022), the results are derived only for tabular Q-learning *without any function approximation.* Even in a single-agent setting, it is well understood that an extension from the tabular setting to the function-approximation setting is highly non-trivial, and several papers have focused precisely on this extension. In particular, for the tabular setting, a relatively straightforward argument (requiring no more than a couple of lines) suffices to establish that the Q-table iterates remain uniformly bounded throughout the course of the algorithm; see, for instance, Guannan Qu (2020). This fact is leveraged throughout the analysis. Unfortunately, under function approximation, it is no longer the case (in general) that the iterates are uniformly bounded deterministically (with a non-vacuous upper bound). This complicates the analysis significantly since each of the terms in the R.H.S. of our main recursion in Lemma 4 are *iterate-dependent* and cannot just be replaced by uniformly bounded perturbations (as is possible in a tabular setting).

• *Comparison with heterogeneous federated RL/SA papers.* The only two papers we know of that consider finite-time analysis of heterogeneous FRL algorithms under function approximation are Zhang et al. (2024) and Wang et al. (2024b). Our analysis departs from both these papers in two main ways. First, as we explained earlier in our paper, Zhang et al. (2024) and Wang et al. (2024b) study `FedAvg`-style algorithms where each agent uses just its own local operator to update model parameters between communication rounds. As we show in Proposition 1, such algorithms converge to incorrect fixed points. In contrast, we study a

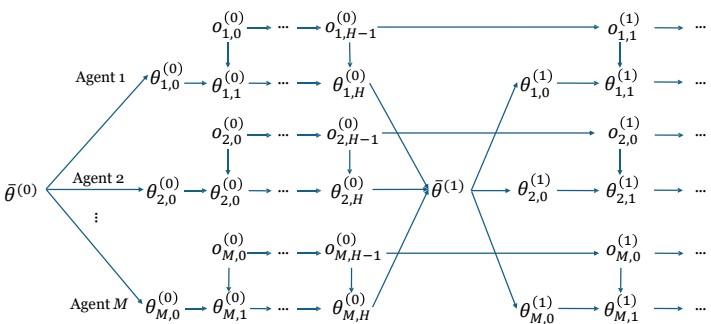

Figure 1: Illustration of the different statistical correlations in our heterogeneous federated SA setting.

different algorithm altogether, the dynamics of which are not the same as the ones in Zhang et al. (2024) and Wang et al. (2024b).

On a more technical note, both Zhang et al. (2024) and Wang et al. (2024b) consider a projection step in their algorithm to ensure that the iterates remain uniformly bounded. This considerably simplifies the analysis since the Markovian bias term is *iterate-dependent*, and projection ensures that such a term is essentially a uniformly bounded perturbation. In sharp contrast, we make no such assumption of a projection step, and, as such, we do not have the luxury of a uniformly bounded Markovian bias. Thus, a much finer analysis relative to Zhang et al. (2024) and Wang et al. (2024b) is needed to control each of the iterate-dependent "error" terms on the R.H.S. of our main one-step recursion in Lemma 4 (established in Appendix C). In addition, our proof also requires establishing that no heterogeneity-induced bias shows up in our final bound (unlike Zhang et al. (2024) and Wang et al. (2024b)).

• *Comparison with single-agent RL/SA papers.* Although Markovian sampling has been studied in single-agent RL/SA, there are considerable challenges in extending such results to our specific setting. As illustrated in Figure 1, our analysis has to account for two types of data correlations: (i) temporal correlations in the data for any given agent, and (ii) correlations induced by exchanging data across agents. For each agent $i$, the observation $o_{i,\ell}^{(t)}$ is statistically dependent on all prior observations for itself since they are all part of a *single Markovian trajectory.* Such an issue does not arise in the standard FL setting where one assumes i.i.d. data over time. Additionally, the local parameter $\theta_{i,\ell}^{(t)}$ is jointly influenced by all the parameters of all agents up to the beginning of round $t$, due to communication. Finally, all such parameters inherit randomness from prior Markovian observations. In short, *combining information generated by heterogeneous Markov chains creates complex spatial and temporal correlations.*

Moreover, relative to single-agent RL papers, we need to establish a finer result involving the linear speedup effect. To achieve this, we need to show that the Markovian bias term can be upper-bounded by terms that are either higher-order in the step-size or scaled down by the number of agents. **This is the hardest part of our analysis, and to our knowledge, no other paper in FRL has been able to establish such a result without making simplifying assumptions of projection steps.**

• *Comparison with Mangold et al. (2024).* In our discussion on more related work in Section 1, we have already explained why the nonlinearity of operators in our setting precludes the approach pursued in the work of Mangold et al. (2024). There is one other crucial difference with this work. In Mangold et al. (2024), the bias-corrected algorithm used for linear SA is studied only under i.i.d. sampling; **thus, our work is the first to analyze the effects of control variates under Markov sampling.** For vanilla SA, the approach in Mangold et al. (2024) to control Markov sampling is the "blocking technique", where one simply discards several data points by sub-sampling. By a coupling argument, the analysis then boils down to that for i.i.d. sampling. In sharp contrast, our approach does not discard any data points, making the Markovian analysis significantly more challenging since the samples used to generate iterates are temporally correlated.

**Summary.** To sum up, our discussion above highlights that **no prior analysis is enough to subsume the specific challenges that arise in our setting** from a combination of function approximation, nonlinear operators with heterogeneous roots, the lack of projection steps, and the need to establish a linear speedup

effect despite complex correlations. As such, we believe that the analysis framework developed in our paper can serve as a template for reasoning about more involved complex SA/RL schemes in the future.

**Note.** For notational simplicity, we omit the observation variable in the operator in subsequent analysis when the time-index of the observation matches with that of the parameter, i.e., we use the shorthand $G_i(\theta_{i,\ell}^{(t)}, o_{i,\ell}^{(t)}) := G_i(\theta_{i,\ell}^{(t)})$. The observation variable is made explicit only when its time-index differs from that of the associated parameter.

**Proof Sketch.** We now provide a high-level technical overview of our analysis; the details are deferred to Appendix C. Our first step is to establish a one-step recursion that captures the progress made by `FedHSA` in each communication round (Lemma 4 in Appendix C). Up to a higher order term in $\alpha$, the R.H.S. of this recursion comprises four terms: a "good" term that leads to a contraction in the mean-square error, a "noise variance" term that gets scaled down by $M$, a "drift" term due to heterogeneity, and a bias term due to Markov sampling. The challenging part of this result is showing the variance-reduction effect under Markov sampling; to do so, we carefully exploit the geometric mixing properties of the agents' Markov chains and Assumption 4.

Next, to control the drift effect due to heterogeneity, we show in Lemma 1 (in Appendix A) that if $\eta \leq 1/(LH)$, then the following is true *deterministically*:

$$\left\| \theta_{i,\ell}^{(t)} - \bar{\theta}^{(t)} \right\|_2^2 \leq \mathcal{O}(\eta^2 L^2 H^2) \left( \left\| \bar{\theta}^{(t)} - \theta^\star \right\|_2^2 + \sigma^2 \right).$$

To build intuition, notice that when there is no noise, i.e., $\sigma = 0$, if $\bar{\theta}^{(t)} = \theta^\star$, meaning the iterate at the start of the round is at the desired value, there would be no client-drift at all, precisely as desired. The most challenging part of our analysis pertains to controlling a Markovian bias term of the following form:

$$T_{bias} =: \mathbb{E}\left[ \left\langle \bar{\theta}^{(t)} - \theta^\star, \frac{2\alpha}{MH} \sum_{i=1}^{M} \sum_{\ell=0}^{H-1} \left( G_i(\theta_{i,\ell}^{(t)}) - \bar{G}_i(\theta_{i,\ell}^{(t)}) \right) \right\rangle \right],$$

that arises due to temporal correlations in data. Such a term vanishes in the standard FL setting where one assumes i.i.d. data. In the heterogeneous FRL setting in Wang et al. (2024b) and Zhang et al. (2024), such a term is simplified by assuming a projection step. Since we do not assume a projection step in `FedHSA`, we cannot benefit from such simplifications. Nonetheless, we establish the following key result.

**Claim (Informal). (Markovian Bias Control).** *Under the conditions of Theorem 2, we have:*

$$T_{bias} \leq \left( \frac{\alpha\mu}{2} + \mathcal{O}\left( \bar{\tau}L^2\alpha^2 + \frac{L^4\alpha^3}{\mu} \right) \right) \mathbb{E}\left[ \left\| \bar{\theta}^{(t)} - \theta^\star \right\|_2^2 \right] + \mathcal{O}\left( \frac{L^4\sigma^2\alpha^3}{\mu} + \frac{\bar{\tau}L^2\sigma^2\alpha^2}{MH(1-\rho)} \right).$$

A formal version of this claim appears as Lemma 6 in Appendix C. For the Markovian sampling result to closely resemble the i.i.d. case, we need to crucially ensure that (i) the iterate-dependent term in the bias can be dominated by the contractive "good" term, and (ii) the noise terms are either $\mathcal{O}(\alpha^3)$, or $\mathcal{O}(\alpha^2/M)$, i.e., the noise terms need to be either higher-order in $\alpha$ or exhibit an inverse scaling with $M$, to preserve the linear speedup effect. In the single-agent case (Bhandari et al., 2018; Srikant & Ying, 2019), one need not worry about linear speedups, and, as such, the corresponding analysis is much less involved. In summary, to arrive at our desired bounds, we need to significantly generalize known analysis techniques for both single- and multi-agent SA under Markov sampling.

## 6 Conclusion

We studied a general federated stochastic approximation problem subject to heterogeneity in the agents' local operators, and temporally correlated Markovian data at each agent. For this setting, we first showed that standard local SA algorithms may fail to converge to the right point. Based on this observation, we developed a new class of heterogeneity-aware federated SA algorithms that simultaneously (i) guarantee convergence to the right point while matching centralized rates, and (ii) enjoy linear speedups in sample-complexity that

are not degraded by the presence of an additive heterogeneity-induced bias term. Our results subsume the standard federated optimization setting and have implications for a broad class of SA-based RL algorithms. There are various interesting questions that we plan to explore in the future:

1. One natural question is to ask how much communication is necessarily needed to achieve optimal speedups in sample-complexity?

2. At the moment, our bounds scale with the mixing time of the slowest mixing Markov chain among the agents. Can this bound be further refined?

3. Can the techniques developed in this paper be employed for more complex SA-based RL algorithms with nonlinear function approximators? The extension to two-time-scale algorithms is also of interest.

4. In our analysis, we leveraged the "strong-monotonicity" condition in Assumption 2 to obtain contractive terms that led to overall progress towards the desired point in each iteration. However, in the context of say stochastic non-convex optimization, such an assumption may not hold. It is an interesting direction (even in the single-agent) case to explore the relaxation of Assumption 2 under Markov sampling, without making further simplifying assumptions such as a PL condition or projection steps or bounded gradients.

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

# A   Basic Results

In this section, we will compile some known facts and derive certain preliminary results that will play a key role in our subsequent analysis of `FedHSA`. Before we proceed, for the reader's convenience, we assemble all relevant notation in Table 2.

Table 2:   Notation

| Notation | Definition |
|---|---|
| $[N]$ | The set of $N$ numbers from 1 to $N$ |
| $\mathcal{M}_i$ | Markov chain of agent $i$ |
| $\mathcal{S}, \mathbb{R}^d$ | Common finite state space of agents' Markov chains, and $d$-dimensional real space |
| $s$ | Instance of state |
| $o_{i,t}, o_{i,\ell}^{(t)}$ | Observation of agent $i$ at time-step $t$, and observation of agent $i$ in local iteration $\ell$ of round $t$ |
| $\mu_i$ | stationary distribution of $\mathcal{M}_i$ |
| $d_{TV}$ | Total variation distance |
| $\bar{\theta}^{(t)}, \theta_{i,\ell}^{(t)}$ | Global parameter at round $t$ and local parameter of agent $i$ in local iteration $\ell$ of round $t$ |
| $G_i, \bar{G}_i$ | Noisy and true local operators of agent $i$ |
| $G, \bar{G}$ | Noisy and true global operators |
| $\theta^\star, \theta_i^\star$ | Root of global operator $\bar{G}(\theta)$, and root of local operator $\bar{G}_i(\theta)$ of agent $i$ |
| $\sigma_i, \sigma$ | Parameters capturing the effect of noise, with $\sigma := \max\{\sigma_1, \cdots, \sigma_M\}$ |
| $\rho_i, \rho$ | Parameters capturing the mixing-time property, with $\rho := \max\{\rho_1, \cdots, \rho_M\}$ |
| $L, \mu$ | Lipschitz constant and strong-monotonicity constant |
| $\eta, \alpha_g, \alpha$ | Local step-size, global step-size, and effective step-size |
| $\mathcal{O}, \tilde{\mathcal{O}}$ | Big-O notation hiding universal constants, and big-O notation hiding poly-logarithmic terms |
| $\tau, \bar{\tau}$ | Functions capturing the mixing time and the slowest mixing time defined as $\bar{\tau} := \tau(\alpha^2)$ |
| $\mathcal{F}_\ell^{(t)}$ | Filtration containing all randomness prior to round $t$ and local iteration $\ell$ across all agents |

- Given $m$ vectors $(x_1, \cdots, x_m) \in \mathbb{R}^d \times \cdots \times \mathbb{R}^d$, the following holds true by a simple application of Jensen's inequality:

$$\left\| \sum_{i=1}^m x_i \right\|_2^2 \leq m \sum_{i=1}^m \|x_i\|_2^2. \tag{15}$$

- Given $m$ vectors $(x_1, \cdots, x_m) \in \mathbb{R}^d \times \cdots \times \mathbb{R}^d$, the following is a generalization of the triangle inequality:

$$\left\| \sum_{i=1}^m x_i \right\|_2 \leq \sum_{i=1}^m \|x_i\|_2. \tag{16}$$

- Given any two vectors $(x, y) \in \mathbb{R}^d \times \mathbb{R}^d$, the following holds for any $\xi > 0$:

$$\langle x, \, y \rangle \leq \frac{\xi}{2} \|x\|_2^2 + \frac{1}{2\xi} \|y\|_2^2. \tag{17}$$

**Proof of Proposition 1** We first provide the proof of Proposition 1. Before we start, recall from Section 2 that the local update formula is

$$\theta_{i,\ell+1}^{(t)} = \theta_{i,\ell}^{(t)} + \eta \bar{G}_i(\theta_{i,\ell}^{(t)}), \ell = 0, 1, \ldots, H-1, \tag{18}$$

where $\bar{G}_i(\theta) = \bar{A}_i\theta - \bar{b}_i$ and $H = 2$. Also recall that the aggregation formula at the server is

$$\bar{\theta}^{(t+1)} = \frac{1}{M} \sum_{i \in [M]} \theta_{i,H}^{(t)}. \tag{19}$$

*Proof.* By definition of $\bar{G}_i$, we can write

$$
\begin{aligned}
\theta_{i,\ell+1}^{(t)} &= \theta_{i,\ell}^{(t)} + \eta \left( \bar{A}_i \theta_{i,\ell}^{(t)} - \bar{b}_i \right) \\
&= \left( I + \eta \bar{A}_i \right) \theta_{i,\ell}^{(t)} - \eta \bar{b}_i \\
&= \left( I + \eta \bar{A}_i \right) \left( \theta_{i,\ell}^{(t)} - \theta^\star \right) + \theta^\star + \eta \left( \bar{A}_i \theta^\star - \bar{b}_i \right),
\end{aligned} \tag{20}
$$

which yields

$$\theta_{i,\ell+1}^{(t)} - \theta^\star = \left( I + \eta \bar{A}_i \right) \left( \theta_{i,\ell}^{(t)} - \theta^\star \right) + \eta \left( \bar{A}_i \theta^\star - \bar{b}_i \right). \tag{21}$$

Iterating Eq. (21) for $H = 2$ steps, we obtain

$$\theta_{i,2}^{(t)} - \theta^\star = \left( I + \eta \bar{A}_i \right)^2 \left( \theta_{i,0}^{(t)} - \theta^\star \right) + \eta \left( I + \left( I + \eta \bar{A}_i \right) \right) \left( \bar{A}_i \theta^\star - \bar{b}_i \right). \tag{22}$$

Therefore, by Eq. (19) and the fact that $\theta_{i,0}^{(t)} = \bar{\theta}^{(t)}$ we have

$$\bar{\theta}^{(t+1)} - \theta^\star = \left( \frac{1}{M} \sum_{i=1}^{M} \left( I + \eta \bar{A}_i \right)^2 \right) \left( \bar{\theta}^{(t)} - \theta^\star \right) + \eta \left( \frac{1}{M} \sum_{i=1}^{M} \left( 2I + \eta \bar{A}_i \right) \left( \bar{A}_i \theta^\star - \bar{b}_i \right) \right). \tag{23}$$

Using the fact that $\bar{A}\theta^\star - \bar{b} = 0$ and $\bar{b}_i = \bar{A}_i \theta_i^\star$, we have

$$\frac{1}{M} \sum_{i=1}^{M} \left( 2I + \eta \bar{A}_i \right) \left( \bar{A}_i \theta^\star - \bar{b}_i \right) = \frac{\eta}{M} \sum_{i=1}^{M} \left( \bar{A}_i^2 \left( \theta^\star - \theta_i^\star \right) \right). \tag{24}$$

Also, we have

$$
\begin{aligned}
\frac{1}{M} \sum_{i=1}^{M} \left( I + \eta \bar{A}_i \right)^2 &= \frac{1}{M} \sum_{i=1}^{M} \left( I + 2\eta \bar{A}_i + \eta^2 \bar{A}_i^2 \right) \\
&= I + 2\eta \bar{A} + \eta^2 \bar{A}',
\end{aligned} \tag{25}
$$

where recall that $\bar{A} = (1/M) \sum_{i \in [M]} \bar{A}_i$ and $\bar{A}' = (1/M) \sum_{i \in [M]} \bar{A}_i^2$.

Now, defining $e_t := \bar{\theta}^{(t)} - \theta^\star$, and combining Eq. (23) - Eq. (25), we obtain

$$e_{t+1} = F e_t + \bar{v}, \quad \forall t \geq 0, \tag{26}$$

where $F = I + 2\eta \bar{A} + \eta^2 \bar{A}'$ and $\bar{v} = (\eta^2/M) \sum_{i=1}^{M} \left( \bar{A}_i^2 \left( \theta^\star - \theta_i^\star \right) \right)$. We then conclude that Eq. (26) represents a linear dynamical system in discrete time with state transition matrix $F$.

For stability, we need $F$ to be Schur-stable. Suppose $\eta$ is chosen to ensure that $F$ is Schur-stable. Now, applying Eq.(26) iteratively yields

$$e_t = F^t e_0 + \left( \sum_{k=0}^{t-1} F^k \right) \bar{v}. \tag{27}$$

Taking limits on both sides and using the fact that $F$ is Schur-stable, we obtain

$$\lim_{t \to \infty} F^t e_0 = 0, \quad \lim_{t \to \infty} \left( \sum_{k=0}^{t-1} F^k \right) = (I - F)^{-1} = \frac{-1}{\eta} \left( 2\bar{A} + \eta \bar{A}' \right)^{-1}. \tag{28}$$

Therefore,

$$\lim_{t \to \infty} e_t = \frac{\eta}{M} \left(2\bar{A} + \eta \bar{A}'\right)^{-1} \sum_{i=1}^{M} \left(\bar{A}_i^2 \left(\theta_i^\star - \theta^\star\right)\right). \tag{29}$$

We have thus proved that the error $e_t$ is non-vanishing. $\qquad\square$

The following corollary will come in handy at several points in our analysis.

**Corollary 2.** *Given Assumption 1, for any given $\theta, o$, the following holds for all $i \in [M]$:*

$$\left\|G_i(\theta, o) - \bar{G}_i(\theta)\right\|_2 \leq \mathcal{O}\left(L\right)\left(\|\theta - \theta^\star\|_2 + \sigma\right)$$
$$\|G_i(\theta, o)\|_2 \leq \mathcal{O}\left(L\right)\left(\|\theta - \theta^\star\|_2 + \sigma\right) \tag{30}$$
$$\left\|\bar{G}_i(\theta)\right\|_2 \leq \mathcal{O}\left(L\right)\left(\|\theta - \theta^\star\|_2 + \sigma\right)$$

*Proof.*

$$\begin{aligned}
\left\|G_i(\theta, o) - \bar{G}_i(\theta)\right\|_2 &\leq \|G_i(\theta, o)\|_2 + \left\|\bar{G}_i(\theta)\right\|_2 \\
&\leq 2L\left(\|\theta\|_2 + \sigma\right) \\
&= 2L\left(\|\theta - \theta^\star\|_2 + \|\theta^\star\|_2 + \sigma\right) \\
&\leq 2L\left(\|\theta - \theta^\star\|_2 + 2\sigma\right).
\end{aligned} \tag{31}$$

Here, the second inequality follows from (5), and the last inequality uses the definition of $\sigma$. The rest two bounds can be obtained similarly. $\qquad\square$

We also state upfront the following lemma that bounds the drift term $\left\|\theta_{i,\ell}^{(t)} - \bar{\theta}^{(t)}\right\|_2^2$. Notably, this result holds deterministically, i.e., it applies to both i.i.d. and Markovian sampling.

**Lemma 1.** *Suppose Assumptions 1 and 3 hold. By selecting $\eta \leq 1/(LH)$, the following is true for `FedHSA`:*

$$\left\|\theta_{i,\ell}^{(t)} - \bar{\theta}^{(t)}\right\|_2^2 \leq \mathcal{O}(\eta^2 L^2 H^2)\left(\left\|\bar{\theta}^{(t)} - \theta^\star\right\|_2^2 + \sigma^2\right). \tag{32}$$

**Proof of Lemma 1.**

*Proof.* A high-level intuition for proving this lemma is to use the update rule of `FedHSA` in tandem with Corollary 2 to obtain a recursion for the term $\left\|\theta_{i,\ell}^{(t)} - \bar{\theta}^{(t)}\right\|_2$. With that aim in mind, from (10) we can write

$$\begin{aligned}
\left\|\theta_{i,\ell+1}^{(t)} - \bar{\theta}^{(t)}\right\|_2 &= \left\|\theta_{i,\ell}^{(t)} - \bar{\theta}^{(t)} + \eta\left(G_i(\theta_{i,\ell}^{(t)}) + G(\bar{\theta}^{(t)}) - G_i(\bar{\theta}^{(t)})\right)\right\|_2 \\
&= \left\|\theta_{i,\ell}^{(t)} - \bar{\theta}^{(t)} + \eta\left(G_i(\theta_{i,\ell}^{(t)}, o_{i,\ell}^{(t)}) - G_i(\bar{\theta}^{(t)}, o_{i,\ell}^{(t)})\right) + \eta G(\bar{\theta}^{(t)}) + \eta G_i(\bar{\theta}^{(t)}, o_{i,\ell}^{(t)}) - \eta G_i(\bar{\theta}^{(t)}, o_{i,0}^{(t)})\right\|_2 \\
&\leq \left\|\theta_{i,\ell}^{(t)} - \bar{\theta}^{(t)}\right\|_2 + \left\|\eta\left(G_i(\theta_{i,\ell}^{(t)}, o_{i,\ell}^{(t)}) - G_i(\bar{\theta}^{(t)}, o_{i,\ell}^{(t)})\right)\right\|_2 + \left\|\eta G(\bar{\theta}^{(t)}) + \eta G_i(\bar{\theta}^{(t)}, o_{i,\ell}^{(t)}) - \eta G_i(\bar{\theta}^{(t)}, o_{i,0}^{(t)})\right\|_2 \\
&\leq \left\|\theta_{i,\ell}^{(t)} - \bar{\theta}^{(t)}\right\|_2 + \eta L\left\|\theta_{i,\ell}^{(t)} - \bar{\theta}^{(t)}\right\|_2 + \left\|\eta G(\bar{\theta}^{(t)})\right\|_2 + \left\|\eta G_i(\bar{\theta}^{(t)}, o_{i,\ell}^{(t)})\right\|_2 + \left\|\eta G_i(\bar{\theta}^{(t)}, o_{i,0}^{(t)})\right\|_2 \\
&\leq (1 + \eta L)\left\|\theta_{i,\ell}^{(t)} - \bar{\theta}^{(t)}\right\|_2 + \mathcal{O}(\eta L)\left(\left\|\bar{\theta}^{(t)} - \theta^\star\right\|_2 + \sigma\right),
\end{aligned} \tag{33}$$

where the in the second last inequality we used Assumption 1, and the last inequality follows from Corollary 2.

Applying (33) iteratively, we obtain

$$\begin{aligned}
\left\|\theta_{i,\ell}^{(t)} - \bar{\theta}^{(t)}\right\|_2 &\leq (1 + \eta L)^\ell \left\|\theta_{i,0}^{(t)} - \bar{\theta}^{(t)}\right\|_2 + \sum_{j=0}^{\ell-1}(1 + \eta L)^j \mathcal{O}(\eta L)\left(\left\|\bar{\theta}^{(t)} - \theta^\star\right\|_2 + \sigma\right) \\
&\leq H(1 + \eta L)^H \mathcal{O}(\eta L)\left(\left\|\bar{\theta}^{(t)} - \theta^\star\right\|_2 + \sigma\right) \\
&\leq \mathcal{O}(\eta LH)\left(\left\|\bar{\theta}^{(t)} - \theta^\star\right\|_2 + \sigma\right).
\end{aligned} \tag{34}$$

Here, the second inequality holds because $\theta_{i,0}^{(t)} = \bar{\theta}^{(t)}$ and $\ell \leq H$. The last one follows by choosing $\eta \leq 1/(LH)$, and thus $H(1 + \eta L)^H \mathcal{O}(\eta L) \leq (1 + 1/H)^H \mathcal{O}(\eta LH) \leq \mathcal{O}(\eta LH)$, where in the last inequality we used $(1 + 1/x)^x \leq e, \forall x > 0$. Squaring both sides of (34) yields the final form of Lemma 1. $\qquad \square$

Lemma 1 states that if we select $\alpha_g = 1$ and $\alpha = H\alpha_g\eta$, we can then bound the drift term $\left\| \theta_{i,\ell}^{(t)} - \bar{\theta}^{(t)} \right\|_2^2$ in the form of the squared norm of the distance of the global iterate to the global root plus the effect of noise, damped by the factor $\alpha^2$.

# B   Analysis of `FedHSA` under I.I.D. Sampling

We begin our analysis of the `FedHSA` algorithm's convergence by examining its performance in a simplified i.i.d. scenario. Specifically, we assume that for each agent $i \in [M]$, its observation $o_{i,t}$ at time step $t$ is drawn in an i.i.d. manner from its stationary distribution $\mu_i$. Our analysis of the i.i.d. setting will provide a foundational understanding of the finite-time behavior of `FedHSA`. In turn, it will offer a smoother transition to the more complex case of Markovian sampling.

For our subsequent analysis, we remind the reader again of the shorthand $G_i(\theta_{i,\ell}^{(t)}, o_{i,\ell}^{(t)}) := G_i(\theta_{i,\ell}^{(t)})$ used whenever the observation variable and the parameter share the same time-index.

We start by introducing the following lemma, which provides a recursion for the progress towards $\theta^\star$ made by `FedHSA` in each communication round.

**Lemma 2.** *Suppose Assumption 1 - 3 and 4 hold. Also, suppose the data samples of each agent $i \in [M]$ are drawn i.i.d. from the stationary distribution $\mu_i$ of its underlying Markov chain $\mathcal{M}_i$. Then, the following holds for `FedHSA` $\forall t \geq 0$:*

$$
\mathbb{E}\left[\left\|\bar{\theta}^{(t+1)} - \theta^\star\right\|_2^2\right] \leq \left(1 - \alpha\mu + \mathcal{O}\left(\alpha^2 L^2\right)\right) \mathbb{E}\left[\left\|\bar{\theta}^{(t)} - \theta^\star\right\|_2^2\right] + \mathcal{O}\left(\frac{\alpha^2 L^2 \sigma^2}{MH}\right)
$$
$$
+ \mathcal{O}\left(\frac{\alpha L^2}{\mu MH} + \frac{\alpha^2 L^2}{MH}\right) \sum_{i=1}^{M} \sum_{\ell=0}^{H-1} \mathbb{E}\left[\left\|\theta_{i,\ell}^{(t)} - \bar{\theta}^{(t)}\right\|_2^2\right]. \tag{35}
$$

*Proof.* From the update rule of `FedHSA` in Eq. (11) and the definition of $G(\bar{\theta}^{(t)})$, we obtain

$$
\bar{\theta}^{(t+1)} - \bar{\theta}^{(t)} = \frac{\alpha_g \eta}{M} \sum_{i=1}^{M} \sum_{\ell=0}^{H-1} \left(G_i(\theta_{i,\ell}^{(t)}) - G_i(\bar{\theta}^{(t)}, o_{i,0}^{(t)}) + G(\bar{\theta}^{(t)})\right)
$$
$$
= \frac{\alpha_g \eta}{M} \sum_{i=1}^{M} \sum_{\ell=0}^{H-1} G_i(\theta_{i,\ell}^{(t)}). \tag{36}
$$

Using the definition of the effective step-size $\alpha = H\eta\alpha_g$, we can then write the squared norm of the distance of the global iterate to the optimum $\theta^*$ as

$$
\left\|\bar{\theta}^{(t+1)} - \theta^\star\right\|_2^2 = \left\|\bar{\theta}^{(t)} - \theta^\star + \frac{\alpha}{MH} \sum_{i=1}^{M} \sum_{\ell=0}^{H-1} G_i(\theta_{i,\ell}^{(t)})\right\|_2^2
$$
$$
= \left\|\bar{\theta}^{(t)} - \theta^\star\right\|_2^2 + \underbrace{\left\langle \bar{\theta}^{(t)} - \theta^\star, \frac{2\alpha}{MH} \sum_{i=1}^{M} \sum_{\ell=0}^{H-1} G_i(\theta_{i,\ell}^{(t)})\right\rangle}_{T_1} + \underbrace{\left\|\frac{\alpha}{MH} \sum_{i=1}^{M} \sum_{\ell=0}^{H-1} G_i(\theta_{i,\ell}^{(t)})\right\|_2^2}_{T_2}. \tag{37}
$$

To bound this equation in expectation, it then boils down to bounding the expectations of the terms $T_1$ and $T_2$ separately. We proceed to do this next.

$$
\begin{aligned}
\mathbb{E}[T_1] &\overset{(a)}{=} 2\alpha \mathbb{E}\left[\left\langle \bar{\theta}^{(t)} - \theta^\star, \frac{1}{MH}\sum_{i=1}^{M}\sum_{\ell=0}^{H-1}\bar{G}_i(\theta_{i,\ell}^{(t)})\right\rangle\right] \\
&= 2\alpha\mathbb{E}\left[\left\langle\bar{\theta}^{(t)}-\theta^\star, \frac{1}{MH}\sum_{i=1}^{M}\sum_{\ell=0}^{H-1}\left(\bar{G}_i(\theta_{i,\ell}^{(t)})-\bar{G}_i(\bar{\theta}^{(t)})\right)\right\rangle\right] + 2\alpha\mathbb{E}\left[\left\langle\bar{\theta}^{(t)}-\theta^\star, \frac{1}{MH}\sum_{i=1}^{M}\sum_{\ell=0}^{H-1}\bar{G}_i(\bar{\theta}^{(t)})\right\rangle\right] \\
&= 2\alpha\mathbb{E}\left[\left\langle\bar{\theta}^{(t)}-\theta^\star, \frac{1}{MH}\sum_{i=1}^{M}\sum_{\ell=0}^{H-1}\left(\bar{G}_i(\theta_{i,\ell}^{(t)})-\bar{G}_i(\bar{\theta}^{(t)})\right)\right\rangle\right] + 2\alpha\mathbb{E}\left[\left\langle\bar{\theta}^{(t)}-\theta^\star, \bar{G}(\bar{\theta}^{(t)})\right\rangle\right] \\
&\overset{(b)}{\leq} \frac{2\alpha}{MH}\sum_{i=1}^{M}\sum_{\ell=0}^{H-1}\mathbb{E}\left[\left\langle\bar{\theta}^{(t)}-\theta^\star, \bar{G}_i(\theta_{i,\ell}^{(t)})-\bar{G}_i(\bar{\theta}^{(t)})\right\rangle\right] - 2\alpha\mu\mathbb{E}\left[\left\|\bar{\theta}^{(t)}-\theta^\star\right\|_2^2\right] \\
&\leq \frac{\alpha}{MH}\sum_{i=1}^{M}\sum_{\ell=0}^{H-1}\mathbb{E}\left[\mu\left\|\bar{\theta}^{(t)}-\theta^\star\right\|_2^2 + \frac{1}{\mu}\left\|\bar{G}_i(\theta_{i,\ell}^{(t)})-\bar{G}_i(\bar{\theta}^{(t)})\right\|_2^2\right] - 2\alpha\mu\mathbb{E}\left[\left\|\bar{\theta}^{(t)}-\theta^\star\right\|_2^2\right] \\
&\overset{(c)}{\leq} \frac{\alpha}{MH}\sum_{i=1}^{M}\sum_{\ell=0}^{H-1}\mathbb{E}\left[\mu\left\|\bar{\theta}^{(t)}-\theta^\star\right\|_2^2 + \frac{L^2}{\mu}\left\|\theta_{i,\ell}^{(t)}-\bar{\theta}^{(t)}\right\|_2^2\right] - 2\alpha\mu\mathbb{E}\left[\left\|\bar{\theta}^{(t)}-\theta^\star\right\|_2^2\right] \\
&= -\alpha\mu\mathbb{E}\left[\left\|\bar{\theta}^{(t)}-\theta^\star\right\|_2^2\right] + \frac{\alpha L^2}{\mu MH}\sum_{i=1}^{M}\sum_{\ell=0}^{H-1}\mathbb{E}\left[\left\|\theta_{i,\ell}^{(t)}-\bar{\theta}^{(t)}\right\|_2^2\right],
\end{aligned}
$$
(38)

where $(b)$ follows from Assumption 2; $(c)$ is a result of Assumption 1. Before we proceed, we define $\mathcal{F}_\ell^{(t)}$ as the $\sigma$-algebra capturing all the randomness up to the $\ell$-th local iteration of round $t$. Building on this definition, we reason about $(a)$ separately as follows:

$$
\begin{aligned}
\mathbb{E}[T_1] &= 2\alpha\mathbb{E}\left[\left\langle\bar{\theta}^{(t)}-\theta^\star, \frac{1}{MH}\sum_{i=1}^{M}\sum_{\ell=0}^{H-1}G_i(\theta_{i,\ell}^{(t)})\right\rangle\right] \\
&= 2\alpha\mathbb{E}\left[\left\langle\bar{\theta}^{(t)}-\theta^\star, \frac{1}{MH}\sum_{i=1}^{M}\sum_{\ell=0}^{H-2}G_i(\theta_{i,\ell}^{(t)})\right\rangle\right] + 2\alpha\mathbb{E}\left[\left\langle\bar{\theta}^{(t)}-\theta^\star, \frac{1}{MH}\sum_{i=1}^{M}G_i(\theta_{i,H-1}^{(t)})\right\rangle\right] \\
&\overset{(a1)}{=} 2\alpha\mathbb{E}\left[\left\langle\bar{\theta}^{(t)}-\theta^\star, \frac{1}{MH}\sum_{i=1}^{M}\sum_{\ell=0}^{H-2}G_i(\theta_{i,\ell}^{(t)})\right\rangle\right] + 2\alpha\mathbb{E}\left[\mathbb{E}\left[\left\langle\bar{\theta}^{(t)}-\theta^\star, \frac{1}{MH}\sum_{i=1}^{M}G_i(\theta_{i,H-1}^{(t)})\right\rangle\bigg|\mathcal{F}_{H-2}^{(t)}\right]\right] \\
&\overset{(a2)}{=} 2\alpha\mathbb{E}\left[\left\langle\bar{\theta}^{(t)}-\theta^\star, \frac{1}{MH}\sum_{i=1}^{M}\sum_{\ell=0}^{H-2}G_i(\theta_{i,\ell}^{(t)})\right\rangle\right] + 2\alpha\mathbb{E}\left[\left\langle\bar{\theta}^{(t)}-\theta^\star, \frac{1}{MH}\sum_{i=1}^{M}\mathbb{E}\left[G_i(\theta_{i,H-1}^{(t)})\big|\mathcal{F}_{H-2}^{(t)}\right]\right\rangle\right] \\
&\overset{(a3)}{=} 2\alpha\mathbb{E}\left[\left\langle\bar{\theta}^{(t)}-\theta^\star, \frac{1}{MH}\sum_{i=1}^{M}\sum_{\ell=0}^{H-2}G_i(\theta_{i,\ell}^{(t)})\right\rangle\right] + 2\alpha\mathbb{E}\left[\left\langle\bar{\theta}^{(t)}-\theta^\star, \frac{1}{MH}\sum_{i=1}^{M}\bar{G}_i(\theta_{i,H-1}^{(t)})\right\rangle\right].
\end{aligned}
$$
(39)

Here, $(a1)$ follows from the tower property of expectations; $(a2)$ holds from the fact that $\bar{\theta}^{(t)}$ is deterministic conditioned on $\mathcal{F}_{H-2}^{(t)}$, and $(a3)$ is a result of the i.i.d. assumption in tandem with the fact that $\theta_{i,H-1}^{(t)}$ is deterministic conditioned on $\mathcal{F}_{H-2}^{(t)}$. We can repeat this process by iteratively conditioning on $\mathcal{F}_{H-3}^{(t)}, ..., \mathcal{F}_{-1}^{(t)}$ and arrive at $(a)$, where $\mathcal{F}_{-1}^{(t)}$ is all the randomness up to round $t$ before any local steps are taken.

We now move on to bound the expectation of $T_2$.

$$
\begin{aligned}
\mathbb{E}[T_2] &= \alpha^2 \mathbb{E}\left[\left\|\frac{1}{MH}\sum_{i=1}^{M}\sum_{\ell=0}^{H-1}\left(G_i(\theta_{i,\ell}^{(t)}) - \bar{G}_i(\theta_{i,\ell}^{(t)})\right) + \frac{1}{MH}\sum_{i=1}^{M}\sum_{\ell=0}^{H-1}\bar{G}_i(\theta_{i,\ell}^{(t)})\right\|_2^2\right] \\
&\leq 2\alpha^2\mathbb{E}\left[\left\|\frac{1}{MH}\sum_{i=1}^{M}\sum_{\ell=0}^{H-1}\left(G_i(\theta_{i,\ell}^{(t)}) - \bar{G}_i(\theta_{i,\ell}^{(t)})\right)\right\|_2^2\right] + 2\alpha^2\mathbb{E}\left[\left\|\frac{1}{MH}\sum_{i=1}^{M}\sum_{\ell=0}^{H-1}\bar{G}_i(\theta_{i,\ell}^{(t)})\right\|_2^2\right] \\
&\overset{(a)}{\leq} 2\alpha^2\mathbb{E}\left[\frac{1}{M^2H^2}\sum_{i=1}^{M}\sum_{\ell=0}^{H-1}\left\|G_i(\theta_{i,\ell}^{(t)}) - \bar{G}_i(\theta_{i,\ell}^{(t)})\right\|_2^2\right] \\
&\quad + 2\alpha^2\mathbb{E}\left[\left\|\frac{1}{MH}\sum_{i=1}^{M}\sum_{\ell=0}^{H-1}\left(\bar{G}_i(\theta_{i,\ell}^{(t)}) - \bar{G}_i(\bar{\theta}^{(t)})\right) + \frac{1}{MH}\sum_{i=1}^{M}\sum_{\ell=0}^{H-1}\bar{G}_i(\bar{\theta}^{(t)})\right\|_2^2\right] \\
&\overset{(b)}{\leq} 2\alpha^2\mathbb{E}\left[\frac{1}{M^2H^2}\sum_{i=1}^{M}\sum_{\ell=0}^{H-1}\mathcal{O}\left(L^2\right)\left(\left\|\theta_{i,\ell}^{(t)} - \theta^\star\right\|_2^2 + \sigma^2\right)\right] + 4\alpha^2\mathbb{E}\left[\left\|\frac{1}{MH}\sum_{i=1}^{M}\sum_{\ell=0}^{H-1}\left(\bar{G}_i(\theta_{i,\ell}^{(t)}) - \bar{G}_i(\bar{\theta}^{(t)})\right)\right\|_2^2\right] \\
&\quad + 4\alpha^2\mathbb{E}\left[\left\|\frac{1}{MH}\sum_{i=1}^{M}\sum_{\ell=0}^{H-1}\bar{G}_i(\bar{\theta}^{(t)})\right\|_2^2\right] \\
&\leq 2\alpha^2\mathbb{E}\left[\frac{1}{M^2H^2}\sum_{i=1}^{M}\sum_{\ell=0}^{H-1}\mathcal{O}(L^2)\left(\left\|\theta_{i,\ell}^{(t)} - \bar{\theta}^{(t)}\right\|_2^2 + \left\|\bar{\theta}^{(t)} - \theta^\star\right\|_2^2 + \sigma^2\right)\right] \\
&\quad + \frac{4\alpha^2}{MH}\sum_{i=1}^{M}\sum_{\ell=0}^{H-1}\mathbb{E}\left[\left\|\bar{G}_i(\theta_{i,\ell}^{(t)}) - \bar{G}_i(\bar{\theta}^{(t)})\right\|_2^2\right] + 4\alpha^2\mathbb{E}\left[\left\|\bar{G}(\bar{\theta}^{(t)})\right\|_2^2\right] \\
&\overset{(c)}{\leq} \frac{2\alpha^2}{M^2H^2}\sum_{i=1}^{M}\sum_{\ell=0}^{H-1}\mathcal{O}(L^2)\mathbb{E}\left[\left(\left\|\theta_{i,\ell}^{(t)} - \bar{\theta}^{(t)}\right\|_2^2 + \left\|\bar{\theta}^{(t)} - \theta^\star\right\|_2^2 + \sigma^2\right)\right] + \frac{4\alpha^2 L^2}{MH}\sum_{i=1}^{M}\sum_{\ell=0}^{H-1}\mathbb{E}\left[\left\|\theta_{i,\ell}^{(t)} - \bar{\theta}^{(t)}\right\|_2^2\right] \\
&\quad + 4\alpha^2 L^2\mathbb{E}\left[\left\|\bar{\theta}^{(t)} - \theta^\star\right\|_2^2\right] \\
&\leq \mathcal{O}\left(\frac{\alpha^2 L^2}{M^2H^2} + \frac{\alpha^2 L^2}{MH}\right)\sum_{i=1}^{M}\sum_{\ell=0}^{H-1}\mathbb{E}\left[\left\|\theta_{i,\ell}^{(t)} - \bar{\theta}^{(t)}\right\|_2^2\right] + \mathcal{O}\left(\alpha^2 L^2 + \frac{\alpha^2 L^2}{MH}\right)\mathbb{E}\left[\left\|\bar{\theta}^{(t)} - \theta^\star\right\|_2^2\right] + \mathcal{O}\left(\frac{\alpha^2 L^2\sigma^2}{MH}\right) \\
&\leq \mathcal{O}\left(\frac{\alpha^2 L^2}{MH}\right)\sum_{i=1}^{M}\sum_{\ell=0}^{H-1}\mathbb{E}\left[\left\|\theta_{i,\ell}^{(t)} - \bar{\theta}^{(t)}\right\|_2^2\right] + \mathcal{O}\left(\alpha^2 L^2\right)\mathbb{E}\left[\left\|\bar{\theta}^{(t)} - \theta^\star\right\|_2^2\right] + \mathcal{O}\left(\frac{\alpha^2 L^2\sigma^2}{MH}\right),
\end{aligned}
$$
$$(40)$$

where $(b)$ uses Corollary 2, and $(c)$ is a result of Assumption 1. We provide the rationale behind $(a)$ as follows.

The second term in inequality $(a)$ is a direct result of add-and-subtract. To inspect the first term, note that

$$
\mathbb{E}\left[\left\|\sum_{i=1}^{M}\sum_{\ell=0}^{H-1}\left(G_i(\theta_{i,\ell}^{(t)}) - \bar{G}_i(\theta_{i,\ell}^{(t)})\right)\right\|_2^2\right]
$$

$$
= \mathbb{E}\left[\sum_{i=1}^{M}\sum_{\ell=0}^{H-1}\left\|G_i(\theta_{i,\ell}^{(t)}) - \bar{G}_i(\theta_{i,\ell}^{(t)})\right\|_2^2\right] + 2\sum_{j=1}^{M}\sum_{m<n}\mathbb{E}\left[\left\langle G_j(\theta_{j,m}^{(t)}) - \bar{G}_j(\theta_{j,m}^{(t)}), G_j(\theta_{j,n}^{(t)}) - \bar{G}_j(\theta_{j,n}^{(t)})\right\rangle\right]
$$

$$
+ 2\sum_{j<k}\sum_{m=0}^{H-1}\mathbb{E}\left[\left\langle G_j(\theta_{j,m}^{(t)}) - \bar{G}_j(\theta_{j,m}^{(t)}), G_k(\theta_{k,m}^{(t)}) - \bar{G}_k(\theta_{k,m}^{(t)})\right\rangle\right]
$$

$$
+ 2\sum_{j<k}\sum_{m<n}\mathbb{E}\left[\left\langle G_j(\theta_{j,m}^{(t)}) - \bar{G}_j(\theta_{j,m}^{(t)}), G_k(\theta_{k,n}^{(t)}) - \bar{G}_k(\theta_{k,n}^{(t)})\right\rangle\right]
$$

$$
= \mathbb{E}\left[\sum_{i=1}^{M}\sum_{\ell=0}^{H-1}\left\|G_i(\theta_{i,\ell}^{(t)}) - \bar{G}_i(\theta_{i,\ell}^{(t)})\right\|_2^2\right] + 2\sum_{j=1}^{M}\sum_{m<n}\mathbb{E}\left[\mathbb{E}\left[\left\langle G_j(\theta_{j,m}^{(t)}) - \bar{G}_j(\theta_{j,m}^{(t)}), G_j(\theta_{j,n}^{(t)}) - \bar{G}_j(\theta_{j,n}^{(t)})\right\rangle\Big|\mathcal{F}_{n-1}^{(t)}\right]\right]
$$

$$
+ 2\sum_{j<k}\sum_{m=0}^{H-1}\mathbb{E}\left[\mathbb{E}\left[\left\langle G_j(\theta_{j,m}^{(t)}) - \bar{G}_j(\theta_{j,m}^{(t)}), G_k(\theta_{k,m}^{(t)}) - \bar{G}_k(\theta_{k,m}^{(t)})\right\rangle\Big|\mathcal{F}_{m-1}^{(t)}\right]\right]
$$

$$
+ 2\sum_{j<k}\sum_{m<n}\mathbb{E}\left[\mathbb{E}\left[\left\langle G_j(\theta_{j,m}^{(t)}) - \bar{G}_j(\theta_{j,m}^{(t)}), G_k(\theta_{k,n}^{(t)}) - \bar{G}_k(\theta_{k,n}^{(t)})\right\rangle\Big|\mathcal{F}_{n-1}^{(t)}\right]\right]
$$

$$
= \mathbb{E}\left[\sum_{i=1}^{M}\sum_{\ell=0}^{H-1}\left\|G_i(\theta_{i,\ell}^{(t)}) - \bar{G}_i(\theta_{i,\ell}^{(t)})\right\|_2^2\right],
$$

$$\tag{41}$$

where the first equality holds from unrolling the squared norm, and the second equality follows from the tower property of expectation. We analyze the terms separately in the last equality.

For the first cross-term, we can write

$$
2\sum_{j=1}^{M}\sum_{m<n}\mathbb{E}\left[\mathbb{E}\left[\left\langle G_j(\theta_{j,m}^{(t)}) - \bar{G}_j(\theta_{j,m}^{(t)}), G_j(\theta_{j,n}^{(t)}) - \bar{G}_j(\theta_{j,n}^{(t)})\right\rangle\Big|\mathcal{F}_{n-1}^{(t)}\right]\right]
$$

$$
= 2\sum_{j=1}^{M}\sum_{m<n}\mathbb{E}\left[\left\langle G_j(\theta_{j,m}^{(t)}) - \bar{G}_j(\theta_{j,m}^{(t)}), \mathbb{E}\left[G_j(\theta_{j,n}^{(t)}) - \bar{G}_j(\theta_{j,n}^{(t)})\Big|\mathcal{F}_{n-1}^{(t)}\right]\right\rangle\right]
$$

$$
= 0.
$$

$$\tag{42}$$

The first inequality holds from the definition of $\mathcal{F}_{n-1}^{(t)}$. The second one holds because $\theta_{j,n}^{(t)}$ is deterministic conditioned on $\mathcal{F}_{n-1}^{(t)}$, and the only randomness comes from $o_{j,n}^{(t)}$ in $G_j(\theta_{j,n}^{(t)})$. Due to the i.i.d. sampling assumption, we readily know that $\mathbb{E}\left[G_j(\theta_{j,n}^{(t)})\right] = \bar{G}_j(\theta_{j,n}^{(t)})$ for a fixed $\theta_{j,n}^{(t)}$, and the cross-term then equals 0.

For the second cross-term, we can write

$$
2\sum_{j<k}\sum_{m=0}^{H-1}\mathbb{E}\left[\mathbb{E}\left[\left\langle G_j(\theta_{j,m}^{(t)}) - \bar{G}_j(\theta_{j,m}^{(t)}), G_k(\theta_{k,m}^{(t)}) - \bar{G}_k(\theta_{k,m}^{(t)})\right\rangle\Big|\mathcal{F}_{m-1}^{(t)}\right]\right]
$$

$$
\overset{(i)}{=} 2\sum_{j<k}\sum_{m=0}^{H-1}\mathbb{E}\left[\left\langle \mathbb{E}\left[e_{j,m}^{(t)}\big|\mathcal{F}_{m-1}^{(t)}\right], \mathbb{E}\left[e_{k,m}^{(t)}\big|\mathcal{F}_{m-1}^{(t)]}\right]\right\rangle\right]
$$

$$
\overset{(ii)}{=} 0,
$$

$$\tag{43}$$

where we define $e_{i,\ell}^{(t)} := G_i(\theta_{i,\ell}^{(t)}) - \bar{G}_i(\theta_{i,\ell}^{(t)})$. For $(i)$, we used the fact that $\theta_{j,m}^{(t)}$ and $\theta_{k,m}^{(t)}$ are deterministic conditioned on $\mathcal{F}_{m-1}^{(t)}$. So, due to Assumption 4, $e_{j,m}^{(t)}$ and $e_{k,m}^{(t)}$ are conditionally independent. For $(ii)$, we used the i.i.d. sampling assumption.

The third cross-term can be proved to be 0 similarly as for the second one.

Taking expectation on both sides of (37), and plugging in the bounds from (38) and (40), we obtain:

$$\mathbb{E}\left[\left\|\bar{\theta}^{(t+1)} - \theta^\star\right\|_2^2\right] \leq \left(1 - \alpha\mu + \mathcal{O}\left(\alpha^2 L^2\right)\right) \mathbb{E}\left[\left\|\bar{\theta}^{(t)} - \theta^\star\right\|_2^2\right]$$

$$\underbrace{+ \left(\frac{\alpha L^2}{\mu M H} + \frac{\alpha^2 L^2}{M H}\right) \sum_{i=1}^M \sum_{\ell=0}^{H-1} \mathbb{E}\left[\left\|\theta_{i,\ell}^{(t)} - \bar{\theta}^{(t)}\right\|_2^2\right]}_{DRIFT} + \mathcal{O}\left(\frac{\alpha^2 L^2 \sigma^2}{M H}\right). \tag{44}$$

The proof of Lemma 2 is then complete. $\qquad\square$

Lemma 2 breaks the bounding of the recursion $\mathbb{E}\left[\left\|\bar{\theta}^{(t+1)} - \theta^\star\right\|_2^2\right]$ into three parts: (i) the "good term" $\left(1 - \alpha\mu + \mathcal{O}\left(\alpha^2 L^2\right)\right) \mathbb{E}\left[\left\|\bar{\theta}^{(t)} - \theta^\star\right\|_2^2\right]$, where we can select $\alpha$ small enough such that $\mathcal{O}\left(\alpha^2 L^2\right)$ can be absorbed by the negative term $-\alpha\mu$, ensuring that the distance to the optimum $\theta^\star$ decreases; (ii) the effect of noise $\mathcal{O}\left(\alpha^2 L^2 \sigma^2 / (M H)\right)$ which demonstrates the benefit of collaboration (since it gets scaled down by $M$), and (iii) the drift term $DRIFT$ that accounts for the client drift caused by environmental heterogeneity and local steps. We can further bound this term by plugging in Lemma 1, and arrive at the following result of Theorem 1.

**Proof of Theorem 1.**

*Proof.* Plugging the bound from Lemma 1 into the bound from Lemma 2, we obtain:

$$\mathbb{E}\left[\left\|\bar{\theta}^{(t+1)} - \theta^\star\right\|_2^2\right] \leq \left(1 - \alpha\mu + \mathcal{O}\left(\alpha^2 L^2\right)\right) \mathbb{E}\left[\left\|\bar{\theta}^{(t)} - \theta^\star\right\|_2^2\right] + \mathcal{O}\left(\frac{\alpha^2 L^2 \sigma^2}{M H}\right)$$

$$+ \mathcal{O}\left(\frac{\alpha L^2}{\mu M H} + \frac{\alpha^2 L^2}{M H}\right) M H \alpha^2 L^2 \left(\mathbb{E}\left[\left\|\bar{\theta}^{(t)} - \theta^\star\right\|_2^2\right] + \sigma^2\right)$$

$$\leq \left(1 - \alpha\mu + \mathcal{O}\left(\alpha^2 L^2\right) + \mathcal{O}\left(\alpha^4 L^4\right) + \mathcal{O}\left(\frac{\alpha^3 L^4}{\mu}\right)\right) \mathbb{E}\left[\left\|\bar{\theta}^{(t)} - \theta^\star\right\|_2^2\right] \tag{45}$$

$$+ \mathcal{O}\left(\frac{\alpha^2 L^2 \sigma^2}{M H} + \frac{\alpha^3 L^4 \sigma^2}{\mu} + \alpha^4 L^4 \sigma^2\right)$$

$$\leq \left(1 - \alpha\mu + \mathcal{O}\left(\alpha^2 L^2\right)\right) \mathbb{E}\left[\left\|\bar{\theta}^{(t)} - \theta^\star\right\|_2^2\right] + \mathcal{O}\left(\frac{\alpha^2 L^2}{M H} + \frac{\alpha^3 L^4}{\mu}\right)\sigma^2.$$

Here, the first inequality follows from the definition of $\alpha = H\eta\alpha_g$. The last one is a result of the fact that $\alpha \leq \mu/L^2$, $L \geq 1$, $\mu \leq 1$, such that $\alpha^4 L^4 \leq \alpha^2 L^2$, $\alpha^3 L^4/\mu \leq \alpha^2 L^2$, and $\alpha^4 L^4 \sigma^2 \leq \alpha^3 L^4 \sigma^2/\mu$.

Applying Eq. (45) iteratively $T$ times yields

$$\mathbb{E}\left[\left\|\bar{\theta}^{(T)} - \theta^\star\right\|_2^2\right] \leq \left(1 - \frac{\alpha\mu}{2}\right)^T \left\|\bar{\theta}^{(0)} - \theta^\star\right\|_2^2 + \mathcal{O}\left(\frac{\alpha^2 L^2}{M H} + \frac{\alpha^3 L^4}{\mu}\right)\sigma^2 \sum_{t=0}^{T-1}\left(1 - \frac{\alpha\mu}{2}\right)^t$$

$$\leq \exp\left(-\frac{\alpha\mu}{2}T\right)\left\|\bar{\theta}^{(0)} - \theta^\star\right\|_2^2 + \mathcal{O}\left(\frac{\alpha^2 L^2}{M H} + \frac{\alpha^3 L^4}{\mu}\right)\frac{2\sigma^2}{\alpha\mu} \tag{46}$$

$$\leq \exp\left(-\frac{\mu}{2}\alpha T\right)\left\|\bar{\theta}^{(0)} - \theta^\star\right\|_2^2 + \mathcal{O}\left(\frac{\alpha L^2}{\mu M H} + \frac{\alpha^2 L^4}{\mu^2}\right)\sigma^2.$$

Here, the first inequality holds since $\alpha \leq \mu/(2CL^2)$ where $C$ is the dominant constant in $\mathcal{O}\left(\alpha^2 L^2\right)$, and thus $C\alpha^2 L^2 \leq \alpha\mu/2$. The second inequality holds from the fact that $1 - x \leq \exp\left(-x\right)$. The proof is then complete. $\qquad\square$

Thus far, we have demonstrated that by selecting $\alpha$ small enough so that the term $\alpha^2 L^4/\mu^2$ is subsumed by $\alpha L^2/(\mu M H)$, Theorem 1 mirrors the result of the single-agent case as seen in Eq. (3) under i.i.d.

sampling. Additionally, it is important to highlight that a linear speedup is achieved, i.e., the variance term, $\alpha L^2 \sigma^2/(\mu M H)$, is reduced in proportion to the number of agents $M$, further underscoring the benefits of collaboration in a multi-agent setting.

## C Analysis of `FedHSA` under Markovian Sampling

Now, we shift our focus to the central result of our analysis: the convergence of the `FedHSA` algorithm under Markovian sampling. In this scenario, the observations $\{o_{i,t}\}_{t\geq 1}$ for each agent $i \in [M]$ are temporally correlated, as opposed to being statistically independent. This introduces additional complexity to the analysis, as we must account for the intricate interactions between observations across different time steps and among various agents.

Building upon the geometric mixing time property, we present the following corollary to deal with time-correlation in observations.

**Corollary 3.** *For each agent $i \in [M]$, any given $\theta \in \mathbb{R}^d$, and any non-negative integers $\tau, k$ satisfying $\tau \leq k$, the following holds:*

$$\left\| \mathbb{E}\left[G_i(\theta, o_{i,k})|o_{i,k-\tau}\right] - \bar{G}_i(\theta) \right\|_2 \leq \mathcal{O}\left(L\rho^\tau\right)\left(\|\theta - \theta^\star\|_2 + \sigma\right), \tag{47}$$

*where $o_{i,k}$ is the observation made by agent $i$ at the $k$-th time-step.*

We refer the reader to Lemma 3.1 of (Chen et al., 2019) for a proof.

Corollary 3 states that given a fixed parameter $\theta$, the difference in the Euclidean norm $\|\cdot\|_2$ between the true operator and the expectation of the noisy operator, conditioned on the observation from $\tau$ time-steps before, decays exponentially fast, where the $\|\theta - \theta^\star\|_2$ term captures the influence of $\theta$. Corollary 3 is particularly useful for addressing the temporal correlation between observations. Specifically, when observations are sampled i.i.d. from the distribution $\mu_i$, the left-hand side of (47) equals zero, recovering the relationship between $G_i$ and $\bar{G}_i$.

Before delving into the details of the proofs, we first introduce the following lemma that bounds the variance reduction term in the Markovian sampling scenario.

**Lemma 3.** *Suppose Assumptions 1 to 4 hold. Then the following holds for `FedHSA` $\forall t \geq \bar{\tau}$, where $\bar{\tau} = \tau(\alpha^2)$.*

$$\overbrace{2\alpha^2 \mathbb{E}\left[\left\|\frac{1}{MH}\sum_{i=1}^{M}\sum_{\ell=0}^{H-1}\left(G_i(\theta_{i,\ell}^{(t)}) - \bar{G}_i(\theta_{i,\ell}^{(t)})\right)\right\|_2^2\right]}^{VR_{Markov}} \leq \mathcal{O}\left(\frac{L^2\alpha^2}{MH}\right)\sum_{i=1}^{M}\sum_{\ell=0}^{H-1}\mathbb{E}\left[\left\|\theta_{i,\ell}^{(t)} - \bar{\theta}^{(t)}\right\|_2^2\right]$$
$$+ \mathcal{O}\left(L^2\alpha^2\right)\mathbb{E}\left[\left\|\bar{\theta}^{(t)} - \theta^\star\right\|_2^2\right] + \mathcal{O}\left(L^2\alpha^6\sigma^2\right)$$
$$+ \mathcal{O}\left(\frac{L^2\alpha^2\sigma^2}{MH(1-\rho)}\right). \tag{48}$$

*Proof.* Note here that we can no longer bound it as we did in the i.i.d. case due to the presence of Markovian sampling, where data samples for each agent are temporally correlated rather than independent. To handle this, we take advantage of Corollary 3 to address the time correlation. We thus bound $VR_{Markov}$ as follows.

$$VR_{Markov} = 2\alpha^2 \mathbb{E}\left[\left\|\frac{1}{MH}\sum_{i=1}^{M}\sum_{\ell=0}^{H-1}\left(G_i(\theta^\star, o_{i,\ell}^{(t)}) - \bar{G}_i(\theta^\star) + \bar{G}_i(\theta^\star) - \bar{G}_i(\theta_{i,\ell}^{(t)}) + G_i(\theta_{i,\ell}^{(t)}, o_{i,\ell}^{(t)}) - G_i(\theta^\star, o_{i,\ell}^{(t)})\right)\right\|_2^2\right]$$

$$\leq \overbrace{6\alpha^2 \mathbb{E}\left[\left\|\frac{1}{MH}\sum_{i=1}^{M}\sum_{\ell=0}^{H-1}\left(G_i(\theta^\star, o_{i,\ell}^{(t)}) - \bar{G}_i(\theta^\star)\right)\right\|_2^2\right]}^{A_1} + \overbrace{6\alpha^2 \mathbb{E}\left[\left\|\frac{1}{MH}\sum_{i=1}^{M}\sum_{\ell=0}^{H-1}\left(\bar{G}_i(\theta_{i,\ell}^{(t)}) - \bar{G}_i(\theta^\star)\right)\right\|_2^2\right]}^{A_2}$$

$$+ \overbrace{6\alpha^2 \mathbb{E}\left[\left\|\frac{1}{MH}\sum_{i=1}^{M}\sum_{\ell=0}^{H-1}\left(G_i(\theta_{i,\ell}^{(t)}, o_{i,\ell}^{(t)}) - G_i(\theta^\star, o_{i,\ell}^{(t)})\right)\right\|_2^2\right]}^{A_3}. \tag{49}$$

We proceed to bound $A_2$ as:

$$
\begin{aligned}
A_2 &\leq \mathcal{O}\left(\frac{L^2\alpha^2}{MH}\right) \sum_{i=1}^{M} \sum_{\ell=0}^{H-1} \mathbb{E}\left[\left\|\theta_{i,\ell}^{(t)} - \theta^\star\right\|_2^2\right] \\
&\leq \mathcal{O}\left(\frac{L^2\alpha^2}{MH}\right) \sum_{i=1}^{M} \sum_{\ell=0}^{H-1} \mathbb{E}\left[\left\|\theta_{i,\ell}^{(t)} - \bar{\theta}^{(t)}\right\|_2^2\right] + \mathcal{O}\left(\frac{L^2\alpha^2}{MH}\right) \sum_{i=1}^{M} \sum_{\ell=0}^{H-1} \mathbb{E}\left[\left\|\bar{\theta}^{(t)} - \theta^\star\right\|_2^2\right] \\
&= \mathcal{O}\left(\frac{L^2\alpha^2}{MH}\right) \sum_{i=1}^{M} \sum_{\ell=0}^{H-1} \mathbb{E}\left[\left\|\theta_{i,\ell}^{(t)} - \bar{\theta}^{(t)}\right\|_2^2\right] + \mathcal{O}\left(L^2\alpha^2\right) \mathbb{E}\left[\left\|\bar{\theta}^{(t)} - \theta^\star\right\|_2^2\right],
\end{aligned}
\tag{50}
$$

where we used Assumption 1. We can obtain the same bound for $A_3$ with the same reasoning.

Now we continue to bound the term $A_1$. With a slight overload of notation, denote $e_{i,\ell} = G_i(\theta^\star, o_{i,\ell}^{(t)}) - \bar{G}_i(\theta^\star)$, and we obtain

$$
\begin{aligned}
A_1 &= \mathcal{O}\left(\frac{\alpha^2}{M^2H^2}\right) \mathbb{E}\left[\left\|\sum_{i=1}^{M} \sum_{\ell=0}^{H-1} e_{i,\ell}\right\|_2^2\right] \\
&= \overbrace{\mathcal{O}\left(\frac{\alpha^2}{M^2H^2}\right) \sum_{i=1}^{M} \sum_{\ell=0}^{H-1} \mathbb{E}\left[\|e_{i,\ell}\|_2^2\right]}^{B_1} + \overbrace{\mathcal{O}\left(\frac{\alpha^2}{M^2H^2}\right) \sum_{i=1}^{M} \sum_{\ell_1 \neq \ell_2} \mathbb{E}\left[\langle e_{i,\ell_1}, e_{i,\ell_2}\rangle\right]}^{B_2} \\
&\quad + \overbrace{\mathcal{O}\left(\frac{\alpha^2}{M^2H^2}\right) \sum_{i \neq j} \sum_{\ell=0}^{H-1} \mathbb{E}\left[\langle e_{i,\ell}, e_{j,\ell}\rangle\right]}^{B_3} + \overbrace{\mathcal{O}\left(\frac{\alpha^2}{M^2H^2}\right) \sum_{i \neq j} \sum_{\ell_1 \neq \ell_2} \mathbb{E}\left[\langle e_{i,\ell_1}, e_{j,\ell_2}\rangle\right]}^{B_4}.
\end{aligned}
\tag{51}
$$

We then bound the four terms separately.

For the term $B_1$, we have

$$
\begin{aligned}
B_1 &\leq \mathcal{O}\left(\frac{\alpha^2}{M^2H^2}\right) \sum_{i=1}^{M} \sum_{\ell=0}^{H-1} \mathbb{E}\left[\left\|G_i(\theta^\star, o_{i,\ell}^{(t)})\right\|_2^2 + \left\|\bar{G}_i(\theta^\star)\right\|_2^2\right] \\
&\leq \mathcal{O}\left(\frac{L^2\alpha^2\sigma^2}{MH}\right),
\end{aligned}
\tag{52}
$$

where we used Corollary 2.

For the term $B_2$, we have

$$
\begin{aligned}
B_2 &\overset{(a)}{=} \mathcal{O}\left(\frac{\alpha^2}{M^2 H^2}\right) \sum_{i=1}^{M} \sum_{\ell_1 \neq \ell_2} \mathbb{E}\left[\mathbb{E}\left[\langle e_{i,\ell_1}, e_{i,\ell_2}\rangle \,|\, o_{i,\ell_1}^{(t)}\right]\right] \\
&= \mathcal{O}\left(\frac{\alpha^2}{M^2 H^2}\right) \sum_{i=1}^{M} \sum_{\ell_1 \neq \ell_2} \mathbb{E}\left[\left\langle e_{i,\ell_1}, \mathbb{E}\left[e_{i,\ell_2} | o_{i,\ell_1}^{(t)}\right]\right\rangle\right] \\
&\leq \mathcal{O}\left(\frac{\alpha^2}{M^2 H^2}\right) \sum_{i=1}^{M} \sum_{\ell_1 \neq \ell_2} \mathbb{E}\left[\|e_{i,\ell_1}\|_2 \left\|\mathbb{E}\left[e_{i,\ell_2} | o_{i,\ell_1}^{(t)}\right]\right\|_2\right] \\
&\overset{(b)}{\leq} \mathcal{O}\left(\frac{L\alpha^2}{M^2 H^2}\right) \sum_{i=1}^{M} \sum_{\ell_1=0}^{H-1} \sum_{\ell_2 > \ell_1} \rho^{\ell_2 - \ell_1} \sigma \mathbb{E}\left[\|e_{i,\ell_1}\|_2\right] \\
&\leq \mathcal{O}\left(\frac{L\alpha^2}{M^2 H^2}\right) \sum_{i=1}^{M} \sum_{\ell_1=0}^{H-1} \sum_{\ell_2 > \ell_1} \rho^{\ell_2 - \ell_1} \sigma \mathbb{E}\left[\left\|G_i(\theta^\star, o_{i,\ell_1}^{(t)})\right\|_2 + \left\|\bar{G}_i(\theta^\star)\right\|_2\right] \\
&\overset{(c)}{\leq} \mathcal{O}\left(\frac{L^2 \alpha^2}{M^2 H^2}\right) \sum_{i=1}^{M} \sum_{\ell_1=0}^{H-1} \sum_{\ell_2 > \ell_1} \rho^{\ell_2 - \ell_1} \sigma^2 \\
&\leq \mathcal{O}\left(\frac{L^2 \alpha^2 \sigma^2}{M^2 H^2}\right) \sum_{i=1}^{M} \sum_{\ell_1=0}^{H-1} (1 + \rho + \rho^2 + \cdots) \\
&\leq \mathcal{O}\left(\frac{L^2 \alpha^2 \sigma^2}{M H (1 - \rho)}\right),
\end{aligned}
\tag{53}
$$

where we used the tower property of expectation in $(a)$, Corollary 3 in $(b)$, and Corollary 2 in $(c)$.

For the term $B_3$, we have

$$
\begin{aligned}
B_3 &\overset{(a)}{=} \mathcal{O}\left(\frac{\alpha^2}{M^2 H^2}\right) \sum_{i \neq j} \sum_{\ell=0}^{H-1} \langle \mathbb{E}[e_{i,\ell}], \mathbb{E}[e_{j,\ell}]\rangle \\
&\leq \mathcal{O}\left(\frac{\alpha^2}{M^2 H^2}\right) \sum_{i \neq j} \sum_{\ell=0}^{H-1} \|\mathbb{E}[e_{i,\ell}]\|_2 \|\mathbb{E}[e_{j,\ell}]\|_2 \\
&\overset{(b)}{=} \mathcal{O}\left(\frac{\alpha^2}{M^2 H^2}\right) \sum_{i \neq j} \sum_{\ell=0}^{H-1} \|\mathbb{E}[\mathbb{E}[e_{i,\ell}|o_{i,tH+\ell-\bar{\tau}}]]\|_2 \|\mathbb{E}[\mathbb{E}[e_{j,\ell}|o_{j,tH+\ell-\bar{\tau}}]]\|_2 \\
&\overset{(c)}{\leq} \mathcal{O}\left(\frac{\alpha^2}{M^2 H^2}\right) \sum_{i \neq j} \sum_{\ell=0}^{H-1} \mathbb{E}\left[\|\mathbb{E}[e_{i,\ell}|o_{i,tH+\ell-\bar{\tau}}]\|_2\right] \mathbb{E}\left[\|\mathbb{E}[e_{j,\ell}|o_{j,tH+\ell-\bar{\tau}}]\|_2\right] \\
&\overset{(d)}{\leq} \mathcal{O}\left(\frac{L^2 \alpha^2}{M^2 H^2}\right) \sum_{i \neq j} \sum_{\ell=0}^{H-1} \rho^{2\bar{\tau}} \sigma^2 \\
&\overset{(e)}{\leq} \mathcal{O}\left(\frac{L^2 \alpha^6 \sigma^2}{H}\right),
\end{aligned}
\tag{54}
$$

where $(a)$ is a result of Assumption 4, $(b)$ uses the tower property of expectation, $(c)$ uses the fact that $\|\mathbb{E}[x]\|_2 \leq \mathbb{E}[\|x\|_2]$, $(d)$ is the application of Corollary 3, and $(e)$ uses the definition that $\rho^{\bar{\tau}} \leq \alpha^2$. Note here that we are allowed to condition on $\bar{\tau}$ time-steps before because $t \geq \bar{\tau}$.

Similarly, we can achieve the bound for $B_4$ as

$$
B_4 \leq \mathcal{O}\left(L^2 \alpha^6 \sigma^2\right).
\tag{55}
$$

Further plugging the bounds for $B_1$ to $B_4$ into (51) yields

$$
\begin{aligned}
A_1 &\leq \mathcal{O}\left(\frac{L^2\alpha^2\sigma^2}{MH}\right) + \mathcal{O}\left(\frac{L^2\alpha^2\sigma^2}{MH(1-\rho)}\right) + \mathcal{O}\left(\frac{L^2\alpha^6\sigma^2}{H}\right) + \mathcal{O}\left(L^2\alpha^6\sigma^2\right) \\
&\leq \mathcal{O}\left(L^2\alpha^6\sigma^2\right) + \mathcal{O}\left(\frac{L^2\alpha^2\sigma^2}{MH(1-\rho)}\right).
\end{aligned}
\tag{56}
$$

Finally, plugging the bounds for $A_1$ to $A_3$ into (49) yields

$$
VR_{Markov} \leq \mathcal{O}\left(\frac{L^2\alpha^2}{MH}\right) \sum_{i=1}^{M}\sum_{\ell=0}^{H-1} \mathbb{E}\left[\left\|\theta_{i,\ell}^{(t)} - \bar{\theta}^{(t)}\right\|_2^2\right] + \mathcal{O}\left(L^2\alpha^2\right)\mathbb{E}\left[\left\|\bar{\theta}^{(t)} - \theta^\star\right\|_2^2\right] + \mathcal{O}\left(L^2\alpha^6\sigma^2\right) + \mathcal{O}\left(\frac{L^2\alpha^2\sigma^2}{MH(1-\rho)}\right).
\tag{57}
$$

The proof is then complete. $\square$

With the upper-bound for the variance reduction term under Markovian sampling, we now introduce the following lemma, which bounds the one-step recursion of the distance to the optimal parameter $\theta^\star$ in the Markovian sampling setting. This lemma lays the foundation for understanding the impact of time correlations on the convergence behavior of `FedHSA`.

**Lemma 4.** *Suppose Assumptions 1 to 4 hold. Then the following holds for* `FedHSA` $\forall t \geq \bar{\tau}$:

$$
\begin{aligned}
\mathbb{E}\left[\left\|\bar{\theta}^{(t+1)} - \theta^\star\right\|_2^2\right] &\leq \left(1 - \alpha\mu + \mathcal{O}\left(L^2\alpha^2\right)\right)\mathbb{E}\left[\left\|\bar{\theta}^{(t)} - \theta^\star\right\|_2^2\right] + \mathcal{O}\left(\frac{L^2\alpha^2}{MH} + \frac{L^2\alpha}{\mu MH}\right)\sum_{i=1}^{M}\sum_{\ell=0}^{H-1}\mathbb{E}\left[\left\|\theta_{i,\ell}^{(t)} - \bar{\theta}^{(t)}\right\|_2^2\right] \\
&\quad + \mathcal{O}\left(L^2\alpha^6\sigma^2\right) + \mathcal{O}\left(\frac{L^2\alpha^2\sigma^2}{MH(1-\rho)}\right) + \mathbb{E}\left[\left\langle \bar{\theta}^{(t)} - \theta^\star, \frac{2\alpha}{MH}\sum_{i=1}^{M}\sum_{\ell=0}^{H-1}\left(G_i(\theta_{i,\ell}^{(t)}) - \bar{G}_i(\theta_{i,\ell}^{(t)})\right)\right\rangle\right].
\end{aligned}
\tag{58}
$$

*Proof.* As in the i.i.d. case, we first write the error update rule as follows:

$$
\left\|\bar{\theta}^{(t+1)} - \theta^\star\right\|_2^2 = \left\|\bar{\theta}^{(t)} - \theta^\star\right\|_2^2 + \overbrace{\left\langle \bar{\theta}^{(t)} - \theta^\star, \frac{2\alpha}{MH}\sum_{i=1}^{M}\sum_{\ell=0}^{H-1}G_i(\theta_{i,\ell}^{(t)})\right\rangle}^{U_1} + \overbrace{\alpha^2\left\|\frac{1}{MH}\sum_{i=1}^{M}\sum_{\ell=0}^{H-1}G_i(\theta_{i,\ell}^{(t)})\right\|_2^2}^{U_2}.
\tag{59}
$$

We proceed to bound the term $U_2$ using Jensen's inequality. By taking expectation, we obtain:

$$
\mathbb{E}[U_2] \leq \overbrace{2\alpha^2\mathbb{E}\left[\left\|\frac{1}{MH}\sum_{i=1}^{M}\sum_{\ell=0}^{H-1}\left(G_i(\theta_{i,\ell}^{(t)}) - \bar{G}_i(\theta_{i,\ell}^{(t)})\right)\right\|_2^2\right]}^{U_{21}} + \overbrace{2\alpha^2\mathbb{E}\left[\left\|\frac{1}{MH}\sum_{i=1}^{M}\sum_{\ell=0}^{H-1}\bar{G}_i(\theta_{i,\ell}^{(t)})\right\|_2^2\right]}^{U_{22}}.
\tag{60}
$$

Note here that the term $U_{22}$ only involves the true operators, and thus we do not need to consider the effect of Markovian sampling. Therefore, it can be upper-bounded exactly as the term $T_2$ in the i.i.d. case, i.e.:

$$
U_{22} \leq \mathcal{O}\left(\frac{L^2\alpha^2}{MH}\right)\sum_{i=1}^{M}\sum_{\ell=0}^{H-1}\mathbb{E}\left[\left\|\theta_{i,\ell}^{(t)} - \bar{\theta}^{(t)}\right\|_2^2\right] + \mathcal{O}\left(L^2\alpha^2\right)\mathbb{E}\left[\left\|\bar{\theta}^{(t)} - \theta^\star\right\|_2^2\right].
\tag{61}
$$

The term $U_{21}$ is exactly the variance reduction term we bounded in Lemma 3. Plugging the bounds for $U_{21}$ and $U_{22}$ into (60) yields

$$
\mathbb{E}[U_2] \leq \mathcal{O}\left(\frac{L^2\alpha^2}{MH}\right)\sum_{i=1}^{M}\sum_{\ell=0}^{H-1}\mathbb{E}\left[\left\|\theta_{i,\ell}^{(t)} - \bar{\theta}^{(t)}\right\|_2^2\right] + \mathcal{O}\left(L^2\alpha^2\right)\mathbb{E}\left[\left\|\bar{\theta}^{(t)} - \theta^\star\right\|_2^2\right] + \mathcal{O}\left(L^2\alpha^6\sigma^2\right) + \mathcal{O}\left(\frac{L^2\alpha^2\sigma^2}{MH(1-\rho)}\right).
\tag{62}
$$

For the term $\mathbb{E}[U_1]$, we can decompose it into two parts:

$$\mathbb{E}[U_1] = \overbrace{\mathbb{E}\left[\left\langle \bar{\theta}^{(t)} - \theta^\star, \frac{2\alpha}{MH} \sum_{i=1}^{M} \sum_{\ell=0}^{H-1} \left(G_i(\theta_{i,\ell}^{(t)}) - \bar{G}_i(\theta_{i,\ell}^{(t)})\right)\right\rangle\right]}^{C_1} + \overbrace{\mathbb{E}\left[\left\langle \bar{\theta}^{(t)} - \theta^\star, \frac{2\alpha}{MH} \sum_{i=1}^{M} \sum_{\ell=0}^{H-1} \bar{G}_i(\theta_{i,\ell}^{(t)})\right\rangle\right]}^{C_2}.$$
(63)

Note that the second part $C_2$ can be bounded similarly as the term $T_1$ in the i.i.d. case:

$$C_2 \leq -\alpha\mu\mathbb{E}\left[\left\|\bar{\theta}^{(t)} - \theta^\star\right\|_2^2\right] + \frac{\alpha L^2}{\mu MH} \sum_{i=1}^{M} \sum_{\ell=0}^{H-1} \mathbb{E}\left[\left\|\theta_{i,\ell}^{(t)} - \bar{\theta}^{(t)}\right\|_2^2\right].$$
(64)

Plugging (64) into (63), together with (62) into (59) and taking expectation on both sides, we obtain

$$\mathbb{E}\left[\left\|\bar{\theta}^{(t+1)} - \theta^\star\right\|_2^2\right] \leq \left(1 - \alpha\mu + \mathcal{O}\left(L^2\alpha^2\right)\right)\mathbb{E}\left[\left\|\bar{\theta}^{(t)} - \theta^\star\right\|_2^2\right] + \mathcal{O}\left(\frac{L^2\alpha^2}{MH} + \frac{L^2\alpha}{\mu MH}\right) \sum_{i=1}^{M} \sum_{\ell=0}^{H-1} \mathbb{E}\left[\left\|\theta_{i,\ell}^{(t)} - \bar{\theta}^{(t)}\right\|_2^2\right]$$

$$+ \overbrace{\mathbb{E}\left[\left\langle \bar{\theta}^{(t)} - \theta^\star, \frac{2\alpha}{MH} \sum_{i=1}^{M} \sum_{\ell=0}^{H-1} \left(G_i(\theta_{i,\ell}^{(t)}) - \bar{G}_i(\theta_{i,\ell}^{(t)})\right)\right\rangle\right]}^{T_{bias}} + \mathcal{O}\left(L^2\alpha^6\sigma^2\right) + \mathcal{O}\left(\frac{L^2\alpha^2\sigma^2}{MH(1-\rho)}\right).$$
(65)

$\square$

Comparing Lemma 4 with Lemma 2, we observe that the one-step bound for the i.i.d. case can be recovered in Lemma 4 if data are i.i.d., i.e., $\rho = 0$. The only term that impedes further bounding in Lemma 4 is the one $T_{bias}$ capturing the bias caused by Markovian sampling, which equals zero in the i.i.d. case as proved in Eq. (39). The next lemma then focuses on bounding this distinct bias term in the Markovian case.

## C.1 Bounding of the Markovian Bias

It is worth noting that the **goal** of bounding the bias term is to ensure that the term $\mathbb{E}\left[\left\|\bar{\theta}^{(t)} - \theta^\star\right\|_2^2\right]$ on the R.H.S. has a scaling factor no larger than $\alpha\mu$, along with a constant term scaled by an order of $\alpha$ that is either higher than or equal to 3, or an order of 2 but scaled down by the number of agents $M$, thus demonstrating the benefits of collaboration. Unfortunately, directly applying Eq. (17) cannot yield the desired result. Instead, by applying Corollary 3 to account for the time correlation in data samples, we condition on a parameter at least $\bar{\tau}$ time-steps earlier in time. Specifically, since in the bias term, the parameter $\theta_{i,\ell}^{(t)}$ is from the $t$-th round and $\ell$-th local step with $t \geq \bar{\tau}$, we condition on $\bar{\theta}^{(t-\bar{\tau})}$, which is at least $\bar{\tau}$ time-steps earlier. For this reason, we first provide an upper-bound for the term $\mathbb{E}\left[\left\|\bar{\theta}^{(t)} - \bar{\theta}^{(t-\tau)}\right\|_2^2\right]$, which comes in handy in subsequent proofs.

**Lemma 5.** *Suppose all the conditions in Lemma 4 hold. Then there exist a universal constant $\bar{C} \geq 1$ such that by selecting $\eta \leq 1/(\bar{\tau}\tilde{C}HL^2)$ and $\alpha_g = 1$, the following is true for FedHSA for all $t \geq 2\bar{\tau}$:*

$$\mathbb{E}\left[\left\|\bar{\theta}^{(t)} - \bar{\theta}^{(t-\bar{\tau})}\right\|_2^2\right] \leq \mathcal{O}\left(\bar{\tau}^2 L^2\alpha^2\right)\mathbb{E}\left[\left\|\bar{\theta}^{(t)} - \theta^\star\right\|_2^2\right] + \mathcal{O}\left(\bar{\tau}^2 L^4\alpha^4\sigma^2 + \frac{\bar{\tau}^2 L^2\alpha^2\sigma^2}{MH(1-\rho)}\right).$$
(66)

*Proof.* Observe that

$$
\begin{aligned}
\left\|\bar{\theta}^{(t+1)} - \theta^\star\right\|_2^2 &= \left\|\bar{\theta}^{(t)} - \theta^\star\right\|_2^2 + 2\alpha\left\langle \bar{\theta}^{(t)} - \theta^\star, \frac{1}{MH}\sum_{i=1}^{M}\sum_{\ell=0}^{H-1} G_i(\theta_{i,\ell}^{(t)})\right\rangle + \alpha^2\left\|\frac{1}{MH}\sum_{i=1}^{M}\sum_{\ell=0}^{H-1} G_i(\theta_{i,\ell}^{(t)})\right\|_2^2 \\
&\leq \left\|\bar{\theta}^{(t)} - \theta^\star\right\|_2^2 + \alpha\left\|\frac{1}{MH}\sum_{i=1}^{M}\sum_{\ell=0}^{H-1} G_i(\theta_{i,\ell}^{(t)})\right\|_2^2 + \alpha\left\|\bar{\theta}^{(t)} - \theta^\star\right\|_2^2 + \alpha^2\left\|\frac{1}{MH}\sum_{i=1}^{M}\sum_{\ell=0}^{H-1} G_i(\theta_{i,\ell}^{(t)})\right\|_2^2 \\
&\leq (1+\alpha)\left\|\bar{\theta}^{(t)} - \theta^\star\right\|_2^2 + 2\alpha\left\|\frac{1}{MH}\sum_{i=1}^{M}\sum_{\ell=0}^{H-1} G_i(\theta_{i,\ell}^{(t)})\right\|_2^2,
\end{aligned}
\tag{67}
$$

where we used the fact that $\alpha \leq 1$, and $2\langle a, b\rangle \leq \|a\|_2^2 + \|b\|_2^2$, for all $(a,b) \in \mathbb{R}^d \times \mathbb{R}^d$. Taking expectation of both sides and plugging in the bound for $2\mathbb{E}[U_2]/\alpha$ in Eq. (62) yields

$$
\begin{aligned}
\mathbb{E}\left[\left\|\bar{\theta}^{(t+1)} - \bar{\theta}^\star\right\|_2^2\right] &\leq (1+\alpha)\mathbb{E}\left[\left\|\bar{\theta}^{(t)} - \theta^\star\right\|_2^2\right] + \mathcal{O}\left(\frac{L^2\alpha}{MH}\right)\sum_{i=1}^{M}\sum_{\ell=0}^{H-1}\mathbb{E}\left[\left\|\theta_{i,\ell}^{(t)} - \bar{\theta}^{(t)}\right\|_2^2\right] + \mathcal{O}\left(L^2\alpha\right)\mathbb{E}\left[\left\|\bar{\theta}^{(t)} - \theta^\star\right\|_2^2\right] \\
&\quad + \mathcal{O}\left(L^2\alpha^5\sigma^2\right) + \mathcal{O}\left(\frac{L^2\alpha\sigma^2}{MH(1-\rho)}\right) \\
&\leq (1+\alpha)\mathbb{E}\left[\left\|\bar{\theta}^{(t)} - \theta^\star\right\|_2^2\right] + \mathcal{O}\left(L^4\alpha^3\right)\left(\mathbb{E}\left[\left\|\bar{\theta}^{(t)} - \theta^\star\right\|_2^2\right] + \sigma^2\right) + \mathcal{O}\left(L^2\alpha\right)\mathbb{E}\left[\left\|\bar{\theta}^{(t)} - \theta^\star\right\|_2^2\right] \\
&\quad + \mathcal{O}\left(L^2\alpha^5\sigma^2\right) + \mathcal{O}\left(\frac{L^2\alpha\sigma^2}{MH(1-\rho)}\right) \\
&\leq \left(1+\alpha + \mathcal{O}\left(L^4\alpha^3 + L^2\alpha\right)\right)\mathbb{E}\left[\left\|\bar{\theta}^{(t)} - \theta^\star\right\|_2^2\right] + \mathcal{O}\left(L^2\alpha^5\sigma^2\right) + \mathcal{O}\left(\frac{L^2\alpha\sigma^2}{MH(1-\rho)}\right) \\
&\quad + \mathcal{O}\left(L^4\alpha^3\sigma^2\right) \\
&\leq \left(1 + \mathcal{O}\left(L^2\alpha\right)\right)\mathbb{E}\left[\left\|\bar{\theta}^{(t)} - \theta^\star\right\|_2^2\right] + \mathcal{O}\left(L^4\alpha^3\sigma^2\right) + \mathcal{O}\left(\frac{L^2\alpha\sigma^2}{MH(1-\rho)}\right),
\end{aligned}
\tag{68}
$$

where in the second inequality we used Lemma 1 and we selected $\bar{\tau}$ large enough such that $\rho^{\bar{\tau}} \leq \alpha^2$, and in the last inequality we used the fact that $\alpha L \leq 1$.

Therefore, for any $t' \in [t - \bar{\tau}, t]$, we can write

$$
\begin{aligned}
\mathbb{E}\left[\left\|\bar{\theta}^{(t')} - \theta^\star\right\|_2^2\right] &\leq \left(1 + \mathcal{O}\left(L^2\alpha\right)\right)^{\bar{\tau}}\mathbb{E}\left[\left\|\bar{\theta}^{(t-\tau)} - \theta^\star\right\|_2^2\right] + \sum_{\ell=0}^{\tau-1}\left(1 + \mathcal{O}\left(L^2\alpha\right)\right)^{\ell}\mathcal{O}\left(L^4\alpha^3\sigma^2 + \frac{L^2\alpha\sigma^2}{MH(1-\rho)}\right) \\
&\leq \mathcal{O}\left(\mathbb{E}\left[\left\|\bar{\theta}^{(t-\tau)} - \theta^\star\right\|_2^2\right]\right) + \tau\mathcal{O}\left(L^4\alpha^3\sigma^2 + \frac{L^2\alpha\sigma^2}{MH(1-\rho)}\right),
\end{aligned}
\tag{69}
$$

where we selected $\alpha \leq 1/(\tilde{C}L^2\bar{\tau})$, where $\tilde{C} \geq 1$ is the dominant constant in $\mathcal{O}\left(L^2\alpha\right)$, such that $\left(1 + \mathcal{O}\left(L^2\alpha\right)\right)^{\bar{\tau}} \leq (1 + 1/\bar{\tau})^{\bar{\tau}} \leq e$.

Next, observe that

$$
\begin{aligned}
\mathbb{E}\left[\left\|\bar{\theta}^{(t)} - \bar{\theta}^{(t-\bar{\tau})}\right\|_2^2\right] &\leq \bar{\tau} \sum_{m=t-\bar{\tau}}^{t-1} \mathbb{E}\left[\left\|\bar{\theta}^{(m+1)} - \bar{\theta}^{(m)}\right\|_2^2\right] \\
&= \bar{\tau} \sum_{m=t-\bar{\tau}}^{t-1} \mathbb{E}\left[\alpha^2 \left\|\frac{1}{MH}\sum_{i=1}^{M}\sum_{\ell=0}^{H-1} G_i(\theta_{i,\ell}^{(m)})\right\|_2^2\right] \\
&\overset{(a)}{\leq} \bar{\tau} \sum_{m=t-\bar{\tau}}^{t-1} \mathcal{O}\left(L^2\alpha^2\mathbb{E}\left[\left\|\bar{\theta}^{(m)} - \theta^\star\right\|_2^2\right] + L^4\alpha^4\sigma^2 + \frac{L^2\alpha^2\sigma^2}{MH(1-\rho)}\right) \\
&\overset{(b)}{\leq} \bar{\tau} \sum_{m=t-\bar{\tau}}^{t-1} \mathcal{O}\left(L^2\alpha^2\mathbb{E}\left[\left\|\bar{\theta}^{(t-\bar{\tau})} - \theta^\star\right\|_2^2\right] + \bar{\tau}L^6\alpha^5\sigma^2 + \frac{\bar{\tau}L^4\alpha^3\sigma^2}{MH(1-\rho)} + L^4\alpha^4\sigma^2 + \frac{L^2\alpha^2\sigma^2}{MH(1-\rho)}\right) \\
&\overset{(c)}{\leq} \bar{\tau} \sum_{m=t-\bar{\tau}}^{t-1} \mathcal{O}\left(L^2\alpha^2\mathbb{E}\left[\left\|\bar{\theta}^{(t-\bar{\tau})} - \theta^\star\right\|_2^2\right] + L^4\alpha^4\sigma^2 + \frac{L^2\alpha^2\sigma^2}{MH(1-\rho)}\right) \\
&\leq \mathcal{O}\left(\bar{\tau}^2 L^2\alpha^2\right)\mathbb{E}\left[\left\|\bar{\theta}^{(t-\bar{\tau})} - \theta^\star\right\|_2^2\right] + \mathcal{O}\left(\bar{\tau}^2 L^4\alpha^4\sigma^2 + \frac{\bar{\tau}^2 L^2\alpha^2\sigma^2}{MH(1-\rho)}\right) \\
&\leq \mathcal{O}\left(\bar{\tau}^2 L^2\alpha^2\right)\mathbb{E}\left[\left\|\bar{\theta}^{(t-\bar{\tau})} - \bar{\theta}^{(t)}\right\|_2^2\right] + \mathcal{O}\left(\bar{\tau}^2 L^2\alpha^2\right)\mathbb{E}\left[\left\|\bar{\theta}^{(t)} - \theta^\star\right\|_2^2\right] \\
&\quad + \mathcal{O}\left(\bar{\tau}^2 L^4\alpha^4\sigma^2 + \frac{\bar{\tau}^2 L^2\alpha^2\sigma^2}{MH(1-\rho)}\right),
\end{aligned}
\tag{70}
$$

where in $(a)$ we used the bound for $\mathbb{E}[U_2]$ and plugged in Lemma 1, i.e.,

$$
\mathbb{E}[U_2] \leq \mathcal{O}\left(L^2\alpha^2\right)\mathbb{E}\left[\left\|\bar{\theta}^{(t)} - \theta^\star\right\|_2^2\right] + \mathcal{O}\left(L^4\alpha^4\sigma^2\right) + \mathcal{O}\left(\frac{L^2\alpha^2\sigma^2}{MH(1-\rho)}\right).
\tag{71}
$$

Note here that we are allowed to use the bound for $\mathbb{E}[U_2]$ because we require $t \geq 2\bar{\tau}$, and thus $m \geq t - \bar{\tau} \geq \bar{\tau}$; in $(b)$ we used the result in Eq. (69), and in $(c)$ we selected $\alpha \leq 1/(L^2\bar{\tau})$ such that $\bar{\tau}L^6\alpha^5\sigma^2 \leq L^4\alpha^4\sigma^2$ and $\bar{\tau}L^4\alpha^3\sigma^2 \leq L^2\alpha^2\sigma^2$.

Suppose that the dominating constant in $\mathcal{O}\left(\bar{\tau}^2 L^2\alpha^2\right)\mathbb{E}\left[\left\|\bar{\theta}^{(t-1)} - \bar{\theta}^{(t)}\right\|_2^2\right]$ is $\bar{C} \geq 1$. We can then write

$$
(1 - \bar{C}\bar{\tau}^2 L^2\alpha^2)\mathbb{E}\left[\left\|\bar{\theta}^{(t)} - \bar{\theta}^{(t-1)}\right\|_2^2\right] \leq \mathcal{O}\left(\bar{\tau}^2 L^2\alpha^2\right)\mathbb{E}\left[\left\|\bar{\theta}^{(t)} - \theta^\star\right\|_2^2\right] + \mathcal{O}\left(\bar{\tau}^2 L^4\alpha^4\sigma^2 + \frac{\bar{\tau}^2 L^2\alpha^2\sigma^2}{MH(1-\rho)}\right).
\tag{72}
$$

By selecting $\alpha$ such that $\bar{C}\bar{\tau}^2 L^2\alpha^2 \leq 1/2$, i.e., $\alpha \leq 1/(\sqrt{2\bar{C}}\bar{\tau}L)$ we obtain

$$
\mathbb{E}\left[\left\|\bar{\theta}^{(t)} - \bar{\theta}^{(t-\bar{\tau})}\right\|_2^2\right] \leq \mathcal{O}\left(\bar{\tau}^2 L^2\alpha^2\right)\mathbb{E}\left[\left\|\bar{\theta}^{(t)} - \theta^\star\right\|_2^2\right] + \mathcal{O}\left(\bar{\tau}^2 L^4\alpha^4\sigma^2 + \frac{\bar{\tau}^2 L^2\alpha^2\sigma^2}{MH(1-\rho)}\right).
\tag{73}
$$

The proof is then complete. $\qquad\square$

With Lemma 5 at hand, we can arrive at the following lemma that bounds the Markovian bias:

**Lemma 6.** *Suppose all the conditions in Lemma 5 hold. Then the following is true for* `FedHSA` *for all $t \geq 2\bar{\tau}$:*

$$
\mathbb{E}\left[\overbrace{\left\langle \bar{\theta}^{(t)} - \theta^\star, \frac{2\alpha}{MH}\sum_{i=1}^{M}\sum_{\ell=0}^{H-1}\left(G_i(\theta_{i,\ell}^{(t)}) - \bar{G}_i(\theta_{i,\ell}^{(t)})\right)\right\rangle}^{C_1}\right] \leq \left(\frac{\alpha\mu}{2} + \mathcal{O}\left(\bar{\tau}L^2\alpha^2 + \frac{L^4\alpha^3}{\mu}\right)\right)\mathbb{E}\left[\left\|\bar{\theta}^{(t)} - \theta^\star\right\|_2^2\right]
$$
$$
+ \mathcal{O}\left(\frac{L^4\sigma^2\alpha^3}{\mu} + \frac{\tau L^2\sigma^2\alpha^2}{MH(1-\rho)}\right).
\tag{74}
$$

*Proof.* Observe that

$$
\overbrace{\mathbb{E}\left[\left\langle \bar{\theta}^{(t)} - \theta^{\star}, \frac{2\alpha}{MH} \sum_{i=1}^{M} \sum_{\ell=0}^{H-1} \left(G_i(\theta_{i,\ell}^{(t)}) - \bar{G}_i(\theta_{i,\ell}^{(t)})\right)\right\rangle\right]}^{C_1} = \overbrace{\mathbb{E}\left[\left\langle \bar{\theta}^{(t)} - \bar{\theta}^{(t-\bar{\tau})}, \frac{2\alpha}{MH} \sum_{i=1}^{M} \sum_{\ell=0}^{H-1} \left(G_i(\theta_{i,\ell}^{(t)}) - \bar{G}_i(\theta_{i,\ell}^{(t)})\right)\right\rangle\right]}^{D_1}
$$

$$
+ \overbrace{\mathbb{E}\left[\left\langle \bar{\theta}^{(t-\bar{\tau})} - \theta^{\star}, \frac{2\alpha}{MH} \sum_{i=1}^{M} \sum_{\ell=0}^{H-1} \left(G_i(\theta_{i,\ell}^{(t)}) - \bar{G}_i(\theta_{i,\ell}^{(t)})\right)\right\rangle\right]}^{D_2}.
$$

$$(75)$$

We then continue to bound the terms $D_1$ and $D_2$ separately. For the term $D_1$, we have

$$
\begin{aligned}
D_1 &\leq \frac{1}{\bar{\tau}}\mathbb{E}\left[\left\|\bar{\theta}^{(t)} - \bar{\theta}^{(t-\bar{\tau})}\right\|_2^2\right] + \bar{\tau}\alpha^2 \mathbb{E}\left[\left\|\frac{1}{MH}\sum_{i=1}^{M}\sum_{\ell=0}^{H-1}\left(G_i(\theta_{i,\ell}^{(t)}) - \bar{G}_i(\theta_{i,\ell}^{(t)})\right)\right\|_2^2\right] \\
&\leq \mathcal{O}\left(\bar{\tau}L^2\alpha^2\right)\mathbb{E}\left[\left\|\bar{\theta}^{(t)} - \theta^{\star}\right\|_2^2\right] + \mathcal{O}\left(\bar{\tau}L^4\alpha^4\sigma^2 + \frac{\bar{\tau}L^2\alpha^2\sigma^2}{MH(1-\rho)}\right) \\
&\quad + \mathcal{O}\left(\bar{\tau}L^2\alpha^2\right)\mathbb{E}\left[\left\|\bar{\theta}^{(t)} - \theta^{\star}\right\|_2^2\right] + \mathcal{O}\left(\bar{\tau}L^4\alpha^4\sigma^2\right) + \mathcal{O}\left(\frac{\bar{\tau}L^2\alpha^2\sigma^2}{MH(1-\rho)}\right) \\
&\leq \mathcal{O}\left(\bar{\tau}L^2\alpha^2\right)\mathbb{E}\left[\left\|\bar{\theta}^{(t)} - \theta^{\star}\right\|_2^2\right] + \mathcal{O}\left(\bar{\tau}L^4\alpha^4\sigma^2 + \frac{\bar{\tau}L^2\alpha^2\sigma^2}{MH(1-\rho)}\right),
\end{aligned}
$$

$$(76)$$

where we use Lemma 5 and Lemma 3 with the bound from Lemma 1 plugged in. Clearly the bound for $D_1$ is eligible for our goal.

For the term $D_2$, again, directly bounding will not suffice. Therefore, we introduce intermediate terms $\bar{G}_i(\bar{\theta}^{(t)})$ and $G_i(\bar{\theta}^{(t)}, o_{i,\ell}^{(t)})$.

$$
D_2 = \overbrace{\mathbb{E}\left[\left\langle \bar{\theta}^{(t-\bar{\tau})} - \theta^{\star}, \frac{2\alpha}{MH}\sum_{i=1}^{M}\sum_{\ell=0}^{H-1}\left(G_i(\bar{\theta}^{(t)}, o_{i,\ell}^{(t)}) - \bar{G}_i(\bar{\theta}^{(t)})\right)\right\rangle\right]}^{E_1}
$$

$$
+ \overbrace{\mathbb{E}\left[\left\langle \bar{\theta}^{(t-\bar{\tau})} - \theta^{\star}, \frac{2\alpha}{MH}\sum_{i=1}^{M}\sum_{\ell=0}^{H-1}\left(G_i(\theta_{i,\ell}^{(t)}, o_{i,\ell}^{(t)}) - G_i(\bar{\theta}^{(t)}, o_{i,\ell}^{(t)})\right)\right\rangle\right]}^{E_2}
$$

$$(77)$$

$$
+ \overbrace{\mathbb{E}\left[\left\langle \bar{\theta}^{(t-\bar{\tau})} - \theta^{\star}, \frac{2\alpha}{MH}\sum_{i=1}^{M}\sum_{\ell=0}^{H-1}\left(\bar{G}_i(\bar{\theta}^{(t)}) - \bar{G}_i(\theta_{i,\ell}^{(t)})\right)\right\rangle\right]}^{E_3}.
$$

We then bound these three terms separately.

For the term $E_2$, we have

$$
\begin{aligned}
E_2 &\leq \frac{\alpha}{\beta}\mathbb{E}\left[\left\|\bar{\theta}^{(t-\bar{\tau})} - \theta^\star\right\|_2^2\right] + \alpha\beta\mathbb{E}\left[\left\|\frac{1}{MH}\sum_{i=1}^{M}\sum_{\ell=0}^{H-1}\left(G_i(\theta_{i,\ell}^{(t)}, o_{i,\ell}^{(t)}) - G_i(\bar{\theta}^{(t)}, o_{i,\ell}^{(t)})\right)\right\|_2^2\right] \\
&\overset{(a)}{\leq} \frac{2\alpha}{\beta}\mathbb{E}\left[\left\|\bar{\theta}^{(t-\bar{\tau})} - \bar{\theta}^{(t)}\right\|_2^2\right] + \frac{2\alpha}{\beta}\mathbb{E}\left[\left\|\bar{\theta}^{(t)} - \theta^\star\right\|_2^2\right] + \alpha\beta\frac{1}{MH}\sum_{i=1}^{M}\sum_{\ell=0}^{H-1}L^2\mathbb{E}\left[\left\|\theta_{i,\ell}^{(t)} - \bar{\theta}^{(t)}\right\|_2^2\right] \\
&\overset{(b)}{\leq} \mathcal{O}\left(\bar{\tau}^2 L^2\alpha^3\right)\frac{1}{\beta}\mathbb{E}\left[\left\|\bar{\theta}^{(t)} - \theta^\star\right\|_2^2\right] + \frac{1}{\beta}\mathcal{O}\left(\bar{\tau}^2 L^4\alpha^5\sigma^2 + \frac{\bar{\tau}^2 L^2\alpha^3\sigma^2}{MH(1-\rho)}\right) + \frac{2\alpha}{\beta}\mathbb{E}\left[\left\|\bar{\theta}^{(t)} - \theta^\star\right\|_2^2\right] \\
&\quad + \mathcal{O}\left(\beta L^4\alpha^3\right)\left(\mathbb{E}\left[\left\|\bar{\theta}^{(t)} - \theta^\star\right\|_2^2\right] + \sigma^2\right) \\
&\leq \left(\frac{2\alpha}{\beta} + \mathcal{O}\left(\frac{\bar{\tau}^2 L^2\alpha^3}{\beta} + \beta L^4\alpha^3\right)\right)\mathbb{E}\left[\left\|\bar{\theta}^{(t)} - \theta^\star\right\|_2^2\right] + \mathcal{O}\left(\frac{\bar{\tau}^2 L^4\sigma^2\alpha^5}{\beta} + \frac{\bar{\tau}^2 L^2\sigma^2\alpha^3}{MH\beta(1-\rho)} + \beta L^4\sigma^2\alpha^3\right),
\end{aligned}
\tag{78}
$$

where we use Assumption 1 in $(a)$, Corollary 2 and the bound from Lemma 5 in $(b)$. We can achieve identical bound for $E_3$ with the same reasoning. With a proper choice of the parameter $\beta$ (which will be made later), the bounds for $E_2$ and $E_3$ also comply with the requirement.

For the term $E_1$, we further decompose it into three terms:

$$
\begin{aligned}
E_1 &= \overbrace{\mathbb{E}\left[\left\langle\bar{\theta}^{(t-\bar{\tau})} - \theta^\star, \frac{2\alpha}{MH}\sum_{i=1}^{M}\sum_{\ell=0}^{H-1}\left(G_i(\bar{\theta}^{(t-\bar{\tau})}, o_{i,\ell}^{(t)}) - \bar{G}_i(\bar{\theta}^{(t-\bar{\tau})})\right)\right\rangle\right]}^{F_1} \\
&\quad + \overbrace{\mathbb{E}\left[\left\langle\bar{\theta}^{(t-\bar{\tau})} - \theta^\star, \frac{2\alpha}{MH}\sum_{i=1}^{M}\sum_{\ell=0}^{H-1}\left(G_i(\bar{\theta}^{(t)}, o_{i,\ell}^{(t)}) - G_i(\bar{\theta}^{(t-\bar{\tau})}, o_{i,\ell}^{(t)})\right)\right\rangle\right]}^{F_2} \\
&\quad + \underbrace{\mathbb{E}\left[\left\langle\bar{\theta}^{(t-\bar{\tau})} - \theta^\star, \frac{2\alpha}{MH}\sum_{i=1}^{M}\sum_{\ell=0}^{H-1}\left(\bar{G}_i(\bar{\theta}^{(t-\bar{\tau})}) - \bar{G}_i(\bar{\theta}^{(t)})\right)\right\rangle\right]}_{F_3}.
\end{aligned}
\tag{79}
$$

For the term $F_2$, we have

$$
\begin{aligned}
F_2 &\leq \mathbb{E}\left[2\alpha L\left\|\bar{\theta}^{(t-\bar{\tau})} - \theta^\star\right\|_2\left\|\bar{\theta}^{(t-\bar{\tau})} - \bar{\theta}^{(t)}\right\|_2\right] \\
&\leq \alpha L\mathbb{E}\left[\bar{\tau}L\alpha\left\|\bar{\theta}^{(t-\bar{\tau})} - \theta^\star\right\|_2^2 + \frac{1}{\bar{\tau}L\alpha}\left\|\bar{\theta}^{(t-\bar{\tau})} - \bar{\theta}^{(t)}\right\|_2^2\right] \\
&\leq \alpha L\mathcal{O}\left(\mathbb{E}\left[\bar{\tau}L\alpha\left\|\bar{\theta}^{(t-\bar{\tau})} - \bar{\theta}^{(t)}\right\|_2^2 + \bar{\tau}L\alpha\left\|\bar{\theta}^{(t)} - \theta^\star\right\|_2^2 + \frac{1}{\bar{\tau}L\alpha}\left\|\bar{\theta}^{(t-\bar{\tau})} - \bar{\theta}^{(t)}\right\|_2^2\right]\right) \\
&\leq \alpha L\mathcal{O}\left(\mathbb{E}\left[\bar{\tau}L\alpha\left\|\bar{\theta}^{(t)} - \theta^\star\right\|_2^2 + \frac{1}{\bar{\tau}L\alpha}\left\|\bar{\theta}^{(t-\bar{\tau})} - \bar{\theta}^{(t)}\right\|_2^2\right]\right) \\
&\leq \mathcal{O}\left(\bar{\tau}L^2\alpha^2\right)\left(\mathbb{E}\left[\left\|\bar{\theta}^{(t)} - \theta^\star\right\|_2^2\right]\right) + \mathcal{O}\left(\bar{\tau}L^4\alpha^4\sigma^2 + \frac{\bar{\tau}L^2\alpha^2\sigma^2}{MH(1-\rho)}\right),
\end{aligned}
\tag{80}
$$

where we use Assumption 1, the fact that $\alpha\bar{\tau}L \leq 1$, and the bound from Lemma 3. Similarly, we can obtain the same bound for $F_3$. The bounds for $F_2, F_3$ satisfy the requirement as well.

Finally, for the term $F_1$, we have

$$
\begin{aligned}
F_1 &= \mathbb{E}\left[\mathbb{E}\left[\left\langle \bar{\theta}^{(t-\bar{\tau})} - \theta^\star, \frac{2\alpha}{MH}\sum_{i=1}^{M}\sum_{\ell=0}^{H-1}\left(G_i(\bar{\theta}^{(t-\bar{\tau})}, o_{i,\ell}^{(t)}) - \bar{G}_i(\bar{\theta}^{(t-\bar{\tau})})\right)\right\rangle \middle| \mathcal{F}_{-1}^{(t-\bar{\tau})}\right]\right] \\
&\stackrel{(a)}{=} \mathbb{E}\left[\left\langle \bar{\theta}^{(t-\bar{\tau})} - \theta^\star, \frac{2\alpha}{MH}\sum_{i=1}^{M}\sum_{\ell=0}^{H-1}\left(\mathbb{E}\left[G_i(\bar{\theta}^{(t-\bar{\tau})}, o_{i,\ell}^{(t)})\middle| \mathcal{F}_{-1}^{(t-\bar{\tau})}\right] - \bar{G}_i(\bar{\theta}^{(t-\bar{\tau})})\right)\right\rangle\right] \\
&\leq \mathbb{E}\left[\frac{2\alpha}{MH}\sum_{i=1}^{M}\sum_{\ell=0}^{H-1}\left\|\bar{\theta}^{(t-\bar{\tau})} - \theta^\star\right\|_2\left\|\mathbb{E}\left[G_i(\bar{\theta}^{(t-\bar{\tau})}, o_{i,\ell}^{(t)})\middle| \mathcal{F}_{-1}^{(t-\bar{\tau})}\right] - \bar{G}_i(\bar{\theta}^{(t-\bar{\tau})})\right\|_2\right] \\
&\stackrel{(b)}{\leq} \mathcal{O}\left(L\rho^{\bar{\tau}}\alpha\right)\mathbb{E}\left[\left\|\bar{\theta}^{(t-\bar{\tau})} - \theta^\star\right\|_2\left(\left\|\bar{\theta}^{(t-\bar{\tau})} - \theta^\star\right\|_2 + \sigma\right)\right] \\
&= \mathcal{O}\left(L\rho^{\bar{\tau}}\alpha\right)\mathbb{E}\left[\left\|\bar{\theta}^{(t-\bar{\tau})} - \theta^\star\right\|_2^2\right] + \mathcal{O}\left(L\rho^{\bar{\tau}}\alpha\sigma\right)\mathbb{E}\left[\left\|\bar{\theta}^{(t-\bar{\tau})} - \theta^\star\right\|_2\right] \\
&\leq \mathcal{O}\left(L\rho^{\bar{\tau}}\alpha\right)\mathbb{E}\left[\frac{1}{\alpha}\left\|\bar{\theta}^{(t-\bar{\tau})} - \theta^\star\right\|_2^2 + 2\sigma\left\|\bar{\theta}^{(t-\bar{\tau})} - \theta^\star\right\|_2 + \alpha\sigma^2\right] \\
&\stackrel{(c)}{\leq} \mathcal{O}\left(L\alpha^3\right)\mathbb{E}\left[\left(\frac{1}{\sqrt{\alpha}}\left\|\bar{\theta}^{(t-\bar{\tau})} - \theta^\star\right\|_2 + \sqrt{\alpha}\sigma\right)^2\right] \\
&\leq \mathcal{O}\left(L\alpha^3\right)\mathbb{E}\left[\frac{1}{\alpha}\left\|\bar{\theta}^{(t-\bar{\tau})} - \theta^\star\right\|_2^2 + \alpha\sigma^2\right] \\
&= \mathcal{O}\left(L\alpha^2\mathbb{E}\left[\left\|\bar{\theta}^{(t-\bar{\tau})} - \theta^\star\right\|_2^2\right] + L\alpha^4\sigma^2\right) \\
&\leq \mathcal{O}\left(L\alpha^2\mathbb{E}\left[\left\|\bar{\theta}^{(t-\bar{\tau})} - \theta^{(t)}\right\|_2^2\right] + L\alpha^2\mathbb{E}\left[\left\|\bar{\theta}^{(t)} - \theta^\star\right\|_2^2\right] + L\alpha^4\sigma^2\right) \\
&\stackrel{(d)}{\leq} \mathcal{O}\left(\bar{\tau}^2L^3\alpha^4\mathbb{E}\left[\left\|\bar{\theta}^{(t)} - \theta^\star\right\|_2^2\right] + \bar{\tau}^2L^5\alpha^6\sigma^2 + \frac{\bar{\tau}^2L^3\alpha^4\sigma^2}{MH(1-\rho)} + L\alpha^2\mathbb{E}\left[\left\|\bar{\theta}^{(t)} - \theta^\star\right\|_2^2\right] + L\alpha^4\sigma^2\right) \\
&\stackrel{(e)}{\leq} \mathcal{O}\left(L\alpha^2\right)\mathbb{E}\left[\left\|\bar{\theta}^{(t)} - \theta^\star\right\|_2^2\right] + \mathcal{O}\left(L\alpha^4\sigma^2 + \frac{\bar{\tau}^2L^3\alpha^4\sigma^2}{MH(1-\rho)}\right).
\end{aligned}
\tag{81}
$$

Here, in $(a)$ we used the fact that $\bar{\theta}^{(t-\bar{\tau})}$ is deterministic conditioned on $\mathcal{F}_{-1}^{(t-\bar{\tau})}$, $(b)$ is a result of Corollary 3 and Assumption 4, $(c)$ used the fact that $\rho^{\bar{\tau}} \leq \alpha^2$, $(d)$ used previous bounds for $\mathbb{E}\left[\left\|\bar{\theta}^{(t)} - \bar{\theta}^{(t-\bar{\tau})}\right\|_2^2\right]$, and $(e)$ used the fact that $\alpha \leq 1/(\bar{\tau}L^2)$, and hence $\bar{\tau}^2L^5\alpha^6 \leq L\alpha^4$.

We now plug in the bounds recursively to arrive at the final bound for the Markovian bias term. First, plugging in the bounds for $E_1$ with $E_1 = F_1 + F_2 + F_3$ yields

$$
\begin{aligned}
E_1 &\leq \mathcal{O}\left(L\alpha^2\right)\mathbb{E}\left[\left\|\bar{\theta}^{(t)} - \theta^\star\right\|_2^2\right] + \mathcal{O}\left(L\alpha^4\sigma^2\right) + \mathcal{O}\left(\frac{\bar{\tau}^2L^3\alpha^4\sigma^2}{MH(1-\rho)}\right) \\
&\quad + \mathcal{O}\left(\bar{\tau}L^2\alpha^2\right)\left(\mathbb{E}\left[\left\|\bar{\theta}^{(t)} - \theta^\star\right\|_2^2\right]\right) + \mathcal{O}\left(\bar{\tau}L^4\alpha^4\sigma^2 + \frac{\bar{\tau}L^2\alpha^2\sigma^2}{MH(1-\rho)}\right) \\
&\leq \mathcal{O}\left(\bar{\tau}L^2\alpha^2\right)\mathbb{E}\left[\left\|\bar{\theta}^{(t)} - \theta^\star\right\|_2^2\right] + \mathcal{O}\left(\bar{\tau}L^4\sigma^2\alpha^4\right) + \mathcal{O}\left(\frac{\bar{\tau}L^2\alpha^2\sigma^2}{MH(1-\rho)}\right),
\end{aligned}
\tag{82}
$$

where we use the fact that $L \geq 1$, $\bar{\tau} \geq 1$ and $\alpha^2 \leq 1/(\bar{\tau}L)$.

Second, with $D_2 = E_1 + E_2 + E_3$, we obtain

$$
\begin{aligned}
D_2 &\le \mathcal{O}\left(\bar{\tau}L^2\alpha^2\right)\mathbb{E}\left[\left\|\bar{\theta}^{(t)} - \theta^\star\right\|_2^2\right] + \mathcal{O}\left(\bar{\tau}L^4\sigma^2\alpha^4\right) + \mathcal{O}\left(\frac{\bar{\tau}L^2\alpha^2\sigma^2}{MH(1-\rho)}\right) \\
&\quad + \left(\frac{4\alpha}{\beta} + \mathcal{O}\left(\frac{\bar{\tau}^2L^2\alpha^3}{\beta} + \beta L^4\alpha^3\right)\right)\mathbb{E}\left[\left\|\bar{\theta}^{(t)} - \theta^\star\right\|_2^2\right] + \mathcal{O}\left(\frac{\bar{\tau}^2L^4\sigma^2\alpha^5}{\beta} + \frac{\bar{\tau}^2L^2\sigma^2\alpha^3}{MH\beta(1-\rho)} + \beta L^4\sigma^2\alpha^3\right) \\
&\le \left(\frac{4\alpha}{\beta} + \mathcal{O}\left(\bar{\tau}L^2\alpha^2 + \frac{\bar{\tau}^2L^2\alpha^3}{\beta} + \beta L^4\alpha^3\right)\right)\mathbb{E}\left[\left\|\bar{\theta}^{(t)} - \theta^\star\right\|_2^2\right] \\
&\quad + \mathcal{O}\left(\frac{\bar{\tau}^2L^4\sigma^2\alpha^5}{\beta} + \frac{\bar{\tau}^2L^2\sigma^2\alpha^3}{MH\beta(1-\rho)} + \beta L^4\sigma^2\alpha^3 + \bar{\tau}L^4\sigma^2\alpha^4 + \frac{\bar{\tau}L^2\sigma^2\alpha^2}{MH(1-\rho)}\right).
\end{aligned}
\tag{83}
$$

Finally, using $C_1 = D_1 + D_2$ yields

$$
\begin{aligned}
C_1 &\le \mathcal{O}\left(\bar{\tau}L^2\alpha^2\right)\mathbb{E}\left[\left\|\bar{\theta}^{(t)} - \theta^\star\right\|_2^2\right] + \mathcal{O}\left(\bar{\tau}L^4\alpha^4\sigma^2 + \frac{\bar{\tau}L^2\alpha^2\sigma^2}{MH(1-\rho)}\right) \\
&\quad + \left(\frac{4\alpha}{\beta} + \mathcal{O}\left(\bar{\tau}L^2\alpha^2 + \frac{\bar{\tau}^2L^2\alpha^3}{\beta} + \beta L^4\alpha^3\right)\right)\mathbb{E}\left[\left\|\bar{\theta}^{(t)} - \theta^\star\right\|_2^2\right] \\
&\quad + \mathcal{O}\left(\frac{\bar{\tau}^2L^4\sigma^2\alpha^5}{\beta} + \frac{\bar{\tau}^2L^2\sigma^2\alpha^3}{MH\beta(1-\rho)} + \beta L^4\sigma^2\alpha^3 + \bar{\tau}L^4\sigma^2\alpha^4 + \frac{\bar{\tau}L^2\sigma^2\alpha^2}{MH(1-\rho)}\right) \\
&\le \left(\frac{4\alpha}{\beta} + \mathcal{O}\left(\bar{\tau}L^2\alpha^2 + \frac{\bar{\tau}^2L^2\alpha^3}{\beta} + \beta L^4\alpha^3\right)\right)\mathbb{E}\left[\left\|\bar{\theta}^{(t)} - \theta^\star\right\|_2^2\right] \\
&\quad + \mathcal{O}\left(\frac{\bar{\tau}^2L^4\sigma^2\alpha^5}{\beta} + \frac{\bar{\tau}^2L^4\sigma^2\alpha^3}{MH\beta(1-\rho)} + \beta L^4\sigma^2\alpha^3 + \bar{\tau}L^4\sigma^2\alpha^4 + \frac{\bar{\tau}L^2\sigma^2\alpha^2}{MH(1-\rho)}\right).
\end{aligned}
\tag{84}
$$

By selecting $4\alpha/\beta = \alpha\mu/2$, i.e., $\beta = 8/\mu$, we obtain

$$
\begin{aligned}
C_1 &\le \left(\frac{\alpha\mu}{2} + \mathcal{O}\left(\bar{\tau}L^2\alpha^2 + \bar{\tau}^2L^2\mu\alpha^3 + \frac{L^4\alpha^3}{\mu}\right)\right)\mathbb{E}\left[\left\|\bar{\theta}^{(t)} - \theta^\star\right\|_2^2\right] \\
&\quad + \mathcal{O}\left(\bar{\tau}^2L^4\mu\sigma^2\alpha^5 + \frac{\bar{\tau}^2L^4\mu\sigma^2\alpha^3}{MH(1-\rho)} + \frac{L^4\sigma^2\alpha^3}{\mu} + \bar{\tau}L^4\sigma^2\alpha^4 + \frac{\bar{\tau}L^2\sigma^2\alpha^2}{MH(1-\rho)}\right) \\
&\le \left(\frac{\alpha\mu}{2} + \mathcal{O}\left(\bar{\tau}L^2\alpha^2 + \frac{L^4\alpha^3}{\mu}\right)\right)\mathbb{E}\left[\left\|\bar{\theta}^{(t)} - \theta^\star\right\|_2^2\right] + \mathcal{O}\left(\frac{L^4\sigma^2\alpha^3}{\mu} + \frac{\bar{\tau}L^2\sigma^2\alpha^2}{MH(1-\rho)}\right),
\end{aligned}
\tag{85}
$$

where we use the fact that $\mu \le 1$ and $\alpha \le 1/\bar{\tau}$. $\qquad\square$

**Lemma 7.** *Suppose all the conditions in Lemma 5 hold. Then the following holds for* `FedHSA` *for any* $t \ge 2\bar{\tau}$:

$$
\mathbb{E}\left[\left\|\bar{\theta}^{(t+1)} - \theta^\star\right\|_2^2\right] \le \left(1 - \frac{\alpha\mu}{2} + \mathcal{O}\left(\bar{\tau}L^2\alpha^2\right)\right)\mathbb{E}\left[\left\|\bar{\theta}^{(t)} - \theta^\star\right\|_2^2\right] + \mathcal{O}\left(\frac{\bar{\tau}\alpha^2L^2}{MH(1-\rho)} + \frac{\alpha^3L^4}{\mu}\right)\sigma^2.
\tag{86}
$$

**Proof of Lemma 7.**

*Proof.* Plugging Lemma 6 and Lemma 1 into Lemma 4, we obtain

$$
\begin{aligned}
\mathbb{E}\left[\left\|\bar{\theta}^{(t+1)} - \theta^\star\right\|_2^2\right] &\leq \left(1 - \alpha\mu + \mathcal{O}\left(L^2\alpha^2\right) + \mathcal{O}\left(\frac{L^4}{\mu}\alpha^3\right)\right)\mathbb{E}\left[\left\|\bar{\theta}^{(t)} - \theta^\star\right\|_2^2\right] + \mathcal{O}\left(\frac{L^4\sigma^2\alpha^3}{\mu}\right) + \mathcal{O}\left(\frac{L^2\alpha^2\sigma^2}{MH(1-\rho)}\right) \\
&\quad + \left(\frac{\alpha\mu}{2} + \mathcal{O}\left(\bar{\tau}L^2\alpha^2 + \frac{L^4\alpha^3}{\mu}\right)\right)\mathbb{E}\left[\left\|\bar{\theta}^{(t)} - \theta^\star\right\|_2^2\right] + \mathcal{O}\left(\frac{L^4\sigma^2\alpha^3}{\mu} + \frac{\bar{\tau}L^2\sigma^2\alpha^2}{MH(1-\rho)}\right) + \mathcal{O}\left(L^2\alpha^6\sigma^2\right) \\
&\leq \left(1 - \frac{\alpha\mu}{2} + \mathcal{O}\left(\bar{\tau}L^2\alpha^2 + \frac{L^4}{\mu}\alpha^3\right)\right)\mathbb{E}\left[\left\|\bar{\theta}^{(t)} - \theta^\star\right\|_2^2\right] + \mathcal{O}\left(\frac{L^4\sigma^2\alpha^3}{\mu} + \frac{\bar{\tau}L^2\alpha^2\sigma^2}{MH(1-\rho)}\right) \\
&\leq \left(1 - \frac{\alpha\mu}{2} + \mathcal{O}\left(\bar{\tau}L^2\alpha^2\right)\right)\mathbb{E}\left[\left\|\bar{\theta}^{(t)} - \theta^\star\right\|_2^2\right] + \mathcal{O}\left(\frac{L^4\sigma^2\alpha^3}{\mu} + \frac{\bar{\tau}L^2\alpha^2\sigma^2}{MH(1-\rho)}\right),
\end{aligned}
\tag{87}
$$

where we used the fact that $\alpha \leq \bar{\tau}\mu/L^2$. □

**Proof of Theorem 2.**

*Proof.* Applying Lemma 7 recursively, we obtain

$$
\begin{aligned}
\mathbb{E}\left[\left\|\bar{\theta}^{(T)} - \theta^\star\right\|_2^2\right] &\leq \left(1 - \frac{\alpha\mu}{4}\right)^{T-2\bar{\tau}}\mathbb{E}\left[\left\|\bar{\theta}^{(2\bar{\tau})} - \theta^\star\right\|_2^2\right] + \mathcal{O}\left(\frac{\bar{\tau}\alpha^2 L^2}{MH(1-\rho)} + \frac{\alpha^3 L^4}{\mu}\right)\sigma^2 \sum_{t=0}^{T-2\bar{\tau}-1}\left(1 - \frac{\alpha\mu}{4}\right)^t \\
&\leq \left(1 - \frac{\alpha\mu}{4}\right)^{T-2\bar{\tau}}\mathbb{E}\left[\left\|\bar{\theta}^{(2\bar{\tau})} - \theta^\star\right\|_2^2\right] + \mathcal{O}\left(\frac{\bar{\tau}\alpha L^2}{\mu MH(1-\rho)} + \frac{\alpha^2 L^4}{\mu^2}\right)\sigma^2,
\end{aligned}
\tag{88}
$$

where the first inequality holds because we select $\alpha \leq \mu/(4\bar{\tau}L^2 C')$, where $C'$ is greater than or equal to the dominant constant in $\mathcal{O}\left(\bar{\tau}L^2\alpha^2\right)$ such that $\mathcal{O}\left(\bar{\tau}L^2\alpha^2\right) \leq \alpha\mu/4$.

We proceed to bound the term $\mathbb{E}\left[\left\|\bar{\theta}^{(2\bar{\tau})} - \theta^\star\right\|_2^2\right]$. From the update rule of `FedHSA` in Eq. (10) and Eq. (11), we obtain

$$
\bar{\theta}^{(t+1)} - \theta^\star = \bar{\theta}^{(t)} - \theta^\star + \frac{\alpha}{MH}\sum_{i=1}^{M}\sum_{\ell=0}^{H-1} G_i(\theta_{i,\ell}^{(t)}),
\tag{89}
$$

where we subtracted $\theta^\star$ on both sides. Taking norm on both sides and applying triangle inequality yields

$$
\begin{aligned}
\left\|\bar{\theta}^{(t+1)} - \theta^\star\right\|_2 &\leq \left\|\bar{\theta}^{(t)} - \theta^\star\right\|_2 + \left\|\frac{\alpha}{MH}\sum_{i=1}^{M}\sum_{\ell=0}^{H-1} G_i(\theta_{i,\ell}^{(t)})\right\|_2 \\
&\leq \left\|\bar{\theta}^{(t)} - \theta^\star\right\|_2 + \frac{\alpha}{MH}\sum_{i=1}^{M}\sum_{\ell=0}^{H-1}\left\|G_i(\theta_{i,\ell}^{(t)})\right\|_2 \\
&\overset{(a)}{\leq} \left\|\bar{\theta}^{(t)} - \theta^\star\right\|_2 + \frac{\alpha L}{MH}\sum_{i=1}^{M}\sum_{\ell=0}^{H-1}\left(\left\|\theta_{i,\ell}^{(t)}\right\|_2 + \sigma\right) \\
&= \left\|\bar{\theta}^{(t)} - \theta^\star\right\|_2 + \frac{\alpha L}{MH}\sum_{i=1}^{M}\sum_{\ell=0}^{H-1}\left(\left\|\theta_{i,\ell}^{(t)} - \bar{\theta}^{(t)} + \bar{\theta}^{(t)} - \theta^\star + \theta^\star\right\|_2 + \sigma\right) \\
&\leq \left\|\bar{\theta}^{(t)} - \theta^\star\right\|_2 + \frac{\alpha L}{MH}\sum_{i=1}^{M}\sum_{\ell=0}^{H-1}\left(\left\|\theta_{i,\ell}^{(t)} - \bar{\theta}^{(t)}\right\|_2 + \left\|\bar{\theta}^{(t)} - \theta^\star\right\|_2 + \|\theta^\star\|_2 + \sigma\right) \\
&\overset{(b)}{\leq} \left\|\bar{\theta}^{(t)} - \theta^\star\right\|_2 + \frac{\alpha L}{MH}\sum_{i=1}^{M}\sum_{\ell=0}^{H-1}\left(L\alpha\left\|\bar{\theta}^{(t)} - \theta^\star\right\|_2 + L\alpha\sigma + \left\|\bar{\theta}^{(t)} - \theta^\star\right\|_2 + 2\sigma\right) \\
&\leq \left\|\bar{\theta}^{(t)} - \theta^\star\right\|_2 + \alpha L\left(2\left\|\bar{\theta}^{(t)} - \theta^\star\right\|_2 + 3\sigma\right) \\
&= (1 + 2\alpha L)\left\|\bar{\theta}^{(t)} - \theta^\star\right\|_2 + 3\alpha L\sigma,
\end{aligned}
\tag{90}
$$

where $(a)$ holds due to Assumption 1, and $(b)$ is a result of Lemma 1. Therefore, applying Eq. (90) recursively yields

$$
\begin{aligned}
\left\|\bar{\theta}^{(2\bar{\tau})} - \theta^\star\right\|_2 &\leq (1 + 2\alpha L)^{2\bar{\tau}}\left\|\bar{\theta}^{(0)} - \theta^\star\right\|_2 + 3\alpha L\sigma\sum_{t=0}^{2\bar{\tau}-1}(1 + 2\alpha L)^t \\
&\leq \left(1 + \frac{1}{2\bar{\tau}}\right)^{2\bar{\tau}}\left\|\bar{\theta}^{(0)} - \theta^\star\right\|_2 + 3\alpha L\sigma\sum_{t=0}^{2\bar{\tau}-1}\left(1 + \frac{1}{2\bar{\tau}}\right)^{2\bar{\tau}} \\
&\leq e\left\|\bar{\theta}^{(0)} - \theta^\star\right\|_2 + 2e\sigma \\
&= \mathcal{O}\left(\left\|\bar{\theta}^{(0)} - \theta^\star\right\|_2 + \sigma\right),
\end{aligned}
\tag{91}
$$

where we selected $\alpha \leq 1/(4\bar{\tau}L)$, and used the fact that $(1 + 1/x)^x \leq e, \forall x > 0$. Plugging Eq. (91) into Eq. (88) yields

$$
\begin{aligned}
\mathbb{E}\left[\left\|\bar{\theta}^{(T)} - \theta^\star\right\|_2^2\right] &\leq \left(1 - \frac{\alpha\mu}{4}\right)^{T-2\bar{\tau}}\mathcal{O}\left(\left\|\bar{\theta}^{(0)} - \theta^\star\right\|_2^2 + \sigma\right) + \mathcal{O}\left(\frac{\bar{\tau}\alpha L^2}{\mu MH(1-\rho)} + \frac{\alpha^2 L^4}{\mu^2}\right)\sigma^2 \\
&\leq \exp\left(-\frac{\mu}{4}\alpha(T - 2\bar{\tau})\right)\mathcal{O}\left(\left\|\bar{\theta}^{(0)} - \theta^\star\right\|_2^2 + \sigma\right) + \mathcal{O}\left(\frac{\bar{\tau}\alpha L^2}{\mu MH(1-\rho)} + \frac{\alpha^2 L^4}{\mu^2}\right)\sigma^2 \\
&= \exp\left(-\frac{\mu}{4}\alpha T\right)\exp\left(\frac{\mu\bar{\tau}\alpha}{2}\right)\mathcal{O}\left(\left\|\bar{\theta}^{(0)} - \theta^\star\right\|_2^2 + \sigma\right) + \mathcal{O}\left(\frac{\bar{\tau}\alpha L^2}{\mu MH(1-\rho)} + \frac{\alpha^2 L^4}{\mu^2}\right)\sigma^2 \\
&\leq \exp\left(-\frac{\mu}{4}\alpha T\right)\mathcal{O}\left(\left\|\bar{\theta}^{(0)} - \theta^\star\right\|_2^2 + \sigma\right) + \mathcal{O}\left(\frac{\bar{\tau}\alpha L^2}{\mu MH(1-\rho)} + \frac{\alpha^2 L^4}{\mu^2}\right)\sigma^2
\end{aligned}
\tag{92}
$$

where we used the fact that $1 - x \leq \exp(-x)$, $\alpha \leq 1/\bar{\tau}$, and $\mu \leq 1$ such that $\exp(\mu\tau\alpha/2) \leq \mathcal{O}(1)$. $\qquad\square$

# D  Experimental Results

In this section, we present numerical results for three heterogeneous federated SA tasks subject to Markovian noise, which provide empirical support for our theoretical framework. In our experiments, we aim to convey **two key messages**: (i) Our `FedHSA` algorithm eliminates the heterogeneity bias and (ii) `FedHSA` achieves linear speedup w.r.t. the number of agents. The experimental setups are described in detail in the sequel.

## D.1  Federated Quadratic Loss Minimization Problem

Consider the classical FL framework involving $M$ agents:

$$\min_{\theta \in \mathbb{R}^d} f(\theta) = \frac{1}{M} \sum_{i=1}^{M} f_i(\theta). \tag{93}$$

Here,

$$f_i(\theta) = \frac{1}{2} \theta^T A_i \theta - b_i^T \theta + c_i, \qquad \forall i \in [M] \tag{94}$$

where $A_i \in \mathbb{R}^{d \times d}$ is a positive definite matrix, $b_i \in \mathbb{R}^d$ is a $d$-dimensional vector, and $c_i \in \mathbb{R}$ is a scalar for $i = 1, \cdots, M$. Since $\{f_i\}_{i=1}^{M}$ are quadratic functions with positive definite $A_i$'s, the gradients are given by $\nabla f_i(\theta) = A_i \theta - b_i$, for $i = 1, \cdots, M$.

Problem (93) is a good fit for our federated SA framework. Specifically, the true local operator $\bar{G}_i(\cdot)$ corresponds to the true negative gradient $-\nabla f_i(\cdot)$ for each $i \in [M]$. At time step $t$, each agent $i \in [M]$ has access only to an estimator $G_i(\cdot)$ of the true operator, which is corrupted by additive Markovian noise:

$$G_i(\theta^{(t)}) = \bar{G}_i(\theta^{(t)}) + \xi_i^{(t)}, \qquad \forall i \in [M] \tag{95}$$

where the noise samples $\{\xi_i^{(t)}\}_{t \geq 0}$ are drawn from a discrete-time continous-state Markov chain.

We now explain how the Markovian noise is generated. For each agent $i$, we maintain a state vector $z_i \in \mathbb{R}^d$, initialized to zero, which evolves as

$$z_i^{(t+1)} = Q_i z_i^{(t)} + \epsilon_i^{(t)}. \tag{96}$$

Here, $Q_i \in \mathbb{R}^{d \times d}$ is a *Schur-stable matrix*, ensuring that all eigenvalues of $Q_i$ lie within the unit ball. This guarantees that $z_i^{(t)}$ does not diverge. The noise $\epsilon_i^{(t)}$ is a zero-mean Gaussian noise with variance $\sigma_\epsilon^2$ and covariance matrix given by $\sigma_\epsilon^2 I$, i.e., $\epsilon_i^{(t)} \sim \mathcal{N}(0, \sigma_\epsilon^2 I)$. The Markovian noise $\xi_i^{(t)}$ is directly taken from the state vector:

$$\xi_i^{(t)} = z_i^{(t)}. \tag{97}$$

It can be shown that this noise is Markovian (Tu & Recht, 2018), and mixes geometrically fast (as needed by our theory).

To validate (i), we consider Problem (93) and solve it using the conventional FL framework, where each agent performs local steps. We compare `FedHSA` with the existing local SGD approach, which does not include a correction term during local updates as shown in (9) (referred to as the "`Local SA` approach" in what follows). These two approaches are evaluated under both noiseless conditions and with the presence of additive Markovian noise. The experimental configurations are as follows: $M = 20$ agents, each agent performs $H = 10$ local steps, the learning rate is $\eta = 0.001$, the parameter dimension is $d = 10$, and the performance is measured by $E_t := \left\| \bar{\theta}^{(t)} - \theta^\star \right\|_2^2$.

As shown in Figure 2(a), even in the noiseless case where each agent has access to the true operator $\bar{G}_i$, the `Local SA` approach fails to converge to the optimal point $\theta^\star$ due to the heterogeneity bias, as explained in Proposition 1. In contrast, `FedHSA` demonstrates linear convergence towards $\theta^\star$. This result aligns with the theoretical prediction in (12), where the algorithm converges exponentially fast to $\theta^\star$ in expectation when $\sigma^2 = 0$ (noiseless case).

In Figure 2(b), we introduce additive Markovian noise with $\sigma_\epsilon^2 = 0.01$ into the local operators $G_i$'s. In this noisy setting, we observe that the `FedHSA` method exhibits a lower error floor compared to `Local SA`. This is

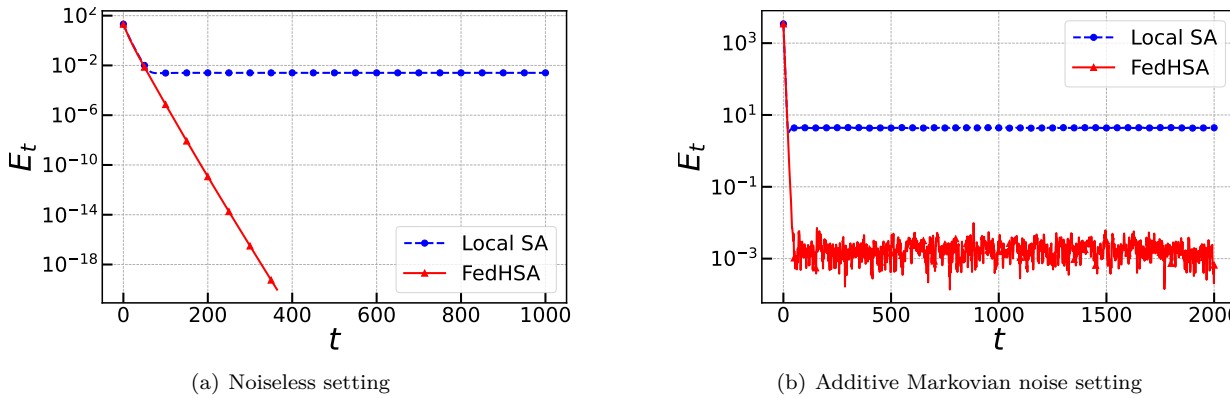

(a) Noiseless setting

(b) Additive Markovian noise setting

Figure 2: Comparison between `Local SA` and `FedHSA`

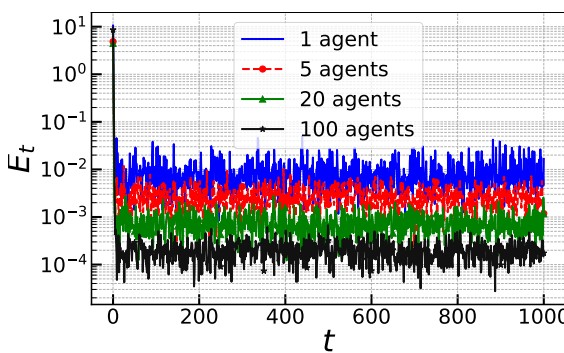

Figure 3: Comparison between different numbers of agents for the `FedHSA` algorithm.

because `FedHSA` effectively eliminates the heterogeneity bias, with the resulting error being solely attributed to the Markovian noise. Additionally, the impact of this noise is mitigated by a factor of $M$, owing to the linear speedup of the `FedHSA` algorithm.

To further verify the linear speedup effect in (ii), we compare the results of `FedHSA` with different numbers of agents. We consider the same problem (93) equipped with the `FedHSA` algorithm for $M = 1, 5, 20, 100$. The other configurations remain the same.

Figure 3 demonstrates a lower error floor with an increasing number of agents for our `FedHSA` algorithm. This is exactly what we expect, since Corollary 1 clearly states that with a proper choice of the step-size $\eta$, the expected error floor $d_T$ is upper-bounded by $\tilde{\mathcal{O}}(1/(MHT))$, which is inversely proportional to the number of agents $M$.

We also present results in scenarios where Assumption 2 does not hold, meaning each $A_i$ for $i \in [M]$ is symmetric but not necessarily positive definite. Consequently, the objective function $f_i$ becomes nonconvex. Removing this assumption significantly expands the applicability of our algorithms. As demonstrated in Figures 4 and 5, the observed results remain consistent with those obtained under strongly convex objectives.

## D.2   Federated TD Learning with Linear Function Approximation

We proceed to explore the application of `FedHSA` to FRL via focusing on the setting of federated TD learning with LFA. We begin by providing a detailed explanation of the problem formulation.

Consider a total of $M$ agents, each agent $i$ interacting with its individual environment equipped with a fixed policy $\mu_i$, which can be modeled as a Markov reward process (MRP). Suppose that all the MRPs have identical finite state and action spaces, though not necessarily the same transition matrices, reward functions,

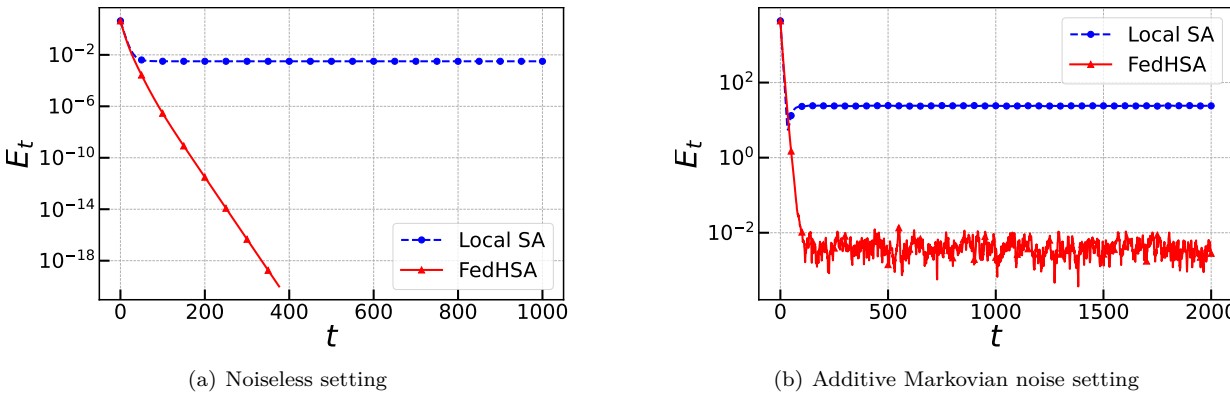

(a) Noiseless setting

(b) Additive Markovian noise setting

Figure 4: Comparison between `Local SA` and `FedHSA` with nonconvex objectives

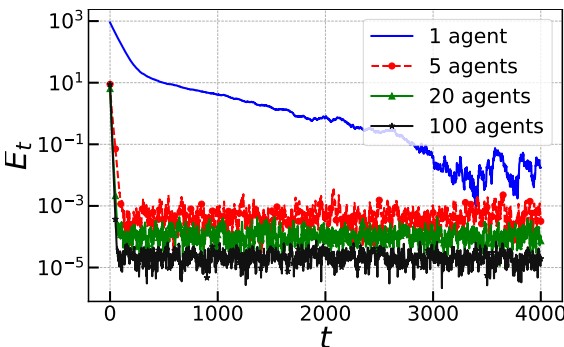

Figure 5: Comparison between different numbers of agents for the `FedHSA` algorithm with nonconvex objectives.

or discount factors. Specifically, the MRP of the $i$-th agent is denoted as $\mathcal{M}_i = (\mathcal{S}, \mathcal{A}, \mathcal{P}_i, R_i, \gamma_i)$, where $\mathcal{S}$ is the state space with cardinality $S$; $\mathcal{A}$ is the action space with cardinality $A$; $\mathcal{P}_i$ is the transition kernel dictated by the local policy $\mu_i$; $R_i$ is the reward function; and $\gamma_i$ is the discount factor with $0 < \gamma_i < 1$. We denote $r_i^{(t)}$ as the reward observed by agent $i$ at time-step $t$.

For self-containedness, we reiterate some basic concepts. For agent $i$ and the underlying MRP $\mathcal{M}_i$, the associated policy is $\mu_i$, and the value-function $V_i$ is defined as:

$$V_i(s) = \mathbb{E}\left[\sum_{t=0}^{\infty} \gamma_i^t r_i^{(t)} \mid s_i^{(t)} = s, \mu_i\right], \tag{98}$$

where $s_i^{(t)}$ is the state of agent $i$ at time-step $t$.

In many RL applications, the state and action spaces can be extremely large, making it impractical to store the value-function for each state $s$. To address this, feature matrices are often used to approximate the value-function. One common approach is the LFA framework:

$$\tilde{V}_i = \Phi_i \theta, \tag{99}$$

where $\tilde{V}_i$ is the approximated value-function for agent $i$ in vector form, $\Phi_i \in \mathbb{R}^{S \times d}$ is the feature matrix specific to agent $i$, consisting of $d$ linearly independent feature vectors $\{\phi_{i,k}\}_{k=1}^d$, and $\theta \in \mathbb{R}^d$ is the nominal parameter. Here, we make the general assumption that all $M$ agents do not necessarily use the same set of feature vectors (Doan, 2023).

The **goal** in this problem is for the agents to collectively find a parameter $\theta^\star$ such that it best approximates the value-functions across all agents, i.e.,

$$\bar{\Phi}\theta^\star \approx \frac{1}{M}\sum_{i=1}^{M} V_i, \tag{100}$$

where $\bar{\Phi} = \frac{1}{M}\sum_{i=1}^{M}\Phi_i$.

To this end, each agent updates its parameters by taking the direction of the negative gradient of the sample Bellman error $BE_i$ at observation $o_{i,t} := \{s_i^{(t)}, r_i^{(t)}, s_i^{(t+1)}\}$ w.r.t. parameter $\theta^{(t)}$ at time-step $t$, obtained via interacting with its own environment $\mathcal{M}_i$:

$$BE_i(\theta^{(t)}, o_{i,t}) := \frac{1}{2}\left(r_i^{(t)} + \gamma\phi_i^T(s_i^{(t+1)})\theta^{(t)} - \phi_i^T(s_i^{(t)})\theta^{(t)}\right)^2, \tag{101}$$

where $\phi_i(s_i^{(t)})$ is the feature vector of agent $i$ for state $s_i^{(t)}$. The server then collects the local parameters for aggregation, exactly as in the FL framework.

The negative gradient step of (101) is given by:

$$g_i(\theta^{(t)}, o_{i,t}) = \left(r_i^{(t)} + \gamma\phi_i^T(s_i^{(t+1)})\theta^{(t)} - \phi_i^T(s_i^{(t)})\theta^{(t)}\right)\phi_i(s_i^{(t)}). \tag{102}$$

Note that in (102), the gradient $g_i(\theta^{(t)}, o_{i,t})$ is *implicitly integrated with Markovian noise* since the states are sampled from the underlying Markov chain.

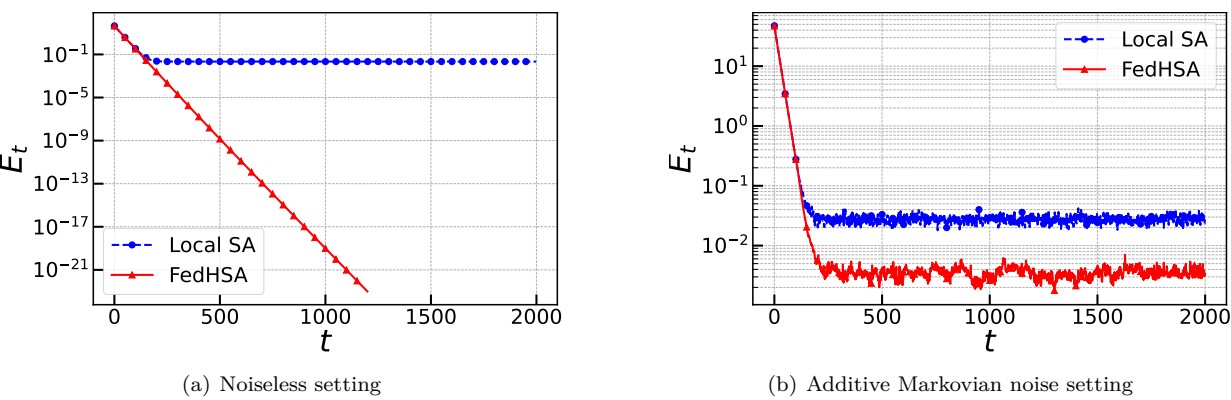

(a) Noiseless setting        (b) Additive Markovian noise setting

Figure 6: Comparison between `Local SA` and `FedHSA`

We also provide the expression for the expected negative gradient step, which one can interpret as the *noiseless gradient*:

$$\bar{g}_i(\theta^{(t)}) = \Phi_i^T D_i(T_i\Phi_i\theta^{(t)} - \Phi_i\theta^{(t)}), \tag{103}$$

where $D_i = \text{Diag}(\pi_i)$, $\pi_i$ is the stationary distribution of $\mathcal{P}_i$, and $T_i$ is the Bellman operator for agent $i$. We refer the reader to (Bhandari et al., 2018) for more details.

Our federated SA setup (1) precisely captures this setting. Specifically, $\bar{G}_i$ corresponds to the noiseless gradient operator $\bar{g}_i$ for agent $i \in [M]$, and $G_i$ corresponds to the noisy operator $g_i$ incorporated with Markovian noise.

To validate message (i), we compare our `FedHSA` algorithm with `Local SA` where there is no correction term in the local update of each agent. We consider a federated TD learning setting with LFA involving $M = 200$ agents, with each agent performing $H = 10$ local steps. Each MRP $\mathcal{M}_i$ has $S = 100$ states, and the rank of each feature matrix $\Phi_i$ is $d = 50$.

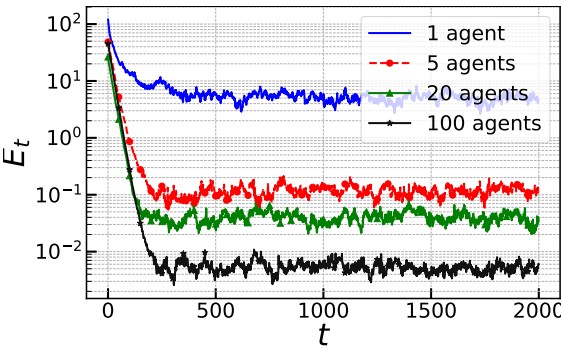

Figure 7: Comparison between different numbers of agents for the `FedHSA` algorithm with the federated TD setting

Figure 6 clearly demonstrates that `FedHSA` converges exponentially fast to $\theta^\star$ in the noiseless case and consistently outperforms `Local SA` both in the noiseless case and the one with Markovian noise.

We validate message (ii) by comparing `FedHSA` for different numbers of agents $M = 1, 5, 20, 100$. Figure 7 shows improved error bounds with an increase in the number of agents, substantiating the linear speedup effect.

### D.3  Federated Finite-Sum Minimization Problem with Quadratic Loss

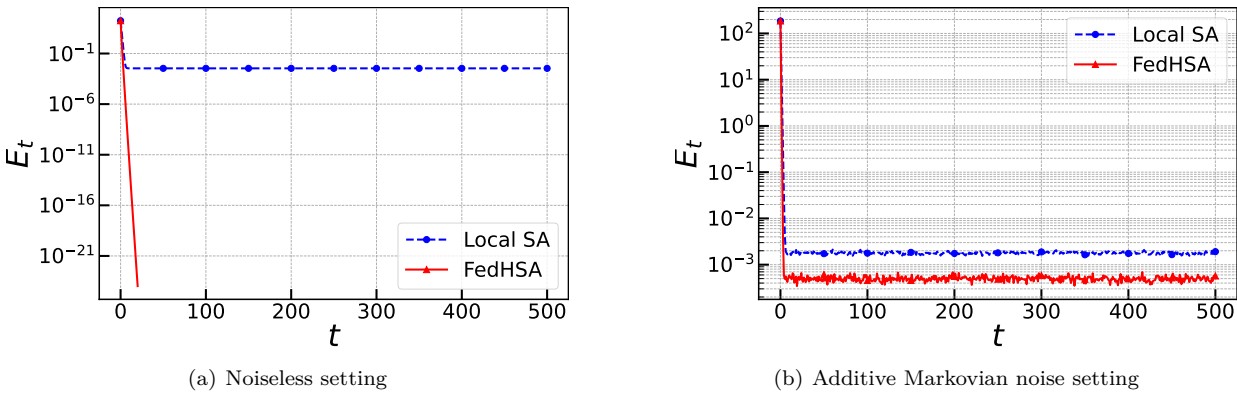

(a) Noiseless setting

(b) Additive Markovian noise setting

Figure 8: Comparison between `Local SA` and `FedHSA`

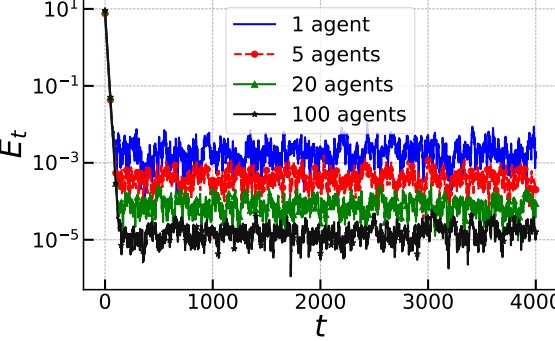

Figure 9: Comparison between different numbers of agents for the `FedHSA` algorithm with the finite-sum setting

In section D.1, Markovian noise was introduced by directly adding noise to the true local operator. Here, we adopt an alternative approach to incorporating Markovian noise: sampling data points from a Markov chain in a finite-sum setting. The detailed problem formulation is presented as follows.

We consider the same federated minimization problem as in (93), but with a different structure for the local loss functions $f_i$'s. Specifically, for each $i \in [M]$, we have

$$f_i(\theta) = \frac{1}{N} \sum_{j=1}^{N} f_{i,j}(\theta) = \frac{1}{N} \sum_{j=1}^{N} \left( \frac{1}{2} \theta^T A_{i,j} \theta - b_{i,j}^T \theta + c_{i,j} \right). \tag{104}$$

Here, $A_{i,j} \in \mathbb{R}^{d \times d}$ is a positive definite matrix, $b_{i,j} \in \mathbb{R}^d$ is a $d$-dimensional vector, and $c_{i,j} \in \mathbb{R}$ is a scalar for $j = 1, \cdots, N$. In essence, each local loss function is the average of $N$ quadratic loss functions, giving rise to the name "finite-sum setting."

We now describe the incorporation of Markovian noise. For intuition, consider the case where the noise is i.i.d., meaning each agent $i \in [M]$ selects one quadratic loss function $f_{i,j}$ *uniformly at random* and computes its gradient to determine the descent direction. Under Markovian noise, however, instead of selecting data samples uniformly at random, agent $i$ selects $f_{i,j}$ based on a discrete Markov chain $\mathcal{M}_i$. The states of the Markov chain $\mathcal{M}_i$ correspond to the indices $[N]$ of the loss functions $\{f_{i,j}\}_{j=1}^{N}$, and its stationary distribution $\mu_i$ is the uniform distribution over $[N]$. This ensures that in the limit, $\bar{G}_i$ is an unbiased estimator of the true operator $\bar{G}_i$, as $\bar{G}_i(\cdot) = \mathbb{E}_{o \sim \mu_i}[G_i(\cdot, o)]$ holds only when $\mu_i$ is uniform over $[N]$.

We set up the experiment with $M = 200$, $N = 2$, $d = 200$ and $\eta = 0.01$. Figure 8 validates message (i) exactly as in the previous experiments. Figure 9 compares the performance of FedHSA under varying numbers of agents, specifically $M = 1, 5, 20, 100$, with $N = 10$. Consistent with the observations from the previous experiments, the error floor decreases as the number of participating agents increases. This empirically validates our theoretical result that $d_T \leq \tilde{\mathcal{O}}(1/(MHT))$, demonstrating the benefits of collaborative speedup in reducing error.

