# OpenReview forum: "Achieving Tighter Finite-Time Rates for Heterogeneous Federated Stochastic Approximation under Markovian Sampling"
_TMLR — Accepted by TMLR_

### Review · Reviewer_Vbyh · 2025-10-29

**Summary Of Contributions:**

This paper makes the following contributions to the problem of federated stochastic approximation:

1. **Primary contribution:** This work proposes a novel algorithm that provably achieves both *exact convergence* and *linear sample-complexity speedup* under both *Markovian noise* and local *heterogeneity* for federated SA under *sparse communication constraints*. This provides a framework that could be adapted to both stochastic approximation and reinforcement learning.

2. **Technical contribution:** The proof of the main theorem (Theorem 2) is established through handling complex statistical correlations and drift terms caused by Markovian sampling, communication between agent and central algorithm and heterogeneity across agents. Notably, the authors avoids relying on a projection step (which ensures boundedness of the iterates) in the algorithm, and therefore have to deal with much harder bias control.

3. **Motivation and connection to various fields:** This work presents a thorough discussion of motivations from different existing fields, and provides a detailed discussion of related works, and provides numerical simulations of the proposed algorithm on those application domains. The authors provide a simple yet illustrative example of linear SA problem in which the naive local-update global-aggregation rule is biased under sparsity of communication, necessitating the design of a new algorithm.

**Additional Comments:**

Comments and questions:
1. Try not to use $\mathcal{O}$ notations when proving the intermediate results. This can sometimes hide problem-dependent parameters.

2. Omitting $o$ in $G$ terms can sometimes be confusing. For example, does $G_i(\bar{\theta}^{(t)})$ refer to $G_i(\bar{\theta}^{(t)},o_{i,\ell}^{(t)})$ or $G_i(\bar{\theta}^{(t)},o_{i,0}^{(t)})$?

3. In the linear speedup (Corollary 1), the choice of step size requires knowing $T$ in advance. Do the results generalize to using diminishing step sizes? For example, using $\alpha_t=\frac{\alpha_0}{t}$?

**Audience:**

Yes

**Audience Explanation:**

This paper studies the problem of federated stochastic approximation, which may be of interest for researchers in federated learning, stochastic approximation, optimization and RL.

**Broader Impact Concerns:**

I do not identify any broader impact concerns beyond the standard scope of theoretical research.

**Claims And Evidence:**

Yes

**Claims Explanation:**

The authors provide detailed proofs for their theoretical results and use numerical simulations to illustrate the effectiveness of their algorithm.

To make the entire proof of main result easier to understand, the authors first provide the proof of Theorem 1 in Appendix B through error decomposition and techniques from stochastic approximation. The authors provide intermediate results in Lemmas 1 and 2 to clarify the essential steps of the proof.

Building upon the previous part, the authors provide a detailed proof of the main result (Theorem 2) using a proof with similar outline to that of Theorem 1, but carefully deals with the noise dependence induced by Markovian sampling, which makes the proof much more complicated. The authors provide bounds for various different Markovian sampling errors in Lemmas 3, 5 and 6, which are clear and easy to follow.

All theorem and lemma statements look reasonable. I checked most of the proofs and didn't find obvious errors.

**Requested Changes:**

I do not find any critical need to change the paper, however, there are a few changes that I would recommend:

1. In the statement of Proposition 1, Schur-stable is not defined throughout the paper, and the symbol $e_t$ is not defined until later. These should be clearly stated within or before the statement. Also, the update rule in equation (8) does not explicitly state how $\theta_{i,0}^{(t)}$ is set, which should be made clear.

2. Both theorems take $\alpha_g=1$ by default and define $\alpha$ (the effective step size) to be dependent on $\alpha_g$. This would make it harder to interpret how the bound depends on different algorithm parameters. For example, the statement in (12) looks like the larger $H$ is (the less frequent agents communicate), the better bounds we have, which is clearly wrong. I would present the theorem without defining $\alpha$ and without the choice of $\alpha_g=1$. It's also beneficial to discuss the communication overhead directly for Theorems 1 and 2, instead of only for the linear speedup.

---

> ### Author Response · Authors · 2025-11-22
> **Comment to Reviewer Vbyh (Part I)**
>
> Dear Reviewer Vbyh,
>
> Thank you very much for your review of our paper. Based on your constructive suggestions (and those made by the other Reviewers), we have made careful revisions to our paper. Below we discuss the modifications made specifically to address your concerns. We use **R** to denote the reviews in the ``Requested Changes'' part and **A** to denote the reviews in the "Additional Comments'' part.
>
> **Response to R1.** We thank the Reviewer for this helpful comment. As per the Reviewer's request, we have now added a footnote to explain Schur-stability, and replaced $e_t$ directly with $\bar\theta^{(t)}-\theta^\star$ on Page 7 of the revised paper.
>
>    For your last concern, we did explicitly state that $\theta_{i,0}^{(t)}$ *is initialized from a common global parameter $\bar\theta^{(t)}$*, in the second line right after we introduced Eq.~(8). We have now explicitly mentioned in the paper that the initial global parameter $\bar\theta^{(0)}$ can be *arbitrary*.
>
> **Response to R2.** We thank the Reviewer for bringing up this issue.
>
>    Let us recall that the single-agent benchmark rate for the SA setting takes the following form:
>
> $$d_T\leq C_1\exp{(-\alpha C_2 T})+\alpha C_3 \sigma^2.$$
>
> Our goal for the federated heterogeneous SA setting is then naturally to align our result with this centralized convergence rate. As such, defining the “effective step-size” $\alpha$ facilitates a more direct comparison between our result and the centralized rate; otherwise, an additional $H$ term would appear in the exponential decaying term. That said, if the reviewer still prefers to replace $\alpha$ with $H\eta$, we are happy to do so.
>
> In the revised paper, we have now clearly explained that increasing $H$ does not need translate to an improvement in performance. We summarize this below.
>
> Next, **what is the relationship between communication overhead and linear speedup?** First, it should be clear that the notion of communication complexity makes no sense unless we associate it with a desired performance. For instance, our algorithm can involve zero communication, but then the performance guarantees would be vacuous. However, to achieve the desired linear speedup effect, there are certain conditions on the communication overhead, as we discuss below.
>
> **Discussion of Communication Complexity.** Recall that for the single-agent centralized setting, given $R$ samples, the standard SA scheme achieves a mean-square error (MSE) on the order of $\tilde{\mathcal{O}}(1/R)$. Now say each of the $M$ agents in our setup has access to $R$ samples. We ask: *How much communication is needed to achieve a mean-square error rate on the order of $\tilde{\mathcal{O}}(1/(MR))$, exhibiting the desired linear speedup effect?* This is a well-posed question, and our goal now is to specify the communication overhead as a function of $M$, $R$ and problem-dependent parameters. We do so below.
>
> Recall that $T$ is the number of communication rounds, and $H$ is the gap between communication rounds. As such, we have $R=TH$. Now recall that Theorem 2 of our paper presents the following result:
>
> $$d_T\leq \exp\left(-\frac{\mu}{4}\alpha T\right)\mathcal{O}(d_0+\sigma^2)+\mathcal{O}\left(\frac{\bar\tau \alpha L^2}{\mu MH(1-\rho)}+\frac{\alpha^2L^4}{\mu^2}\right)\sigma^2.$$
>
> Let us denote the three terms on the RHS as $(\*), (\*\*), (\*\*\*)$ respectively. This is essentially saying that the MSE after $T$ communication rounds is upper-bounded by an exponentially decaying term $(\*)$ plus a term $(\*\*)$ scaled down by the number of agents $M$, and another term $(\*\*\*)$ that is higher-order in $\alpha$, but not scaled down by $M$. In order to achieve linear speedup, it is then straightforward that the term $(\*\*\*)$ should be dominated by $(\*\*)$, imposing the following requirement on the maximum allowable gap $H$ between communication rounds (to preserve the linear speedup effect):
> $$H\leq \mu\bar\tau/(\alpha L^2M(1-\rho)).$$
>
> Substituting the choice of $\alpha=H\alpha_g\eta=4\log(MHT)/(\mu T)$ in Corollary 1, we then have
> $$H\leq \frac{\mu^2\bar\tau T}{4L^2 M (1-\rho) \log (MHT)}.$$ Finally, using $T= R/H$ in the above display yields the requirement:
> $$H\leq \frac{\mu}{2L}\sqrt{\frac{\bar\tau R}{M(1-\rho) \log(MR)}},$$ which is essentially $\tilde{\mathcal{O}}(\sqrt{R/M})$ (as stated in the initial version of our paper, albeit without the other problem dependent constants).
>
> Concretely, given $M$ and $R$, one can then set $H$ to be
> $$ \boxed{H = \Biggl\lfloor\frac{\mu}{2L}\sqrt{\frac{\bar\tau R}{M(1-\rho) \log(MR)}}\Biggr\rfloor.}   $$
>
> Finally, we would like to emphasize that under this communication constraint, **a larger $H$ does not imply a better performance**. To see why, recall that the achieved rate is $\tilde{\mathcal{O}}(1/(MHT))$. Given the number of samples $R$ at each agent is fixed, increasing $H$ does not impact the final bound since $1/(MHT)=1/(MR)$, which is independent of $H$. We hope that this addresses your concern.

---

> > ### Author Response · Authors · 2025-11-22
> > **Comment to Reviewer Vbyh (Part II)**
> >
> > **Response to A1.** We thank the Reviewer for their concern. The reason that we resort to big-O notation is to hide messy universal constants, which might be a digression from the numerous other technical issues that are worth highlighting such as controlling the drift effect, handling the temporal correlation between data, etc. That said, in light of the Reviewer's pertinent comment, we have now double-checked to make sure that we only hide universal constants in the big-O notation in our derivations, and any problem-dependent parameters such as $L,\mu, \sigma$, etc., are explicitly stated in the presentation of the theorems and proofs.
> >
> > **Response to A2.** We sincerely thank the Reviewer for this pertinent suggestion, and completely agree with their assessment. In the revised paper, whenever we have objects of the form $G_i(\bar\theta^{(t)}, o_{i,0}^{(t)})$, where the first argument is a global model at the server, we have now specified the observation. However, aligning with our initial motivation of keeping the technical exposition as clean as possible, for objects of the form $G_i(\theta_{i,\ell}^{(t)}, o_{i,\ell}^{(t)})$, we have taken the liberty of using the shorthand $G_i(\theta_{i,\ell}^{(t)})$. That said, to avoid any confusion, in the revised paper, we have now explicitly mentioned, both in Section 5 and in Appendix B, that we are using this shorthand only when the indices associated with the model (namely $i, \ell, t$) \emph{exactly} match the corresponding indices of the observation variable. We hope the Reviewer finds this acceptable.
> >
> > **Response to A3.** We thank the Reviewer for bringing this issue up. In short, yes, we do believe that our analysis can be extended using standard tools to accommodate time-varying step-sizes. Before we explain how, let us quickly justify why we chose constant step-sizes. As one might expect, a constant step-size admits a much cleaner analysis than a diminishing step-size, which is exactly why we adopt it, to avoid distraction from the main message that we are trying to convey, i.e., we are able to fill the gap of achieving linear speedup with no heterogeneity bias under Markovian data.
> >
> > Next, we note that non-diminishing constant step-sizes are used readily in the literature on SA/RL even in simpler single-agent settings, precisely with the same motivation (as us) of providing a simple, clean analysis; see, for instance, Refs [1] and [2] below. In fact, note that in Theorem 2 (part (a)) of Ref [1], the constant step-size requires knowledge of the horizon $R$, much like us. Other than picking up some additional log factors, this causes no degradation in the overall learning rate.
> >
> > Having explained why we chose constant step-sizes, and the fact that such a choice is common in SA/RL, we now mention two pathways for getting rid of the knowledge of $R$. One direct option is to use a diminishing step-size schedule, exactly like the one the Reviewer suggested. For this setting, we can borrow well understood proof techniques; for instance, the one used to prove item (c) in Theorem 2 of Ref [1]. The other perhaps easier to adapt technique is the "doubling trick" in the online learning literature. The basic idea is as follows. We first set the horizon length to say $R'$, decide the step-size as per $R'$ (like in our paper), and run the algorithm for $R'-1$ iterations. Then, we set $R' \leftarrow 2 R'$, reset the step-size, and repeat. This degrades the performance of the algorithm by at most a constant factor. To see why, note that due to doubling, the $k$-th epoch is of duration $2^k$, if $R'=1$. Suppose the true unknown horizon length is $R$. Let $J$ be the last epoch before the horizon length is exceeded, i.e., $2^{J+1} > R$, implying $2^J > R/2$. But since the duration of the last epoch is precisely $2^J$, we conclude that the last epoch *contains at least half the total number of samples $R$.* Consequently, long story short, the price of not knowing $R$ a priori is a degradation by a constant factor that is no worse than $2$.
> >
> >  We have added a remark now in Section 4, summarizing the above discussion.
> >
> > [1] A Finite time analysis of temporal difference learning with linear function approximation, COLT 2018.
> >
> > [2] Finite-Time Error Bounds For Linear Stochastic Approximation and TD Learning, Srikant and Ying, COLT 2019.
> >
> > We hope the above discussion addresses your concerns, and we are happy to take further questions. Thank you again for your constructive comments.

---

### Review · Reviewer_uWrU · 2025-11-01

**Summary Of Contributions:**

This paper presents FedHSA, a novel federated stochastic approximation (FSA) algorithm designed to address heterogeneous federated learning problems under Markovian data. FedHSA incorporates a drift correction term in local updates to handle heterogeneity in agents' local operators. The paper establishes finite-time convergence guarantees showing convergence to the correct global optimum without heterogeneity-induced bias, achieving linear speedup with regard to the number of agents.

Strengths
1. The work develops and analyzes FSA under more generalized setting (Marokvian sampling and non-linear operators) compared to the previous work [1] mainly considering linear operators and providing limited analysis on Markovian sampling.

Weaknesses
1. Core algorithm design/analysis overlaps with the previous work [1]: [1] studied FSA under heterogeneity and proposed a similar bias-corrected (scaffold-style) algorithm that removes heterogeneity-induced bias and achieves linear speedup.
2. Although the paper differentiates its contribution from [1] by claiming to consider more general non-linear operators, it is unclear how technically challenging or novel this extension is.
3. Similarly, analysis of federated SA under Markovian sampling is common in the federated RL/SA literature, and it is unclear how challenging or novel it is to additionally consider Markovian sampling in this problem.
4. Communication analysis is not that informative. Many federated RL algorithm [2,3] provides communication sample complexity in terms of problem specific parameters (horizon length or discount factors, state-action size, stationary distribution). However, this work does not characterize the communication overhead clearly and present it mainly in terms of H (local updates), R (total samples), which seems superficial.

[1] Paul Mangold, Sergey Samsonov, Safwan Labbi, Ilya Levin, Reda Alami, Alexey Naumov, and Eric Moulines. Scafflsa: Taming heterogeneity in federated linear stochastic approximation and td learning. Advances in Neural Information Processing Systems, 2024.
[2] Sudeep Salgia and Yuejie Chi. The sample-communication complexity trade-off in federated q-learning. In Advances in Neural Information Processing Systems, 2024.
[3] Sajad Khodadadian, Pranay Sharma, Gauri Joshi, and Siva Theja Maguluri. Federated reinforcement learning: Linear speedup under markovian sampling. In International Conference on Machine Learning, pp. 10997–11057. PMLR, 2022.

**Audience:**

Yes

**Audience Explanation:**

The paper provides a principled design and analysis of federated stochastic approximation under heterogeneity for a realistic setup (nonlinear operators and Markovian noise), offering broad applicability and theoretical advancement.

**Broader Impact Concerns:**

No significant concerns. The work mainly focuses on theoretical aspects of the problem.

**Claims And Evidence:**

Yes

**Claims Explanation:**

The paper provides theoretical analysis with detailed proofs establishing convergence rates.

**Requested Changes:**

I suggest the authors to address the following points, which are critical for clarifying the paper's precise technical contributions and novelty.

1. Clarify the technical challenge and novelty of non-linear operators: The paper differentiates its contribution from [1] by claiming the more general consideration of non-linear operators. However, it is currently unclear how technically challenging or novel this extension is. The authors must explicitly detail the specific technical hurdles introduced by moving from linear to non-linear operators and clearly articulate how their analytical techniques overcome these challenges in a novel way.

2. Justify the contribution regarding Markovian sampling: The analysis of federated SA under Markovian sampling is relatively common in the federated RL/SA literature. The paper would be significantly strengthened if the authors could more clearly explain the novelty of their analysis in this context. Please clarify what unique challenges arise from additionally considering Markovian sampling in this specific problem (e.g., in combination with heterogeneous non-linear operators) and how the proposed analysis advances beyond existing work in the federated RL/SA domain.

3. Provide clarifications on why the extension to no-linear operators is important.

4. Provide more detailed characterization of communication complexity in terms of problem specific parameters.

---

> ### Author Response · Authors · 2025-11-21
> **Comment to Reviewer uWrW (Part I)**
>
> Dear Reviewer uWrU,
>
> Thank you very much for your review of our paper. Based on your constructive suggestions (and those made by the other reviewers), we have made careful revisions to our paper. Below we discuss the modifications made specifically to address your concerns. We use **S** to denote the reviews in the "Summary Of Contributions'' part, and **R** to denote the reviews in the "Requested Changes'' part.
>
> **Response to S1.** We humbly disagree with the assessment that the core contribution (in terms of algorithm design and analysis) overlaps with the recent work of Mangold et al. Before we elaborate on the key analytical differences in response to **R1**, it is worth emphasizing that the idea of using bias-correction is not unique to the work of Mangold et al. either; in fact, such an idea has been used for more than a decade now under different names and contexts (e.g., variance reduction, gradient-tracking, control variates etc.) in the learning and control communities. As such, *if using bias-correction in our work is considered to be a limitation, then such a limitation also applies to the work of Mangold et al.*
>
> That said, in our opinion, the existence of such ideas in the past neither diminishes the core contribution of Mangold et al., nor that of our work. The reason is as follows. Bias-correction ideas have been used in the past *almost exclusively in the context of optimization.* We clearly acknowledged this fact in Remark 1 of our initial submission. As we also explain in that remark, as soon as one looks at stochastic approximation (SA) problems beyond optimization (e.g., TD and Q-learning with function approximation), *our bias-corrected algorithm FedHSA represents update rules that have not been analyzed before in federated and reinforcement learning.* Our core contribution lies in *revealing, through rigorous theory backed up by suitable experiments, that the scope of the bias-correction idea extends well beyond optimization to general non-linear SA schemes under Markov sampling.* We believe this is a significant finding, not subsumed by prior work, that further expands the scope of the bias-correction idea (beyond what was known). In particular, while the very nice paper of Mangold et al. (which we did discuss in our initial submission) represents an initial effort in this direction, they analyze their bias-correction scheme only for linear operators and without considering Markov sampling. As we explain below in the rest of our rebuttal, each of these extensions require non-trivial work.
>
> **Response to S2.** Please see our response to **R2**, where we clearly explain that our analysis for nonlinear operators departs fundamentally *right from the first step* compared to that for linear operators.
>
> **Response to S3.** Please see our response to **R3**, where we provide a detailed description of why the current analyses of Markov sampling in the context of both single-agent and federated SA are inadequate for arriving at our results.
>
> **Response to S4.** Please see our response to **R4**, where we clearly provided an upper-bound of the communication overhead explicitly characterizing problem-specific parameters such as $L,\mu,\bar\tau$, etc.

---

> > ### Author Response · Authors · 2025-11-21
> > **Comment to Reviewer uWrW (Part II)**
> >
> > **Response to R1.** On page 5 of our initial submission,  we did provide a reasonably detailed description of the key differences between our paper and that of Mangold et al., a work that we became aware of while preparing the initial draft of our paper. We thank the Reviewer for providing us an opportunity to further elaborate on the major points of difference.
> >
> > Next, let us expand on the technical differences and hurdles in our work relative to that of Mangold.
> >
> > **What breaks down from linear to nonlinear?** To appreciate the main difference, let us first note that the derivations in Mangold et al. are rooted in an analysis framework for linear stochastic approximation in Ref [14] of their paper (Ref [R1] below). The basic starting point in such papers is writing down a recursion of the form (see Section 4 of Mangold et al.):
> >
> > $$
> > \theta_{t+1} - \theta^* = (I-\eta A(Z_{t+1})) (\theta_{t} - \theta^*) + \eta \varepsilon(Z_{t+1}),
> > $$
> >
> > where $\theta_t$ is the iterate at time $t$, $\theta^*$ is the optimal point, $\eta$ is the step-size, $Z_t$ is the noise at time $t$, $\varepsilon(Z_t)$ is a function of $Z_t$, and $A(Z_{t})$ is a random matrix that defines the noisy linear stochastic operator at time $t$. The dynamics in the above equation represents that of a linear time-varying (LTV) dynamical system with an additive input. As is usually done in the analysis of such systems, the next step is to unroll the recursion above which leads to a product of random matrices. The rest of the analysis in Mangold et al. (and Ref [R1] below) focuses on analyzing such a random matrix product.
> >
> > [R1] Finite-time High-probability Bounds for Polyak-Ruppert Averaged Iterates of Linear Stochastic Approximation, Durmus et al., arXiv 2023.
> >
> > *Our analysis departs **fundamentally** from that in Mangold et al. right from the first step*, since we cannot even write down an LTV dynamics of the form as the above equation; thus, there is no occasion for us to analyze an unrolled recursion involving products of random matrices. **This, in itself, should suffice to clarify that our approach cannot (and does not), in fact, build on that of Mangold et al., and the predecessor paper Ref [R1].**
> >
> > **Technical Hurdles in our Analysis.** As mentioned above, the nonlinearity of the operators in our setting precludes the approach in Mangold where one studies an unrolled linear recursion involving matrix products. Instead, the starting point of our analysis involves setting up a scalar one-step recursion for the mean-square error (MSE) in Lemma 4 of our paper; this is done by carefully leveraging regularity properties (such as Lipschitzness and strong-monotonicity) of the underlying operator. Roughly speaking, the right-hand-side (R.H.S.) of this recursion involves four terms:
> >
> > $$
> > \textrm{MSE} \leq \textrm{Contractive term} \hspace{1mm} + \textrm{Drift term} \hspace{1mm} + \textrm{Variance-Reduction Term} + \hspace{1mm} \textrm{Markovian Bias},
> > $$
> >
> > namely, a "good" term that leads to a contraction in the MSE, a "drift" term due to heterogeneity, a "noise variance" term that gets scaled by the number of agents $M$, and a bias term due to Markov sampling. Establishing this result is quite non-trivial since it involves proving a "variance-reduction" effect under Markov sampling; once again, we leverage Lipschitzness of the nonlinear operator in proving this variance reduction effect in Lemma 3 of Appendix C. The next major technical hurdle of our work involves controlling the Markovian bias term in the above inequality. Crucially, *such a challenge is not encountered in Mangold et al., since their work does not consider Markov sampling for their bias-corrected algorithm.* Moreover, as we explain in detail in Section 5 of our paper (titled "Challenges and Proof Sketch"), handling this *Markovian bias term requires  new techniques relative to both single-agent and multi-agent (federated) RL and stochastic approximation*. We elaborate more on the difficulties that arise from Markov sampling in response to your requested change/comment on Markov sampling.
> >
> > In Section 5 of our revised paper, we have now briefly summarized some of the above points.
> >
> > **Summary.** In short, it is quite unclear (to us at least) whether the approach in Mangold et al. that involves analyzing products of random matrices can be extended beyond linear operators. In contrast, we believe that our general framework that exploits regularity properties of the underlying operator lends itself to the analysis of a much broader class of nonlinear operators. As such, we conjecture that our approach will find applicability in the study of complex stochastic approximation algorithms. We hope this clarifies your concern on the distinction relative to the work of Mangold et al.

---

> ### Author Response · Authors · 2025-11-21
> **Comment to Reviewer uWrW (Part III)**
>
> **Response to R2.** As far as we are aware, only a handful of papers have considered Markov sampling in the context of federated RL/SA. To highlight the novelty of our technical analysis, in what follows, we provide a detailed explanation of the specific issues that preclude direct extension of the techniques in these papers to suit our needs. We also explain why existing approaches in single-agent RL/SA are also inapplicable for our setting.
>
> 1. *Comparison with homogeneous federated RL/SA papers.* The first paper to analyze the effects of Markov sampling in FRL was Ref [R2] below, followed by Ref [R3]. In [R2], the authors consider a homogeneous setting, where all agents' operators have the same root, and as a result, a standard FedAvg-style local update rule suffices. Even so, the analysis in this paper (that spans for nearly 40 pages) is quite involved and proceeds by using the concept of Generalized Moreau Envelopes. It is not at all apparent whether such a proof technique could be extended to handle the bias-corrected local update scheme in our paper, subject to additional drift effects introduced by heterogeneity in agents' operators. Our proof, in contrast, does not require going through the framework of Generalized Moreau Envelopes, and analyzes a different local update scheme altogether (relative to that in [R2] and [R3]).
>
>    While the analysis in [R3] sharpens some of the bounds in [R2], the results are derived only for tabular Q-learning *without any function approximation.* Even in a single-agent setting, it is well understood that an extension from the tabular setting to the function-approximation setting is highly non-trivial, and several papers have focused precisely on this extension. In particular, for the tabular setting, a relatively straightforward argument (requiring no more than a couple of lines) suffices to establish that the Q-table iterates remain uniformly bounded throughout the course of the algorithm; see, for instance, Ref [R4] below. This fact is leveraged throughout the analysis. Unfortunately, under function approximation, it is no longer the case (in general) that the iterates are uniformly bounded deterministically (with a non-vacuous upper bound). This complicates the analysis significantly since each of the terms in the R.H.S. of our main recursion in the MSE bound above are *iterate-dependent* and cannot just be replaced by uniformly bounded perturbations (as is possible in a tabular setting).
>
>    [R2] Federated reinforcement learning: Linear speedup under Markovian sampling, Khodadadian et al., ICML 2022.
>
>    [R3] The blessing of heterogeneity in federated Q-learning: Linear speedup and beyond, Woo et al., ICML, 2023.
>
>    [R4] Finite-Time Analysis of Asynchronous Stochastic Approximation and Q-Learning, Qu and Wierman, COLT 2020.
>
> 2. *Comparison with heterogeneous federated RL/SA papers.* The only two papers we know of that consider finite-time analysis of heterogeneous FRL algorithms under function approximation are [R5] and [R6] below. Our analysis departs from both these papers in two main ways. First, as we explain in our paper, [R5] and [R6] study FedAvg-style algorithms where each agent uses just its own local operator to update model parameters between communication rounds. As we show in Proposition 1, such algorithms converge to incorrect fixed points. In contrast, we study a different algorithm altogether, the dynamics of which are not the same as the ones in [R5] and [R6].
>
>    On a more technical note, both [R5] and [R6] consider a projection step in their algorithm to ensure that the iterates remain uniformly bounded. This considerably simplifies the analysis since the Markovian bias term is *iterate-dependent*, and projection ensures that such a term is essentially a uniformly bounded perturbation. In sharp contrast, we make no such assumption of a projection step, and, as such, we do not have the luxury of a uniformly bounded Markovian bias; single-agent papers do not need to show such a result. Thus, a much finer analysis relative to [R5] and [R6] is needed to control each of the iterate-dependent "error" terms on the R.H.S. of the MSE bound above. In addition, our proof also requires establishing that no heterogeneity-induced bias shows up in our final bound (unlike [R5] and [R6]).
>
>    [R5] Finite-time analysis of on-policy heterogeneous federated reinforcement learning, Zhang et al., ICLR 2024.
>
>    [R6] Federated temporal difference learning with linear function approximation under environmental heterogeneity, Wang et al., TMLR 2024.

---

> > ### Author Response · Authors · 2025-11-21
> > **Comment to Reviewer uWrW (Part IV)**
> >
> > 3. *Comparison with single-agent RL/SA papers.* Although Markovian sampling has been studied in single-agent RL/SA, there are considerable challenges in extending such results to our specific setting. Other than the usual issues that arise in FL (communication and heterogeneity-induced drifts), our algorithm induces complex spatial and temporal correlations absent in the single-agent setting; please see Figure 1 of our paper where we illustrate such correlations. Moreover, as we explain in Section 5 of our paper, relative to single-agent RL papers, we need to establish a finer result involving the linear speedup effect. To achieve this, we need to show that the Markovian bias term can be upper-bounded by terms that are either higher-order in the step-size or scaled down by the number of agents. **This is the hardest part of our analysis, and to our knowledge, no other paper in FRL has been able to establish such a result without making simplifying assumptions of projection steps.**
> >
> > 4. *Comparison with the Mangold et al. paper.* In our response to **R1**, we have already explained why the nonlinearity of operators in our setting precludes the approach pursued in the work of Mangold et al. There is one other crucial difference with this work. In Mangold et al., the bias-corrected algorithm used for linear SA is studied only under i.i.d. sampling; **thus, our work is the first to analyze the effects of control variates under Markov sampling.** For vanilla SA, the approach in Mangold et al. to control Markov sampling is the "blocking technique", where one simply discards several data points by sub-sampling. By a coupling argument, the analysis then boils down to that for i.i.d. sampling. In sharp contrast, our approach does not discard any data points, making the Markovian analysis significantly more challenging since the samples used to generate iterates are temporally correlated.
> >
> > **Summary.** To sum up, while we do not contest the fact that Markovian sampling has appeared in prior SA/RL work, our comprehensive discussion above highlights that **no prior analysis is enough to subsume the specific challenges that arise in our setting** from a combination of function approximation, nonlinear operators with heterogeneous roots, the lack of projection steps, and the need to establish a linear speedup effect despite complex correlations. As such, we believe that the analysis framework developed in our paper can serve as a template for reasoning about more involved complex SA/RL schemes in the future.
> >
> > In the revised paper, we have added some more discussions on the above challenges in Section 5.
> >
> > **Response to R3.** Other than simple quadratic optimization problems in supervised learning (like linear regression) and policy evaluation algorithms in RL like Temporal Difference Learning under linear function approximation, we cannot think of examples of *linear* SA. In the context of optimization, as soon as the gradient is not a linear function of the iterate (consider, for instance, logistic regression), we will end up with nonlinear operators. In the context of RL, even for tabular Q-learning, one ends up with a nonlinear operator since the Bellman optimality operator features a `max' term. Thus, *even for the simplest RL control problem, one has to study nonlinear operators.* We should also note that complex neural function approximators used in practice for deep RL will naturally induce nonlinear operators.

---

> > > ### Author Response · Authors · 2025-11-22
> > > **Comment to Reviewer uWrW (Part V)**
> > >
> > > As per the Reviewer's request, in the discussion following the statement of our main results in Section 4, we have now explicitly characterized the communication complexity of our algorithm as a function of relevant problem-dependent parameters, such as $L,\mu,\bar\tau,$ etc. For the Reviewer's convenience, we reproduce that discussion here.
> > >
> > > *Discussion of Communication Complexity.* First, it should be clear that the notion of communication complexity makes no sense unless we associate it with a desired performance. For instance, our algorithm can involve zero communication, but then the performance guarantees would be vacuous. Keeping this in mind, recall that for the single-agent centralized setting, given $R$ samples, the standard SA scheme achieves a mean-square error (MSE) on the order of $\tilde{\mathcal{O}}(1/R)$. Now say each of the $M$ agents in our setup has access to precisely $R$ samples. A natural question then is: *How much communication is needed to achieve a mean-square error rate on the order of $\tilde{\mathcal{O}}(1/(MR))$, exhibiting the desired linear speedup effect?* This is a well-posed question, and our goal now is to specify the communication overhead as a function of $M$, $R$ and problem-dependent parameters. We do so below.
> > >
> > > In our notation, recall that $T$ is the number of communication rounds, and $H$ is the gap between communication rounds. As such, we have $R=TH$. Now recall that Theorem 2 of our paper presents the following result:
> > >
> > > $$d_T\leq \exp\left(-\frac{\mu}{4}\alpha T\right)\mathcal{O}(d_0+\sigma^2)+\mathcal{O}\left(\frac{\bar\tau \alpha L^2}{\mu MH(1-\rho)}+\frac{\alpha^2L^4}{\mu^2}\right)\sigma^2.$$
> > >
> > > Let us denote the three terms on the RHS as $(\*), (\*\*), (\*\*\*)$ respectively. This is essentially saying that the MSE after $T$ communication rounds is upper-bounded by an exponentially decaying term $(\*)$ plus a term $(\*\*)$ scaled down by the number of agents $M$, and another term $(\*\*\*)$ that is higher-order in $\alpha$, but not scaled down by $M$. In order to achieve linear speedup, it is then straightforward that the term $(\*\*\*)$ should be dominated by $(\*\*)$, imposing the following requirement on the maximum allowable gap $H$ between communication rounds (to preserve the linear speedup effect):
> > > $$H\leq \mu\bar\tau/(\alpha L^2M(1-\rho)).$$
> > >
> > > Substituting the choice of $\alpha=H\alpha_g\eta=4\log(MHT)/(\mu T)$ in Corollary 1, we then have
> > > $$H\leq \frac{\mu^2\bar\tau T}{4L^2 M (1-\rho) \log (MHT)}.$$ Finally, using $T= R/H$ in the above display yields the requirement:
> > > $$H\leq \frac{\mu}{2L}\sqrt{\frac{\bar\tau R}{M(1-\rho) \log(MR)}},$$ which is essentially $\tilde{\mathcal{O}}(\sqrt{R/M})$ (as stated in the initial version of our paper, albeit without the other problem dependent constants).
> > >
> > > Concretely, given $M$ and $R$, one can then set $H$ to be
> > > $$ \boxed{H = \Biggl\lfloor\frac{\mu}{2L}\sqrt{\frac{\bar\tau R}{M(1-\rho) \log(MR)}}\Biggr\rfloor.}   $$
> > >
> > > With $H$ as above, the communication overhead is $T = R/H$, which is on the order of $\sqrt{MR}$. We hope this clarifies your concern.
> > >
> > > We hope the above discussion resolves your concerns, and we are happy to take any further questions. Thank you again for your constructive comments.

---

### Review · Reviewer_kPGF · 2025-11-08

**Summary Of Contributions:**

This paper aims to provide an enhanced finite-time convergence analysis for heterogeneous federated stochastic approximation where the data and observations of each client are generated from an ergodic Markov chain rather than independent and identically distributed (i.i.d.) from a stationary distribution. This captures broader interesting problems in dynamic environments like wireless systems with channels changing over time.  However, this also induces some additional challenges in analysis, e.g., leading to additional bias terms such as the one on page 11. The authors provide the first finite-time convergence results for federated optimization under such Markov data, particularly without relying on any projection step or restrictive assumptions.

The developed method considers a general SA setup. Therefore, the method could be applied to broader problems such as multi-agent RL problems. In this setup, there have been quite a few works focusing on developing finite-time results for federated algorithms under different data or setups. In particular, the authors clearly compare their results to these results. For example, their results allow for heterogeneous MDPs across different clients, and compared to existing results such as those in Zhang et al., 2024, their results suffer from a heterogeneity-induced bias term. In addition, a linear speedup result is shown as well. Table 1 is a good summary for readers to quickly compare different related works.

Some methods turn out to be interesting, if more details are given. For example, the authors first show that only the local updates with multiple agents and periodic communication can induce non-vanishing error terms proportional to the constant stepsize. The authors propose to add a correction term relying on the stored aggregation at the beginning of each communication round. It would be even better if the authors can clarify if this is a widely used approach in i.i.d. data case, or it is specially designed under the Markovian data.

**Additional Comments:**

RL experiments should be provided.

**Audience:**

Yes

**Audience Explanation:**

This paper provides the first finite-time convergence results for heterogenous federated optimization under Markovian data, without any projection step, while achieving a linear speedup result. It contributes new result in this setup. The analysis can be of interest to multi-agent and federated RL communities.

**Broader Impact Concerns:**

This is a theoretical paper and there are no broader impact concerns.

**Claims And Evidence:**

Yes

**Claims Explanation:**

This paper is very well written and clearly discuss all relevant works along this direction. I like the comparison to existing results in the introduction, providing readers with sufficient evidence to understand the main contributions and the theoretical advantages over existing literature. In addition, Proposition 1 motivates the algorithmic designs and the proof sketch part helps me to understand the main technical contributions.

**Requested Changes:**

1.	Proposition 1 shows that the error is proportional to the stepsize \eta. I am wondering if this stepsize decreases with time, will it lead to a  worse rates or without linear speedup? The authors may want to provide a formal comparison. In addition, typically, for federated optimization with i.i.d. data, with proper aggregation and stepsizes (not necessarily decreasing with time, i.e., can be small constants), that client drift will not happen. I am wonder if this is the special challenge of SA or Markovian data case?

2.	Assumption 2 seems a little bit strong as it corresponds to the strong-convexity in optimization perspective. This assumption seems quite necessary when proving the Claim on page 11, where the authors show that the bias can be resolved without any projection step. I am wondering if the results also hold for general nonconvex case? If not, what is the challenge?

3.	Since the authors propose new algorithms, I believe some validation experiments are needed. For example, it would be good to validate the results such as linear speedup, the correction term, convergence rate etc.

---

> ### Author Response · Authors · 2025-11-21
> **Comment to Reviewer kPGF (Part I)**
>
> Dear Reviewer kPGF,
>
> Thank you very much for your review of our paper. Based on your constructive suggestions (and those made by the other reviewers), we have made careful revisions to our paper. Below we discuss the modifications made specifically to address your concerns. We use **R** to denote the reviews in the "Requested Changes'' part.
>
> **Response to R1.** We thank the reviewer for their careful inspection of this proposition.
>
> Firstly, we clarify that the setting considered in Proposition 1 is the *noiseless* setting, where each agent $i$ has access to its true operator $\bar G_i$. With Proposition 1, our goal was to reveal that even in this simple noiseless setting, existing local SA algorithms fail to match the rates achievable in the single-agent (centralized) case.
>
> To appreciate this point, recall from equation (3) that such a single-agent rate takes the following form after $T$ iterations:
>
> $$ d_T \leq
> C_1 \exp(-\alpha C_2 T)
> +
> \alpha C_3 \sigma^2 $$
>
> where $d_T$ is the mean-square error after $T$ iterations, $\alpha$ is a constant step-size, and $\sigma^2$ captures the effect of noise. Note that if we set $\sigma^2 = 0$ in the benchmark bound above, one can achieve *exact convergence to the desired point* $\theta^\*$ *exponentially fast*.
>
> In sharp contrast, with existing federated local SA algorithms, even in a noiseless setting, Proposition 1 reveals that the convergence is to a ball of radius $\eta$ if one uses a constant step-size $\eta$. As such, while it is indeed true that if we employ a diminishing step-size, convergence to the exact optimum $\theta^\star$ is guaranteed, **this will come at the cost of a slower, sublinear convergence rate** (i.e., at a worse rate as the Reviewer suspects), which is clearly not desirable. A few crucial comments are in order about this result.
>
> 1. The phenomenon observed in Proposition 1 has *nothing to do with noise at all*, i.e., whether the noise is i.i.d. or Markovian is immaterial as far as the main message of Proposition 1 is concerned.
>
> 2. The key issue at play in Proposition 1 is that the agents' operators have distinct roots, creating the "client-drift" effect under intermittent communication.
>
> 3. Heterogeneous federated optimization can be subsumed as a special case of Proposition 1. In particular, Proposition 1 is stated for linear stochastic approximation algorithms (LSA), and quadratic optimization is a special case of LSA. An immediate corollary of this point is that even in federated optimization (without noise), if one uses a vanilla local update algorithm (with no bias-correction), then a constant step-size will lead to convergence to an incorrect fixed point, i.e., *the client-drift effect cannot be eliminated*. Proposition 1 simply generalizes this observation to more general SA schemes.
>
> To sum up, Proposition 1 has nothing to do with Markovian data, and only serves to highlight that there is a "speed-accuracy" trade-off at play when one uses vanilla local SA algorithms: a constant step-size guarantees exponentially fast convergence, but to an incorrect point; a diminishing step-size guarantees exact convergence to the desired point, but at a slower sub-linear rate. Notably, our proposed algorithm FedHSA is able to overcome this ``speed-accuracy" tradeoff: setting the noise effect $\sigma^2$ to $0$ in Theorem 1 reveals that in a noiseless setting, FedHSA guarantees exponentially fast convergence to the desired point, matching the centralized guarantee.
>
> In the revised paper, we have further clarified these points in the discussion following Proposition 1.

---

> ### Author Response · Authors · 2025-11-21
> **Comment to Reviewer kPGF (Part II)**
>
> **Response to R2.** We thank the reviewer for their constructive comment.
>
> **Why Assumption 2?** First, let us explain our rationale for working under  Assumption 2. Roughly speaking, Assumption 2 captures a contractive property of the underlying operator, and it is precisely under this assumption that one achieves the benchmark single-agent rate. As we explain in quite some detail in Section 1.1. of our paper, even under this assumption, there is a clear gap between the single-agent results and their counterparts in the vast literature of federated heterogeneous SA. Specifically, prior to our work, no other paper had been able to achieve the benchmark single-agent rate for heterogeneous, general nonlinear operators under Markov sampling, with a linear speedup in the variance term (that is not negated by heterogeneity-induced biases.) Our work is the first to close this significant gap by  developing a new algorithm and performing highly non-trivial analysis (despite Assumption 2). Thus, to focus on the topic of closing the gap alluded to above, we did not explore the non-convex setting in this paper.
>
> **Settings under which Assumption 2 holds.** Second, as pointed out in the discussion of the assumptions in Page 6, Assumption 2 covers a fairly broad class of RL settings. To be specific, for TD learning with linear function approximation (LFA) - a setting studied in numerous RL and FRL papers (see Bhandari et al., 2018; Srikant \& Ying, 2019; Khamaru et al., 2020; Tsitsiklis \& Van Roy, 1997; Doan et al., 2019; Liu \& Olshevsky, 2023; Khodadadian et al., 2022; Tian et al., 2024; Wang et al., 2024b), and certain variants of Q-learning with LFA - a well-known RL algorithm to guide the decision-making procedure in environments with uncertainty (see Chen et al., 2022; Zeng et al., 2022), Assumption 2 is **known to hold**.
>
> **Challenge in Relaxing Assumption 2.** In response to the Reviewer's interesting question, we attempted to re-do our analysis without assuming strong-monotonicity, i.e., under a non-convex setting. However, it appears that even for a single-agent setting under Markov noise, this is quite challenging, and, in fact, an **open problem**. To see why this is the case, suppose we try to set up a one-step recursion like we do in Lemma 4 of our paper in Appendix C. Invariably, because of Markovian noise, there will be a Markovian bias term that shows up in this recursion. Now, as revealed by Lemma 6 in Appendix C, when one tries to control this Markov bias term, a "bad" error term of the form $\Vert \theta_t - \theta^* \Vert^2$ will emerge, where $\theta_t$ is the iterate at time $t$. Fortunately for us, under strong-monotonicity, there is a "good" negative term (namely, the first term on the R.H.S. of the recursion in Lemma 4) that counteracts the effect of this "bad" error term. In the absence of strong-monotonicity however, there is no such good term to assist us. Thus, unless one assumes a projection step, there is no immediate way we can think of that enables control over the bad error term of the form $\Vert \theta_t - \theta^* \Vert^2$. This issue does not arise for non-convex stochastic optimization under i.i.d. noise since the Markov bias term is absent in such a scenario.
>
> Since we could not make any progress on this front, we tried to search the existing literature to see if any paper has managed to analyze Markov gradient descent for non-convex functions. The only relevant reference that we could find in this regard is [R1] below which conveniently assumes a PL/gradient-domination condition. In essence, this is again a strong-monotonicity condition which leads to the same analysis as in our own paper.
>
> [R1] Thinh T Doan. Finite-time analysis of markov gradient descent. IEEE Transactions on Automatic Control, 2022.
>
> Thus, to sum up, *we are unaware of any technique that can handle non-convex stochastic optimization under Markov noise without a projection step and without a PL condition*. We have highlighted this as an interesting direction in the Conclusions section of our revised paper.

---

> > ### Author Response · Authors · 2025-11-21
> > **Comment to Reviewer kPGF (Part III)**
> >
> > **Response to R3.** We thank the reviewer for this constructive suggestion, and kindly point the reviewer to Appendix D, where **we did report various simulations in our initial draft**, covering both optimization and RL, and each of the key themes the Reviewer alluded to (linear speedup, correction effects, and fast convergence).
> >
> > Concretely, in Appendix D.1, we consider a federated quadratic loss minimization problem; in Appendix D.2, we consider the setting of federated TD learning with linear function approximation, and in Appendix D.3, we consider the federated finite-sum minimization problem with quadratic loss functions.
> >
> > In each of the three settings, we show the following three plots: (i) a noiseless setting where we isolate the effect of heterogeneity by removing the noise in the observations, and we show that the performance of the local SA algorithm is hindered by the heterogeneity bias term, while **our FedHSA algorithm converges exponentially fast to the desired point** (exactly as we wrote in our response to R1); (ii) a setting with additive Markovian noise, and the lower error floor of our algorithm compared with local SA validates the elimination of the heterogeneity bias in FedHSA, and (iii) a comparison in performance between varying numbers of agents for our FedHSA algorithm, demonstrating a lower error floor with an increasing number of agents, aligning with the achieved *linear-speedup effect* of FedHSA.
> >
> > We hope the above discussion resolves your concerns. We are happy to take any follow-up questions. Thank you again for you constructive comments.

---

### Author Response · Authors · 2025-11-21
**General comment to the AE; summary of rebuttal**

Dear AE,

Thank you for handling the submission of our paper. In response to the constructive comments provided by all the Reviewers, we have now crafted a detailed point-to-point rebuttal, and made appropriate changes in the paper (highlighted in red). We look forward to hearing back from the Reviewers.

We take this opportunity to briefly summarize the main contributions and novelty of this work, which, we believe, address the main concerns of Reviewer uWrU.

1. **Research Gap Addressed by our Work.** As we explain in quite some detail in Section 1.1. of our paper, our work is the first to establish a linear speedup in sample-complexity for general heterogeneous federated stochastic approximation problems under Markov sampling. Given that our formulation subsumes several optimization, RL, and fixed-point problems, we believe our result should be of broad interest to the learning and control communities.

2. **On our Algorithm.** Reviewer uWrU noted that a recent paper by Mangold et al. used a bias-correction idea similar to us for *linear* SA problems. In our initial submission, we acknowledged this fact and provided a detailed comparison with this nice paper, explaining that their contributions do not subsume ours in any way. In fact, the bias-correction idea (also related to control-variates/gradient-tracking/variance-reduction etc.,) has existed for quite some time now, but has been explored primarily in the context of optimization. Our work provides the first rigorous analysis of such a scheme for *general non-linear operators in SA* under *Markov sampling*. Notably, the bias-correction algorithm in Mangold et al. is only studied for linear SA problems under i.i.d. sampling. As such, our work considerably expands the theoretical understanding of bias-corrected SA algorithms.

3. **Technical Novelty in Analysis.** As we explain in detail to Reviewer uWrU in our rebuttal, our proof technique is **fundamentally different** from the one in Mangold et al. for linear SA. In particular, their approach is based on studying an unrolled linear time-varying system and involves analyzing a product of random matrices. We do not have any such "matrix-product" structure in our nonlinear setting, and, as such, our proof departs from Mangold et al. right at the first step. In addition, our analysis for Markovian data is significantly more involved relative to both single-agent and federated SA/RL papers that have looked at Markov data in the past. In short, either these papers look at simple tabular problems where iterates remain bounded trivially, or homogeneous FRL settings where there is no need to study complex bias-correction schemes (like us), or assume a projection step to ensure that the iterate-dependent Markov bias term is uniformly bounded. In contrast, the lack of a projection step makes it particularly challenging to control the Markov bias term in our setting. Moreover, such control is not subsumed by the single-agent RL literature either since we need to establish a finer bound on the Markov bias term to establish the linear speed-up effect (not needed in the single-agent case). Finally, we note that our approach to handling correlated data does not involve unnecessarily discarding samples as is often done in the so called ``blocking technique" where one runs the algorithm on a sub-sampled data-stream.

In light of the above points, we believe that there is considerable technical novelty in the analysis we provide which needs to contend with the interplay between heterogeneous non-linear operators, client-drift effects, and Markov sampling. We hope that our proof template will be useful for the study of more complex SA/RL schemes in the future.

---

### Decision · Action_Editor_AipK · 2025-12-13

**Recommendation:** Accept as is

**Audience:**

Yes

**Audience Explanation:**

This work addresses a research gap in the federated learning and reinforcement learning communities by providing a robust, bias-corrected algorithm and analysis for a highly general and realistic setting, including nonlinear operators and Markovian data. Its applicability to problems like policy evaluation and control with function approximation ensures broad interest among TMLR's audience.

**Claims And Evidence:**

Yes

**Claims Explanation:**

The theoretical claims are supported by the detailed mathematical analysis and proofs provided in the appendices. The authors successfully demonstrated the effectiveness of the newly proposed FedHSA algorithm in achieving exact convergence and linear sample-complexity speedup for heterogeneous federated stochastic approximation under Markovian sampling, a finding not previously established in the literature. The evidence is convincing, as noted by all reviewers.